# Transcriptional and epigenetic rewiring by the NUP98::KDM5A fusion oncoprotein directly activates CDK12

Selina Troester [1], Thomas Eder [1], Nadja Wukowits[1], Martin Piontek[1], Pablo Fernández-Pernas[1], Johannes Schmoellerl [1,2], Ben Haladik [3,4], Gabriele Manhart[1], Melanie Allram[1], Margarita Maurer-Granofszky [3], Nastassja Scheidegger [5], Karin Nebral [3,6], Giulio Superti-Furga [4,7], Roland Meisel[8], Beat Bornhauser [5], Peter Valent [9,10], Michael N. Dworzak[3,11], Johannes Zuber [2,12], Kaan Boztug[3,4] & Florian Grebien [1,3,4] ✉

Nucleoporin 98 (NUP98) fusion oncoproteins are strong drivers of pediatric acute myeloid leukemia (AML) with poor prognosis. Here we show that *NUP98* fusion-expressing AML harbors an epigenetic signature that is characterized by increased accessibility of hematopoietic stem cell genes and enrichment of activating histone marks. We employ an AML model for ligand-induced degradation of the NUP98::KDM5A fusion oncoprotein to identify epigenetic programs and transcriptional targets that are directly regulated by NUP98::KDM5A through CUT&Tag and nascent RNA-seq. Orthogonal genome-wide CRISPR/Cas9 screening identifies 12 direct NUP98::KDM5A target genes, which are essential for AML cell growth. Among these, we validate cyclin-dependent kinase 12 (CDK12) as a druggable vulnerability in NUP98::KDM5A-expressing AML. In line with its role in the transcription of DNA damage repair genes, small-molecule-mediated CDK12 inactivation causes increased DNA damage, leading to AML cell death. Altogether, we show that NUP98::KDM5A directly regulates a core set of essential target genes and reveal CDK12 as an actionable vulnerability in AML with oncogenic *NUP98* fusions.

Acute myeloid leukemia (AML) is a heterogeneous malignant disease of the hematopoietic system that is characterized by the accumulation of immature myeloid blasts in the bone marrow (BM). The landscape of somatic mutations in pediatric AML differs substantially from adult AML and is characterized by lower mutational burden[1]. Hence, targeted therapies against recurrent mutations approved for the treatment of adult AML are often unsuitable therapeutic options for pediatric patients.

[1]Department of Biological Sciences and Pathobiology, University of Veterinary Medicine Vienna, Vienna, Austria. [2]Research Institute of Molecular Pathology (IMP), Vienna, Austria. [3]St. Anna Children's Cancer Research Institute (CCRI), Vienna, Austria. [4]CeMM Research Center for Molecular Medicine of the Austrian Academy of Sciences, Vienna, Austria. [5]Division of Oncology and Children's Research Centre, University Children's Hospital Zurich, University of Zurich, Zurich, Switzerland. [6]Labdia Labordiagnostik, Vienna, Austria. [7]Center for Physiology and Pharmacology, Medical University of Vienna, Vienna, Austria. [8]Division of Pediatric Stem Cell Therapy, Department of Pediatric Oncology, Hematology and Clinical Immunology, Medical Faculty, Heinrich-Heine-University, Düsseldorf, Germany. [9]Department of Internal Medicine I, Division of Hematology and Hemostaseology, Medical University of Vienna, Vienna, Austria. [10]Ludwig Boltzmann Institute for Hematology and Oncology, Medical University of Vienna, Vienna, Austria. [11]Department of Pediatrics and Adolescent Medicine, St. Anna Children's Hospital, Medical University of Vienna, Vienna, Austria. [12]Medical University of Vienna, Vienna BioCenter (VBC), Vienna, Austria. ✉e-mail: florian.grebien@vetmeduni.ac.at

Fusion oncoproteins arise from structural chromosomal rearrangements and frequently act as oncogenic drivers, with a particularly high prevalence in pediatric patients[1]. The family of Nucleoporin 98 (*NUP98*) oncofusions features more than 30 distinct partner genes which often comprise factors that harbor DNA-binding homeodomains, plant homeodomain (PHD) fingers and Su(var)3-9, Enhancer-of-zeste, and Trithorax (SET) domains[2–4].

Fusion of the first 14 exons of the *NUP98* gene to the last exons of the *KDM5A* (also *JARID1A*) gene results in a cryptic translocation giving rise to the expression of the chimeric NUP98::KDM5A protein[5–7]. Besides its enrichment in pediatric acute megakaryoblastic leukemia (AMKL)[6], NUP98::KDM5A is the most common NUP98 fusion found in infant AML and is associated with an aggressive form of the disease and particular poor prognosis[7,8]. The catalytic Jumonji C (JmjC) domain of the H3K4 demethylase KDM5A is not present in the chimeric fusion protein. However, the C-terminal PHD chromatin recognition domain of KDM5A is preserved in the NUP98::KDM5A fusion protein and is necessary for its oncogenic potential[9].

We and others have shown that NUP98 fusion proteins are associated with chromatin and induce global transcriptional dysregulation leading to AML[10–12]. Chromatin association of many NUP98 fusion oncoproteins is mediated by the chromatin-binding domains of the fusion partners and often occurs at genomic regions rich in H3K27ac and H3K4me3[12–14]. NUP98 fusion oncoproteins are able to recruit the transcriptional co-activators CBP/p300 and epigenetic regulators, such as histone deacetylase 1 (HDAC1), Non-Specific Lethal, and MLL1/MENIN complexes[12,15,16]. The formation of transcriptional hubs via the aberrant recruitment of transcription factors and chromatin regulators has been explained by the ability of NUP98 fusion proteins to form biomolecular condensates through liquid-liquid phase separation[11,13,17,18]. This is mediated by the FG repeats in the intrinsically disordered NUP98 N-terminus and is crucial for the oncogenic potential of NUP98 fusion proteins[13,17]. Furthermore, expression of NUP98::KDM5A and other *NUP98* oncofusions leads to sustained upregulation of *HOXA* and *HOXB* genes and other transcription factors that induce a stem/progenitor-like phenotype, such as *MEIS1*[9]. However, it is unclear how NUP98 fusion oncoproteins actively induce and maintain these transcriptional programs and which effector proteins are required to orchestrate the observed patterns.

In this work we reason that a better understanding of how NUP98 fusion proteins deregulate gene expression on the epigenetic and transcriptional level will provide a rational basis for the development of tailored therapies. By comparing global chromatin accessibility of AML patient samples featuring different disease subtypes with healthy blood cells, we identify epigenetic patterns that are specific to *NUP98* fusion-expressing AML. Using a model for ligand-induced degradation of the NUP98::KDM5A fusion oncoprotein we find that NUP98::KDM5A maintains activating H3K27ac and H3K4me3 histone marks on target genes. Furthermore, nascent RNA sequencing upon induced fusion oncoprotein degradation reveals direct transcriptional target genes of NUP98::KDM5A. Integration of these data with results from a genome-wide CRISPR/Cas9 screen in NUP98::KDM5A-driven AML cells identifies a core set of 12 essential direct target genes which are critical for the survival of NUP98::KDM5A-driven leukemia cells. Among them, we validate the kinase CDK12 as an actionable target. NUP98::KDM5A-driven AML expresses high levels of CDK12 and is highly sensitive to genetic and pharmacological CDK12 perturbation. In line with the role of CDK12 in regulating the transcription of DNA damage repair genes, CDK12-inactivating small molecules cause increased DNA damage in NUP98::KDM5A AML cells. These results represent the basis for further investigations of CDK12-targeting compounds and the development of tailored treatments for patients with *NUP98* rearrangements.

## Results
### Epigenomic analysis of pediatric AML reveals NUP98 fusion-specific patterns

To gain a better understanding of the epigenetic landscape of *NUP98* oncofusion-driven leukemia, we performed ATAC-seq on primary AML patient samples to determine global patterns of chromatin accessibility. In addition to samples expressing *NUP98::KDM5A* and *NUP98::NSD1* ($n = 4$, each), we included samples from other pediatric AML patients ($n = 7$) with different oncogenic driver mutations (*KMT2A::MLLT3*, *CBFB::MYH11*, *CBFA2T3::GLIS2*, and normal karyotype) (Supplementary Data 1). We also included publicly available ATAC-seq data from adult AML samples and additional pediatric AML samples (of which two expressed *NUP98::NSD1*)[19], as well as data from healthy hematopoietic stem and progenitor cells (HSPC) and mature myeloid blood cells (Supplementary Data 2). Principal component analysis revealed that chromatin accessibility patterns of AML patient samples were broadly distributed across healthy blood cells, with most of them ranging within different stages of hematopoietic progenitors (Fig. 1A and Supplementary Fig. 1A, B). This is in line with the concept of AML arising from progenitor cells with a block of differentiation that are skewed to particular stages of hematopoietic differentiation[20,21]. To derive epigenetic signatures that are specific for *NUP98* fusion-driven AML we next aimed to identify the healthy progenitor subtype that is closest to the *NUP98* fusion-expressing AML cells. Inter-sample distance visualized by minimum spanning tree analysis revealed that granulocyte-monocyte progenitor (GMP) cells are the closest normal counterpart of *NUP98* fusion-expressing AML cells based on chromatin accessibility (Fig. 1B). Comparison of these two cell populations identified 1921 and 6054 regions close to gene promoters that were significantly more or less accessible in *NUP98* fusion-driven AML cells compared to GMP cells, respectively (Supplementary Data 3). Accessibility of the 1921 more open regions was highest in *NUP98::KDM5A* AML cells over hematopoietic stem cells (HSC), while the same regions were not accessible in GMPs and mature myeloid cells (Fig. 1C). The 6054 regions enriched in GMPs are also highly accessible in HSCs and associated with reduced accessibility in mature myeloid cells (Supplementary Fig. 1C). Overall, these analyses show that *NUP98::KDM5A*-expressing AML exhibits a global reduction in chromatin accessibility when compared to healthy progenitors, but a gain in the accessibility of HSC-specific regions.

Genomic regions with high accessibility are often enriched for activating histone marks[22]. Indeed, by performing CUT&Tag (Cleavage Under Targets and Tagmentation) for H3K27ac and H3K4me3 marks in blasts from a *NUP98::KDM5A* patient-derived xenograft (PDX) AML model, we found that the majority of the 1921 genomic regions that are specifically accessible in *NUP98* fusion-driven AML also showed overlapping H3K27ac and H3K4me3 marks (Fig. 1D). Importantly, genes associated with these genomic regions were highly expressed in *NUP98::KDM5A*-positive AML cells compared to other subtypes of pediatric AML[23] (Supplementary Fig. 1D). Analysis of ChIP-seq data from a murine HA-tagged NUP98::KDM5A model[10] (Table 1) showed that genes associated with these 1921 more accessible regions featured particularly high levels of fusion oncoprotein binding (Supplementary Fig. 1E). Finally, transcription factor motif analysis revealed that consensus motifs of GATA- and AP1-families were enriched in genomic regions with high accessibility in *NUP98::KDM5A* AML, while binding sites for ETS and IRF transcription factors were enriched in regions that were less accessible in *NUP98::KDM5A*-expressing cells (Supplementary Fig. 1F).

Unsupervised clustering of ten *NUP98* fusion-expressing versus ten *NUP98* wild-type AML samples revealed a cluster of regions which were more accessible in AML samples expressing *NUP98::NSD1* and *NUP98::KDM5A* (Fig. 1E). In addition, we also found regions with *NUP98::KDM5A*-specific accessibility. These regions might reflect lineage-specific characteristics associated with acute megakaryocytic leukemia (M7) diagnosed in these *NUP98::KDM5A*-expressing samples.

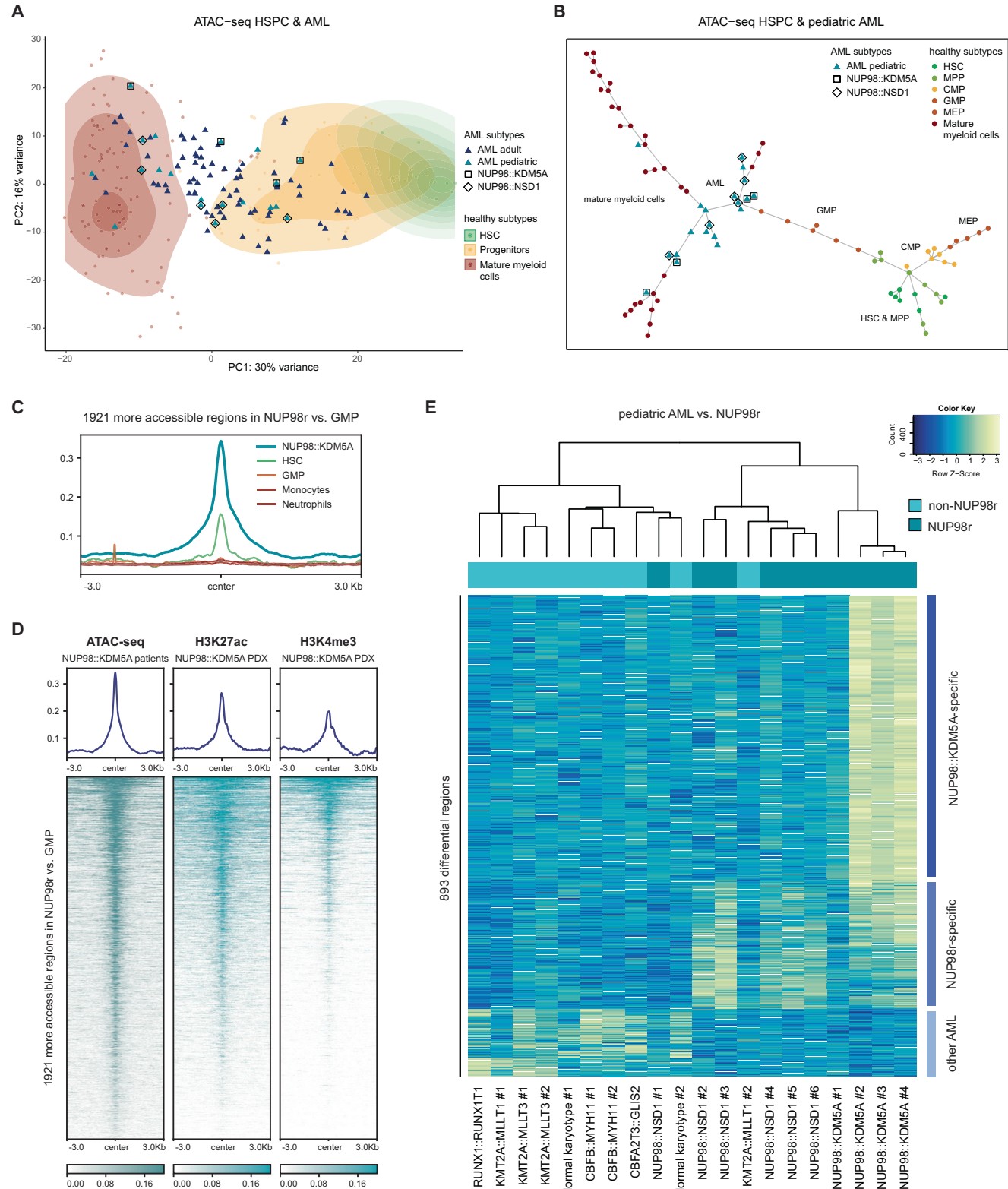

**Fig. 1 | Epigenomic analysis of AML patient data reveals *NUP98* fusion-specific patterns. A** Principal component (PC) analysis of ATAC-seq data from pediatric and adult AML patients and samples from healthy blood cells, including HSCs, progenitor cells, and mature myeloid cells. **B** Minimum spanning tree of distances between ATAC-seq signals of *NUP98* fusion-expressing and other pediatric AML samples and samples of healthy blood cells. **C** ATAC-seq profile plots showing 1921 more accessible regions in *NUP98* fusion-expressing AML vs. GMPs. **D** Heatmap and profile plots of ATAC-seq, H3K27ac, and H3K4me3 CUT&Tag data showing 1921 genomic regions that are more accessible in NUP98 fusion-expressing AML vs. GMPs. **E** Heatmap showing ATAC-seq unsupervised clustering of 893 differentially accessible regions in *NUP98* fusion-expressing AML patient samples compared to NUP98 wild-type pediatric AML patient samples (HSC hematopoietic stem cell, MPP multipotent progenitor, CMP common myeloid progenitor, MEP Megakaryocyte-erythrocyte progenitor, GMP granulocyte-monocyte progenitor). Source data are provided as a Source Data file.

**Table 1 | Murine NUP98::KDM5A cell lines**

| Cell line | Constructs |
|---|---|
| dTAG-NUP98::KDM5A | 1. MSCV-eGFP-IRES-dTAG-3xV5-NUP98::KDM5A<br>2. MSCV-rtTA3-IRES-Nras$^{G12D}$-EF1a-Luc2 |
| dTAG-GFP-NUP98::KDM5A | 1. MSCV-dTAG-3xV5-eGFP-NUP98::KDM5A<br>2. MSCV-rtTA3-IRES-Nras$^{G12D}$-EF1a-Luc2 |
| NUP98::KDM5A (control) | 1. MSCV-eGFP-IRES-NUP98::KDM5A<br>2. MSCV-rtTA3-IRES-Nras$^{G12D}$-EF1a-Luc2 |
| NUP98::KDM5A (Cas9 clone) | 1. MSCV-eGFP-IRES-NUP98::KDM5A<br>2. MSCV-rtTA3-IRES-Nras$^{G12D}$-EF1a-Luc2<br>3. EF1a-Cas9-P2A-BlastiR |
| NUP98::KDM5A (rtTA3)[10] | 1. MSCV-rtTA3-IRES-NUP98::KDM5A<br>2. MSCV-rtTA3-IRES-Nras$^{G12D}$-EF1a-Luc2 |
| Strep/HA-NUP98::KDM5A[10] | 1. MSCV-Strep/HA-NUP98::KDM5A-PGK-Blas-tiR-IRES-mCherry<br>2. MSCV-rtTA3-IRES-Nras$^{G12D}$-Ef1as-Luc2 |
| NUP98::KDM5A (Tet-Off)[10] | 1. TRE3G-GFP-IRES-FLAG-NUP98::KDM5A<br>2. MSCV-Nras$^{G12D}$-IRES-tTA-P2A-Luc2 |

Taken together, our analysis of chromatin accessibility across a multitude of AML patient samples and healthy hematopoietic cell types showed that *NUP98* fusion-driven AML harbors an epigenetic signature that is characterized by increased accessibility of HSC-associated regions and enrichment of activating histone marks.

### Induced NUP98::KDM5A degradation causes terminal differentiation of AML blasts

Given the specific effects of *NUP98::KDM5A* on the epigenome of AML cells we aimed to develop a model that allows the direct investigation of *NUP98::KDM5A*-dependent molecular effects. We used the degradation tag (dTAG) system for ligand-induced protein degradation[24,25] to establish a dTAG-NUP98::KDM5A cell line (Fig. 2A and Supplementary Fig. 2A). The dTAG-NUP98::KDM5A construct was introduced into murine fetal liver-derived HSPCs alongside the clinically observed activated *Nras*$^{G12D}$ co-mutation[3,7,26] and cells were transplanted into sublethally irradiated recipient mice (Supplementary Fig. 2B). In this model, recipient animals developed an aggressive, transplantable AML-like disease (Supplementary Fig. 2C, D)[10]. The immuno-phenotype of leukemic blasts from the BM, spleen and blood of recipient mice was similar to previously established murine *NUP98::KDM5A* models[10], showing moderate levels of c-Kit and high levels of Mac-1 and Gr-1 surface markers, confirming the myeloid origin of the leukemia (Supplementary Fig. 2E, F). The BM of AML-engrafted mice was collected and cultured in vitro until a stably growing dTAG-NUP98::KDM5A-expressing cell line was established (Table 1). ATAC-seq and RNA-seq analysis of dTAG-NUP98::KDM5A cells revealed a strong correlation of chromatin accessibility and gene expression with *NUP98::KDM5A* AML patient data (Supplementary Fig. 2G), further substantiating the clinical relevance of our mouse model for the analysis of the human disease.

Degradation of the dTAG-NUP98::KDM5A fusion oncoprotein was induced by addition of the dTAG13 ligand molecule[24]. Flow cytometric quantification of dTAG-NUP98::KDM5A levels revealed that dTAG13 treatment induces near-complete degradation of the dTAG-NUP98::KDM5A protein without affecting *Nras*$^{G12D}$ levels (Fig. 2B and Supplementary Fig. 2H). 35 nM of the dTAG13 molecule caused efficient NUP98::KDM5A degradation within 15 min of treatment and complete fusion oncoprotein degradation was achieved after 1 h (Fig. 2C, D). Degradation of the NUP98::KDM5A fusion protein leads to a G1-phase cell cycle arrest followed by apoptosis at later stages (Fig. 2E, F). Induced NUP98::KDM5A degradation caused significant morphological changes in AML cells that are characteristic of terminal myeloid differentiation (Fig. 2G), including loss of the progenitor marker c-Kit and an increase of the myeloid differentiation markers Mac-1 and Gr-1 (Fig. 2H–J).

Altogether, these data show that the dTAG-NUP98::KDM5A AML cell line model allows fast and complete dTAG13-induced degradation of the fusion protein. This NUP98::KDM5A-driven model is fully dependent on sustained expression of the oncogenic driver, as its degradation leads to cell cycle arrest, differentiation, and apoptosis of leukemia cells (Supplementary Fig. 2I).

### NUP98::KDM5A actively maintains H3K27ac marks on target genes

To obtain further insights into how NUP98::KDM5A shapes epigenetic patterns to induce malignant transformation of hematopoietic progenitor cells, we investigated changes in the epigenetic landscape upon acute NUP98::KDM5A degradation. Chromatin immunoprecipitation (ChIP)-qPCR showed that the H3K27ac mark was strongly decreased on the promoters of the NUP98::KDM5A target genes *Hoxa9* and *Meis1* already after 3 h of fusion protein degradation (Fig. 3A). While loss of the H3K4me3 mark was also detected in the same regions, its decrease occurred at a slower rate, peaking at 6 h of dTAG13 treatment (Supplementary Fig. 3A). Western blot analysis showed that global levels of H3K27ac and H3K4me3 did not change upon dTAG13 treatment (Supplementary Fig. 3B).

Next, we performed CUT&Tag after 8 h of dTAG13 treatment to investigate global changes of H3K27ac and H3K4me3 distribution on chromatin. 161 genomic regions showed a significant reduction of H3K27ac levels while only 37 regions were associated with reduced H3K4me3 levels upon NUP98::KDM5A degradation. No regions showed an increase in the levels of these histone marks upon degradation of the fusion protein (Fig. 3B and Supplementary Fig. 3C, D). The genomic regions with altered H3K27ac and H3K4me3 levels strongly overlapped and corresponded to 84 and 29 genes, respectively (Supplementary Fig. 3E). The accessibility of chromatin associated with these genes was highly conserved between mouse and human *NUP98::KDM5A* AML cells, further indicating that the NUP98::KDM5A fusion oncoprotein controls similar regulatory circuits in mouse and human AML (Supplementary Fig. 3F). Among the genes associated with reduced H3K27ac levels upon dTAG13 treatment we found known NUP98::KDM5A target genes like *Meis1* and genes of the *Hoxa* cluster, but also several genes that had not been linked to AML before (Fig. 3C and Supplementary Data 4). Gene Ontology (GO) analysis confirmed that genes with differential H3K27ac signals were significantly enriched for regulators of transcription (Supplementary Fig. 3G).

To determine if the observed epigenetic changes upon loss of NUP98::KDM5A are also reflected at the transcriptional level we performed mRNA sequencing (RNA-seq) after 8 h and 24 h of NUP98::KDM5A degradation. We identified four major clusters of genes that depict time-resolved changes in gene expression with different kinetics (Fig. 3D). Differential expression analysis revealed that 153 genes were significantly downregulated and 96 genes were upregulated after 24 h of fusion protein loss (Fig. 3E and Supplementary Data 5). The expression of genes from the *Hoxa* gene cluster, as well as other transcription factors like *Nkx2-3* and *Bcl11a* was already downregulated 8 h after NUP98::KDM5A degradation. However, *Meis1* downregulation appeared to be a later transcriptional event, since the expression of this gene was only significantly downregulated after 24 h. GO analysis confirmed the concomitant loss of stem cell-associated gene expression and the activation of transcriptional programs characteristic of myeloid differentiation, including *Camp*, *Irf8* and *Cebpe* (Fig. 3E and Supplementary Fig. 4A). Chromatin accessibility was significantly higher at genes that were downregulated upon NUP98::KDM5A degradation in mouse and human AML samples (Supplementary Fig. 4B). The intersection of H3K27ac CUT&Tag data with RNA-seq data identified 38 NUP98::KDM5A target genes that exhibited a significant loss of H3K27ac marks and were downregulated after fusion protein

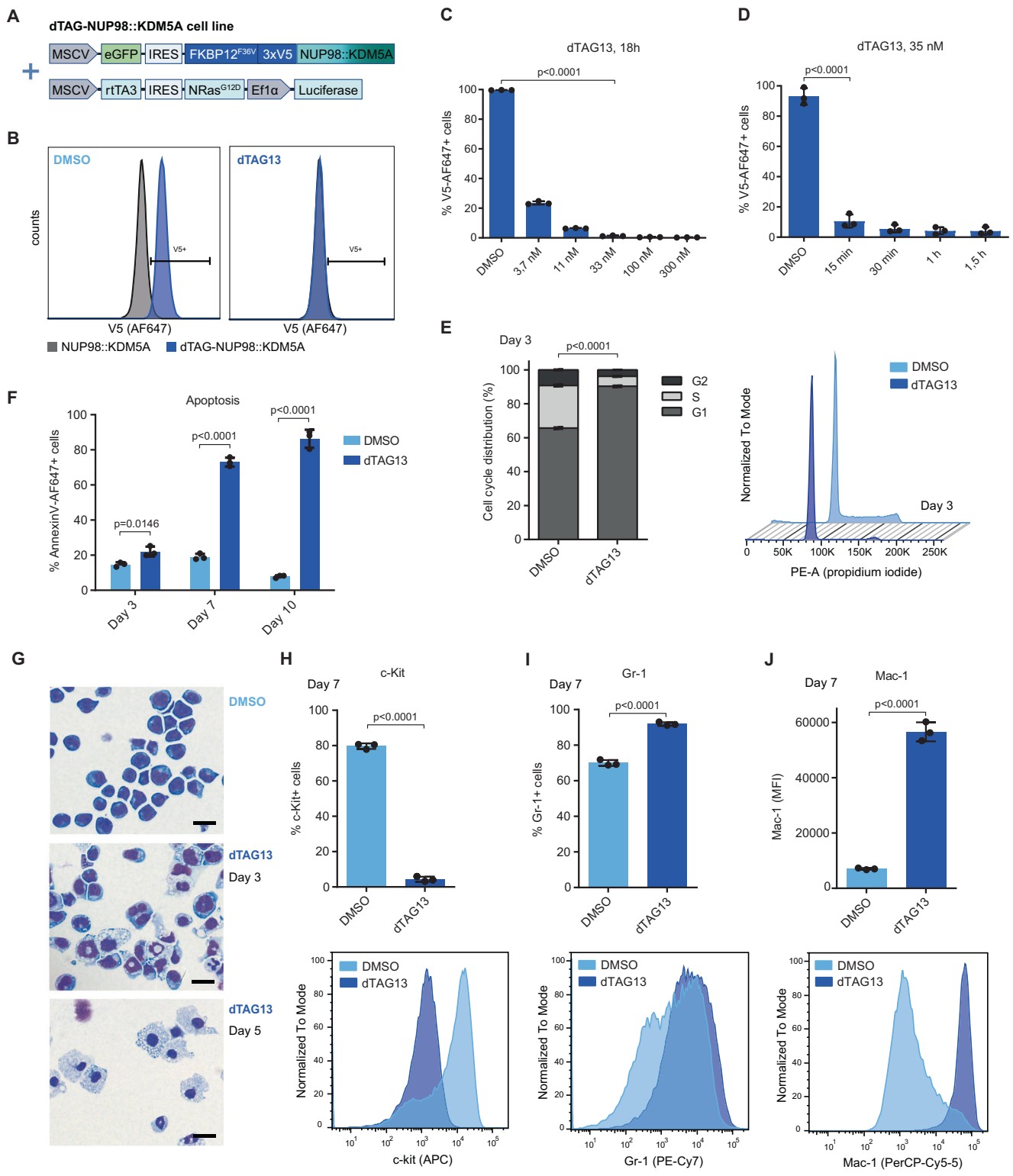

degradation (Fig. 3F, Supplementary Fig. 4C and Supplementary Data 5). These 38 NUP98::KDM5A target genes were characterized by significantly higher H3K27ac signals when compared to all other genes, and chromatin accessibility at these genes was highest in samples of *NUP98::KDM5A* AML patients (Fig. 3G and Supplementary Fig. 4D). Additionally, analysis of ChIP-seq data[10] showed that NUP98::KDM5A fusion oncoprotein binding was exceptionally high in the promoters of these 38 target genes (Supplementary Fig. 4E).

Taken together, these data show that NUP98::KDM5A drives transcriptional programs of hematopoietic progenitors while repressing programs associated with myeloid differentiation via the maintenance of a specific epigenetic signature.

## Identification of direct transcriptional target genes of NUP98::KDM5A

Although our RNA-seq analysis revealed genes that were differentially expressed after NUP98::KDM5A degradation, it is not possible to discern whether these genes are under direct control of the NUP98::KDM5A fusion oncoprotein or if their expression is indirectly affected by changes in NUP98::KDM5A levels. The SLAM-seq (Thiol

**Fig. 2 | Induced NUP98::KDM5A degradation causes terminal differentiation of AML blasts. A** Schematic overview of the constructs used for the generation of the dTAG-NUP98::KDM5A cell line. **B** Representative plots of flow cytometric analysis of intracellular NUP98::KDM5A protein levels upon treatment with dTAG13 or DMSO. (Blue, dTAG-NUP98::KDM5A cell line; grey, untagged NUP98::KDM5A control cell line (Table 1)). **C** Intracellular flow cytometric analysis of NUP98::KDM5A protein levels after treatment of dTAG-NUP98::KDM5A cells with indicated concentrations of dTAG13 for 18 h ($n = 3$ biological replicates, mean ± SD, two-sided, unpaired $t$-test), **D** and after treatment with 35 nM dTAG13 for indicated time points ($n = 3$ biological replicates, mean ± SD, two-sided, unpaired $t$-test). **E** Flow cytometric analysis of cell cycle distribution of dTAG-NUP98::KDM5A cells after treatment with dTAG13 (35 nM) and DMSO for 3 days ($n = 3$ biological replicates,

mean ± SD, two-sided, unpaired $t$-test), including representative histogram plots (right). **F** Flow cytometric analysis of apoptosis after treatment of dTAG-NUP98::KDM5A cells with dTAG13 (35 nM) and DMSO for 3, 7, and 10 days ($n = 3$ biological replicates, mean ± SD, two-sided, unpaired $t$-test). **G** Representative cytospin images showing the morphology of dTAG-NUP98::KDM5A cells after treatment with dTAG13 (35 nM) or DMSO for 3 and 5 days. Scale bar = 20 µm. Representative images of $n = 2$ samples. **H** Flow cytometric analysis of surface expression of c-Kit, (**I**) Gr-1 and (**J**) Mac-1 of dTAG-NUP98::KDM5A cells after treatment with dTAG13 (35 nM) and DMSO for 7 days ($n = 3$ biological replicates, mean ± SD, two-sided, unpaired $t$-test), including representative histogram plots (bottom). Parts of the figure were created in BioRender. Grebien, F. (2025) https://BioRender.com/d32l249. Source data are provided as a Source Data file.

[SH]-linked alkylation for the metabolic sequencing of RNA) approach enables highly sensitive identification of global changes in nascent mRNA transcripts[27,28]. We induced complete degradation of NUP98::KDM5A by dTAG13 treatment and labeled newly synthesized mRNA with 4-thiouridine, which allowed the identification of changes in the synthesis of mRNA transcripts that happen within 1 h after NUP98::KDM5A degradation (Fig. 4A). Differential expression analysis revealed 45 genes whose expression was significantly downregulated upon NUP98::KDM5A loss, while the expression of no gene was significantly induced upon fusion oncoprotein degradation (Fig. 4B and Supplementary Data 6). The expression of all 45 direct NUP98::KDM5A target genes was also found downregulated upon prolonged depletion of the fusion protein in the RNA-seq datasets (8 h and/or 24 h), highlighting the necessity of the driver oncoprotein to maintain high expression of these genes (Fig. 4C). Several of the direct NUP98::KDM5A target genes have been described to play roles in leukemia, such as *CDKN2C*[29], *CDKN1B*[30], *NKX2-3*[31], *IGF2BP3*[32,33], and *MN1*[34]. Other genes in this list are still unexplored or have been implicated in other cancer types, such as *CDK12*[35–40]. Analysis of NUP98::KDM5A ChIP-seq data[10] revealed that the majority of the direct target genes featured high levels of NUP98::KDM5A chromatin binding (Fig. 4D, E). Furthermore, H3K27ac and H3K4me3 marks were strongly enriched at the transcriptional start sites and throughout the gene bodies of these genes (Fig. 4F). dTAG13 treatment caused a strong reduction of the H3K27ac mark on the promoters of the 45 direct NUP98::KDM5A target genes (Fig. 4G). Finally, the direct NUP98::KDM5A target genes were significantly overexpressed in *NUP98::KDM5A*-expressing AML cells compared to control cells that only express the *NUP98* N-terminus and *Nras*^*G12D* [10] and were highly expressed in samples of *NUP98::KDM5A* AML patients (Fig. 4H, I).

In summary, we identified 45 immediate downstream target genes of the NUP98::KDM5A fusion oncoprotein. These direct target genes are bound by NUP98::KDM5A at their promoters, feature high levels of activating histone marks, and are highly expressed.

## Genome-wide CRISPR/Cas9 screening identifies 12 essential direct target genes of NUP98::KDM5A

To identify genes whose mutational inactivation causes a fitness defect in *NUP98::KDM5A* cells, we performed a genome-wide CRISPR/Cas9 loss-of-function screen in a *NUP98::KDM5A* AML cell line (Supplementary Fig. 5A). We generated a single-cell-derived Cas9-expressing variant of murine *NUP98::KDM5A* AML cells (Table 1) and introduced a genome-wide single guide RNA (sgRNA) library to determine the representation of sgRNAs after 14 population doublings. High technical quality of screening results was confirmed by clear separation of core essential and non-essential genes (Supplementary Fig. 5B)[41]. Mutational disruption of 4105 genes caused a fitness defect in *NUP98::KDM5A*-driven cells (Fig. 5A). These genes were enriched for basic cellular functions, but also contained common proto-oncogenes (e.g., *Myc*, *Brd4*) and hematopoietic progenitor cell-specific essential genes like *Myb*[42,43]. Knockout of 645 genes caused a proliferative advantage, including the tumor suppressor genes *Trp53* and *Kdm5c*[44].

Twelve of the 45 direct NUP98::KDM5A target genes were essential for *NUP98::KDM5A*-driven AML cell growth (Fig. 5B). Interestingly, more than half of the essential direct NUP98::KDM5A target genes encode transcription factors, including several members of the *Hoxa* cluster (*Hoxa3, Hoxa7, Hoxa9, Hoxa10*). Additionally, we found the homeobox transcription factors SIX1 and NKX2-3, and the transcription factor BCL11A. While the RNA binding protein IGF2BP3 has previously been shown to be important for AML physiology[32,33,45], the roles of VCPIP1, SLMAP, and EXOC8 in malignant hematopoiesis are unknown. The cyclin-dependent kinase CDK12 is involved in transcriptional elongation and has been mainly studied in association with solid tumors. Among the 45 direct NUP98::KDM5A target genes, knockout of *Mxi1* and *Tnrc18* led to a proliferative advantage.

To validate the essentiality of the 12 target gene candidates using an orthogonal approach, we performed an arrayed competition-based growth proliferation assay using two doxycycline-inducible small hairpin RNAs (shRNA) targeting each transcript in *NUP98::KDM5A* AML cells stably expressing the reverse Tet-transactivator (rtTA3)[10] (Fig. 5C and Table 1). We observed a strong and fast decrease in cell fitness upon knockdown (KD) of the transcription factors *Bcl11a* and *Nkx2-3* that was comparable to the effects of downregulation of the essential gene controls *Myc* and *Myb* and the KD of the *NUP98::KDM5A* fusion transcript itself (Fig. 5D). A similarly strong but slightly slower effect was observed upon *Cdk12* knockdown. Interestingly, we observed a more subtle reduction in cellular fitness upon KD of single *Hoxa* genes, which could be due to redundant functions of genes in this cluster. However, we confirmed that knockdown of all 12 essential direct target genes caused impaired AML cell growth, thus validating the results from the genome-wide CRISPR/Cas9 screen.

dTAG13-mediated NUP98::KDM5A degradation caused downregulated expression and a reduction in H3K27ac marks in the promoters of these 12 direct essential target genes, indicating that the fusion protein maintains high levels of activating histone marks and active transcription of these genes (Fig. 5E). The majority of the 12 essential direct target genes also featured significantly higher H3K27ac marks in human *NUP98::KDM5A* leukemia cells from a PDX model compared to the mean signal of all other genes in this dataset (Fig. 5F).

Collectively, we identified a core set of 12 essential direct target genes of the NUP98::KDM5A fusion oncoprotein. These comprise master transcription factors of the leukemogenic program of *NUP98::KDM5A*-driven AML, as well as other genes which have not been associated with hematologic malignancies.

## CDK12 is an essential direct target gene of the NUP98::KDM5A fusion oncoprotein

Among the essential direct targets of NUP98::KDM5A, the cyclin-dependent kinase 12 (CDK12) stood out, as it is currently the only directly druggable protein in this list. CDK12 plays a critical role in the regulation of transcriptional elongation, as its canonical function is to phosphorylate the C-terminal domain of RNA polymerase II upon its interaction with Cyclin K[46]. Several small molecules to perturb CDK12 have been described[46–50], including kinase inhibitors and the

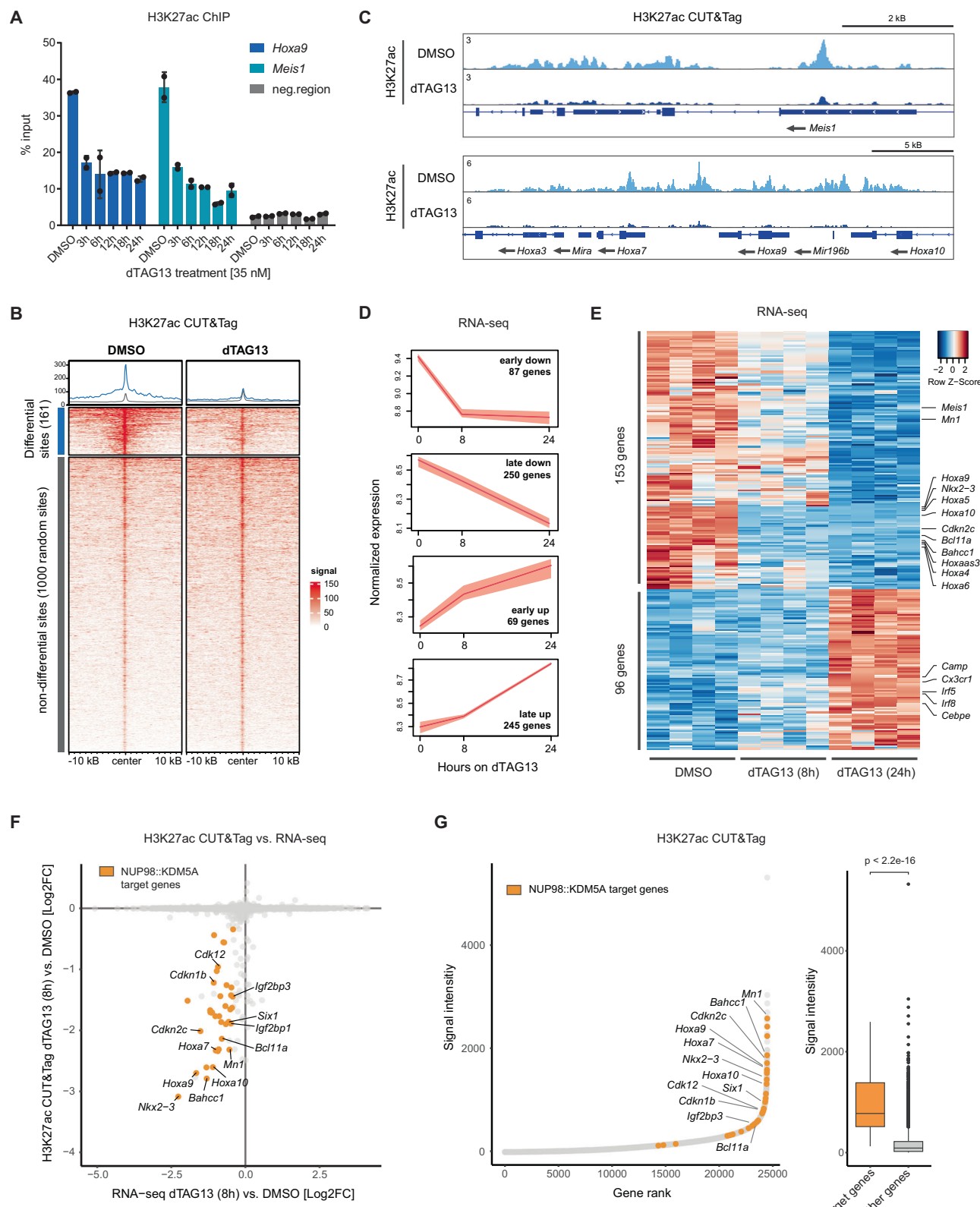

heterobifunctional CDK12-specific degrader BSJ-4-116[51]. CDK12 has been proposed as a functionally relevant, potential biomarker in solid cancers[35,37,52,53], but its role in hematologic malignancies has not been extensively studied.

In a large dataset comprising gene expression data from 2487 pediatric cancer patients[23], *CDK12* expression was slightly but significantly higher in AML patients (*n* = 320) compared to all other

pediatric cancer entities (Fig. 6A). Among AML patients, *NUP98* fusion-driven leukemia patients (*n* = 11) showed even higher *CDK12* expression, with highest levels detected in *NUP98::KDM5A* AML (*n* = 4). In line with this, *CDK12* was the gene with the highest H3K27ac signal in our *NUP98::KDM5A* PDX CUT&Tag data among the 12 essential direct target genes of the fusion oncoprotein (Figs. 5F and 6B). The *CDK12* promoter also showed high H3K4me3 levels in the same PDX sample

**Fig. 3 | NUP98::KDM5A actively maintains H3K27ac marks on target genes.**
**A** H3K27ac ChIP-qPCR in dTAG-NUP98::KDM5A cells for *Hoxa9* and *Meis1* after
treatment with dTAG13 for indicated time points compared to DMSO and a nega-
tive control genomic region (*n* = 2 technical replicates, mean ± SD). **B** Heatmaps
and profile-plots of H3K27ac CUT&Tag data of dTAG13 (35 nM) and DMSO-treated
dTAG-NUP98::KDM5A cells showing 161 significantly downregulated sites com-
pared to 1000 random sites with no significant changes (*n* = 3, FDR < 0.05).
**C** Representative tracks of H3K27ac CUT&Tag data (dTAG13 and DMSO). **D** Time-
series plots showing cluster analysis of up and downregulated transcripts from
RNA-seq data of dTAG-NUP98::KDM5A cells treated with dTAG13 (35 nM) for 8 h or
24 h compared to DMSO (0 h) (*n* = 4 biological replicates per time point). The upper
and lower ends of the bars represent the minimum and maximum expression
values for the respective group. The lines represent the mean expression profile
across replicates for each group. **E** Heatmap of RNA-seq data in dTAG-
NUP98::KDM5A cells after 8 h and 24 h of dTAG13 treatment compared to DMSO,
showing 249 significantly differentially expressed genes at 24 h dTAG13 treatment
(*n* = 4, log2FC > 1 or < −1, padj < 0.05, statistical analysis and *p*-value calculations
were performed using DESeq2). **F** Scatter plot showing log2FC values in gene
expression (QUANT-seq) and H3K27ac levels (CUT&Tag) upon dTAG13 treatment
(35 nM, 8 h) compared to DMSO. **G** Hockey stick plot of H3K27ac CUT&Tag data in
dTAG-NUP98::KDM5A cells (left). Genes are ranked according to normalized read
signal, including a quantitative box plot representation (right, target genes *n* = 38,
other genes *n* = 24,415, two-sided Wilcoxon rank sum test with continuity correc-
tion, *p*-value < 2.2e-16). Data are presented as boxplots where the center line
represents the median, the bounds of the box indicate the first (25th percentile)
and third (75th percentile) quartiles and the whiskers extend to 1.5 × inter-quartile
range from the hinges. Data points beyond this range are shown as individual dots
and represent outliers. Source data are provided as a Source Data file.

and was accessible in *NUP98::KDM5A* patient samples (Fig. 6B and
Supplementary Fig. 6A). The NUP98::KDM5A fusion oncoprotein binds
to the promoter region and the gene body of *Cdk12*, and dTAG13-
induced loss of NUP98::KDM5A caused a reduction in H3K27ac levels
at the *Cdk12* locus and decreased *Cdk12* expression (Fig. 6B). shRNA-
mediated knockdown of *Cdk12* using three individual shRNAs caused a
strong proliferative disadvantage in *NUP98::KDM5A*-driven AML cells,
highlighting the dependency of these cells on CDK12 (Fig. 6C). Pro-
liferation of murine AML cell lines driven by the *KMT2A::MLLT3* fusion
(RN2)[54] and N-terminal *Cebpa* mutations (*Cebpa*[p30/p30])[55] was also
dependent on CDK12, suggesting CDK12 is a broad AML dependency,
in line with a previous report[56] (Supplementary Fig. 6B, C). The kinase
activity of CDK12 was required for the proliferation of *NUP98::KDM5A*
AML cells, as exogenous expression of wild-type CDK12, but not a
kinase-dead mutant of CDK12 (CDK12[D873N])[57] restored proliferation in
cells harboring *Cdk12*-targeting shRNAs (Fig. 6D). Finally, knockdown
of *Cdk12* caused a mild but significant impairment of growth of
*NUP98::KDM5A*-driven AML in vivo (Fig. 6E, F), and immunopheno-
typing of leukemic blasts showed that *Cdk12*-deficient, iRFP670/
CD45.2-positive AML cells displayed reduced levels of the progenitor
marker c-Kit, which is in line with their reduced leukemogenic poten-
tial (Fig. 6G and Supplementary Fig. 6D, E).

## CDK12 is a druggable vulnerability in NUP98::KDM5A-driven leukemia

As the kinase activity of CDK12 was required for the proliferation of
AML cells with NUP98 fusion oncoproteins, we used the CDK12/13
inhibitor THZ531[58] to pharmacologically target CDK12. In line with an
AML-specific vulnerability, primary human AML cells expressing
*NUP98::KDM5A* and *NUP98::NSD1* were 10 times more sensitive to
THZ531-mediated CDK12 inhibition than BM mononuclear cells
(MNCs) and CD34+ progenitor cells from healthy donors (Fig. 7A). In
contrast, the sensitivity of healthy cells and AML cells to the CDK12
degrader BSJ-4-116[51] was comparably low (Supplementary Fig. 7A),
which is likely attributed to the very high potency of BSJ-4-116 that
precluded the faithful evaluation of a potential therapeutic window.
Murine and human AML cells expressing different driver mutations
were also sensitive to THZ531 and BSJ-4-116, in line with reported data[56]
(Fig. 7B and Supplementary Fig. 7B).

CDK12 has been proposed to control the expression of DNA repair
genes[47,51,53]. RT-qPCR and RNA-seq analyses of dTAG-NUP98::KDM5A
cells confirmed that treatment with THZ531 and BSJ-4-116 induced the
downregulation of genes in the DNA damage repair pathway, including
the key factors *Brca2*, *Rad50*, and *Mdc1* (Fig. 7C and Supplemen-
tary 7C, D). As *Cdk12* is a direct target gene of NUP98::KDM5A, we
reasoned that loss of the fusion oncoprotein driver itself would result
in similar gene expression changes. Indeed, the expression of DNA
repair genes was downregulated after NUP98::KDM5A repression in a
doxycycline-controlled *Tet-Off* model (Table 1)[10] (Fig. 7D, E). Addi-
tionally, many genes with functions in DNA damage repair are essential

for the proliferation of *NUP98::KDM5A* AML cells, as their knockout
caused cell depletion in the genome-wide CRISPR screen (Supple-
mentary Fig. 7E). In fact, exogenous expression of wild-type but not
kinase-dead CDK12 was able to partially overcome the proliferation
arrest that is associated with dTAG13-mediated NUP98::KDM5A
degradation (Fig. 7F). To strengthen our hypothesis that
*NUP98::KDM5A*-expressing AML cells depend on intact DNA damage
repair to ensure protection against the accumulation of excessive DNA
damage, we measured the levels of the DNA damage marker phos-
phorylated H2A.X[59] in a dTAG-GFP-NUP98::KDM5A cell line (Table 1).
Indeed, both CDK12 perturbation as well as dTAG13-induced loss of the
NUP98::KDM5A fusion protein caused a significant increase in γH2A.X
foci in these cells (Fig. 7G and Supplementary Fig. 7F, G).

Altogether, we show that NUP98::KDM5A-driven AML is hyper-
sensitive to CDK12 perturbation. Therefore, our observations suggest
that CDK12 inhibition should be investigated further as a potential
therapeutic opportunity for AML patients with NUP98::KDM5A
rearrangements.

## Discussion

Targeted treatment options for pediatric AML are very limited and
the approval of novel therapies is lagging behind recent develop-
ments for the treatment of adult leukemia[60]. In that regard, AML
with *NUP98* rearrangements is particularly problematic, as affected
patients are often refractory to therapy and frequently suffer from
early disease relapse[7,61]. Here, we studied the direct epigenetic and
transcriptional effects of the NUP98::KDM5A fusion oncoprotein
with a focus on identifying its immediate downstream target genes
(Supplementary Fig. 8A). We identify 12 direct transcriptional target
genes of the NUP98::KDM5A fusion oncoprotein that are essential
for AML cell growth. Mechanistically, we find that the strong
dependency of NUP98::KDM5A-driven cells on CDK12 involves
efficient DNA damage repair for the survival of AML cells with
*NUP98* fusions.

Even though global patterns of chromatin accessibility are similar
in adult versus childhood leukemia we found genomic regions that
were selectively accessible in *NUP98*-rearranged AML but not in heal-
thy hematopoietic progenitors and other subtypes of pediatric AML.
These regions were enriched for activating H3K27ac and H3K4me3
histone marks, which is in line with previous results showing that
actively transcribed target genes of NUP98::HOXA9 or NUP98::NSD1
fusion oncoproteins are marked by H3K4me3 and H3K27ac[12–14].

The establishment of a ligand-inducible model for fast and effi-
cient NUP98::KDM5A degradation enabled the investigation of
immediate cellular effects upon complete loss of the oncogenic driver
protein. NUP98::KDM5A degradation induced the loss of H3K27ac
marks at a defined set of genomic regions, but also the loss of
H3K4me3, albeit to a lesser extent. Thus, NUP98::KDM5A chromatin
binding appears to actively maintain the specific epigenetic signature
associated with these genes. As NUP98::KDM5A degradation also

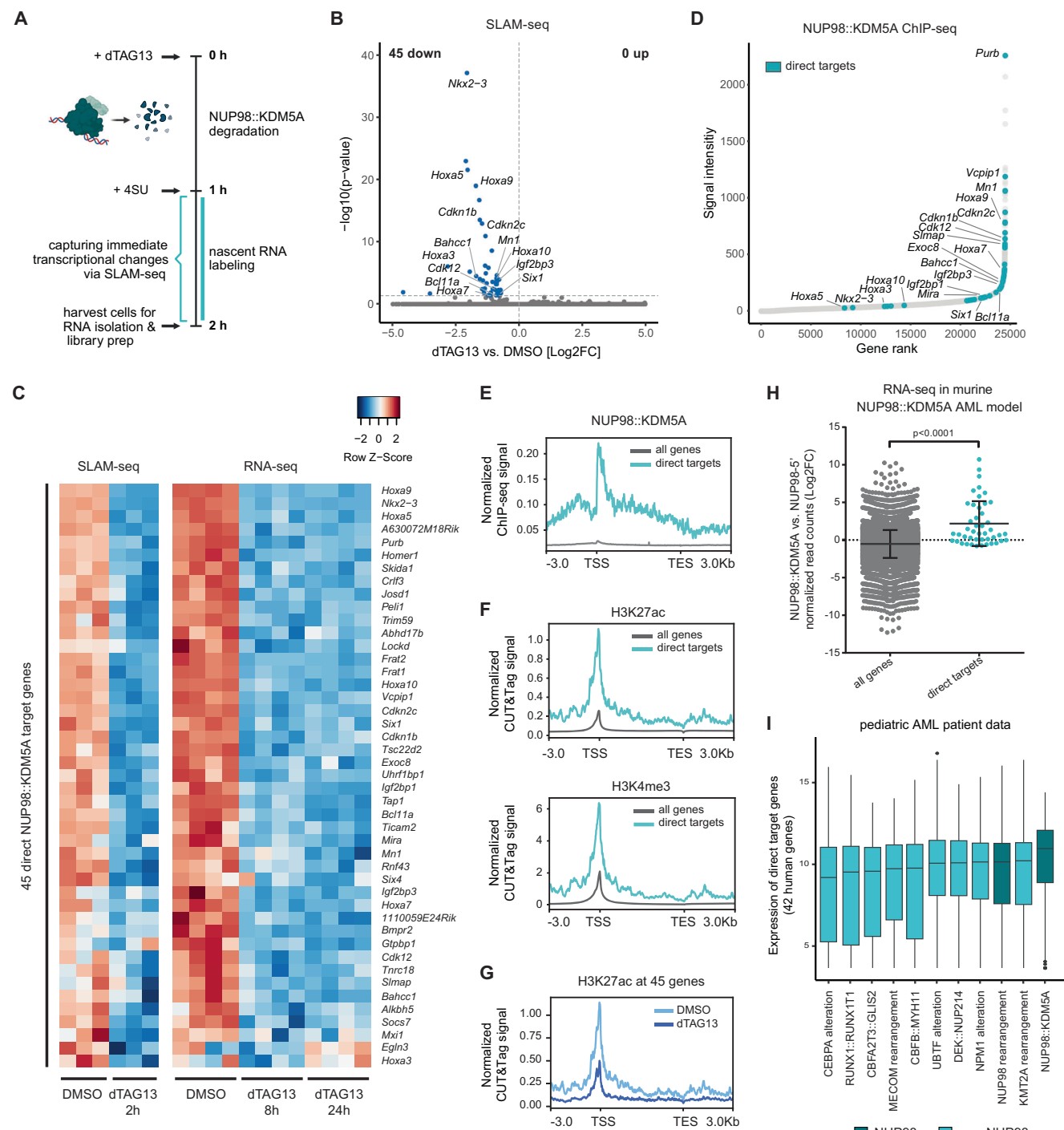

**Fig. 4 | Identification of direct transcriptional target genes of NUP98::KDM5A.**
**A** Schematic representation of the experimental setup for nascent RNA-seq (SLAM-seq) (4SU, 4-thiouridine). **B** Volcano plot of SLAM-seq data in dTAG-NUP98::KDM5A cells showing log2FC and −log10(*p*-value) values after dTAG13 (35 nM, 2 h) vs. DMSO treatment (statistical analysis and p-value calculations were performed using DESeq2). **C** Heatmap showing gene expression changes of the 45 direct target genes 2 h (SLAM-seq), 8 h and 24 h (RNA-seq) after dTAG13 treatment compared to DMSO treatment. **D** Hockey stick plot of ChIP-seq data of NUP98::KDM5A chromatin binding. All genes were ranked according to their normalized read counts and 45 direct targets are highlighted in color. **E** HA-NUP98::KDM5A ChIP-seq data showing normalized read signal of the 45 direct target genes compared to all other genes. **F** H3K27ac and H3K4me3 CUT&Tag data from dTAG-NUP98::KDM5A cells showing normalized read signal of the 45 direct target genes compared to all other genes. **G** H3K27ac CUT&Tag data from dTAG-NUP98::KDM5A cells showing

normalized read signal for the 45 direct target genes after 8 h dTAG13 treatment compared to DMSO. **H** RNA-seq data showing gene expression levels of the 45 direct targets compared to all other genes in murine NUP98::KDM5A AML cells compared to control cells expressing a NUP98 N-terminal fragment (all genes *n* = 18,899, direct targets *n* = 45, two-sided Mann–Whitney *U* test, *p*-value < 0.0001). Horizontal bars represent the mean, error bars +/− SD. **I** Expression of 42 human homologs of direct NUP98::KDM5A target genes in pediatric AML patients from the St. Jude Cloud data repository. Data are presented as boxplots where the center line represents the median, the bounds of the box indicate the first (25th percentile) and third (75th percentile) quartiles and the whiskers extend to 1.5 × inter-quartile range from the hinges. Data points beyond this range are shown as individual dots and represent outliers. Parts of the figure were created in BioRender. Grebien, F. (2025) https://BioRender.com/w52j392. Source data are provided as a Source Data file.

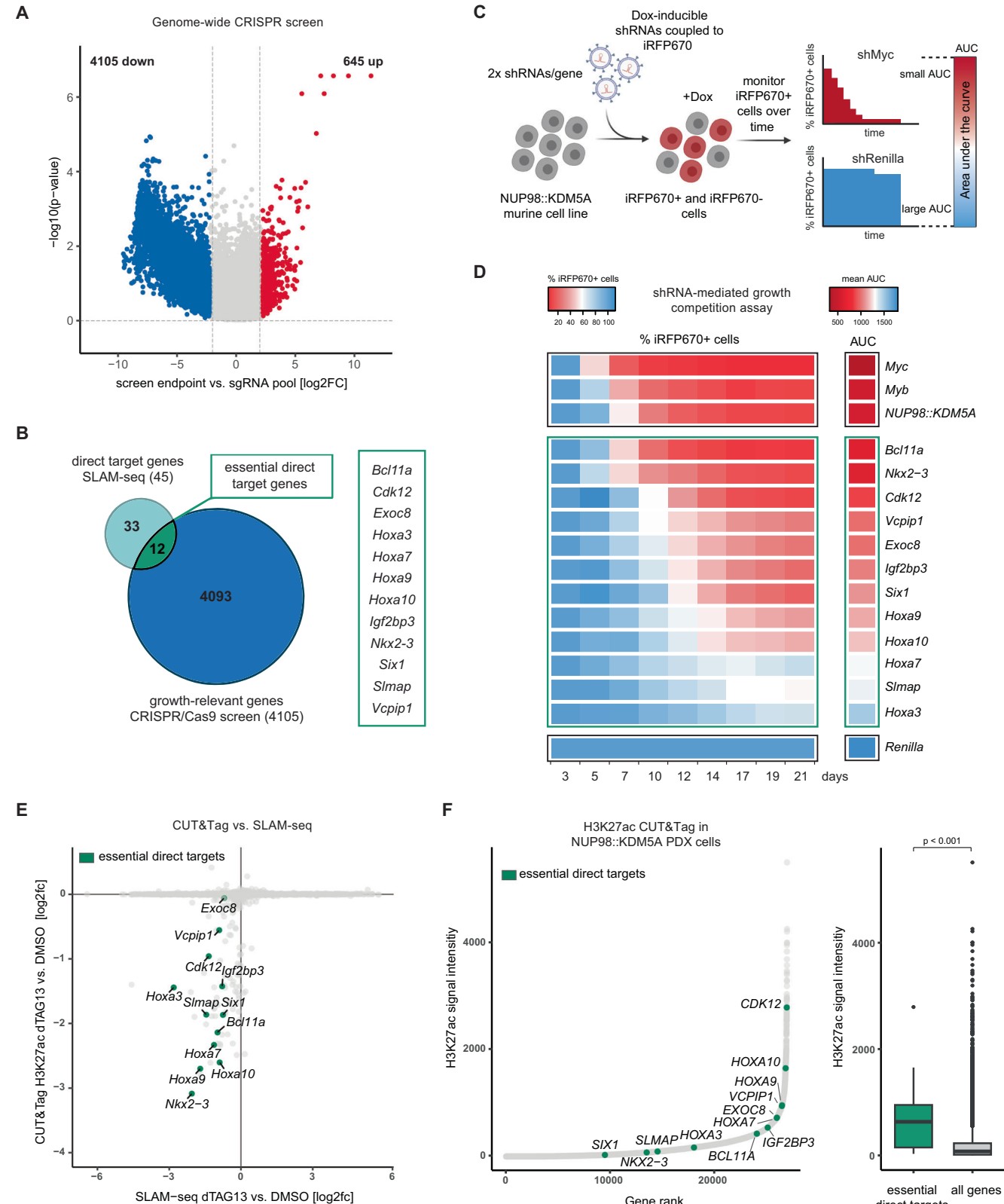

caused the transcriptional downregulation of many of these genes, these data suggest that the NUP98::KDM5A-mediated active maintenance of H3K27ac and H3K4me3 marks at their target genes is involved in the regulation of their expression. Furthermore, the rapid loss of these histone marks upon NUP98::KDM5A degradation suggests that the fusion oncoprotein actively restricts access of histone demethylases and deacetylases to its target genes. As NUP98 fusion proteins were shown to form biomolecular condensates, these higher-order structures likely provide versatile platforms enabling the fast recruitment and/or exclusion of various epigenetic modifiers at target gene loci[11]. While the content and function of NUP98::KDM5A condensates are incompletely understood, these structures may affect additional epigenetic mechanisms that involve the deposition of other histone marks.

**Fig. 5 | Genome-wide CRISPR/Cas9 screening identifies 12 essential direct target genes of NUP98::KDM5A. A** Volcano plot of genome-wide CRISPR/Cas9 screen data from NUP98::KDM5A-expressing cells showing the log2FC and −log10(p-value) values of mean sgRNA abundance at the screen endpoint vs. the sgRNA pool (log2FC > 2.25 and <−2.25, statistical analysis and p-value calculations were performed using MAGeCK). **B** Intersection of SLAM-seq and genome-wide CRISPR screen data. **C** Scheme of competition-based proliferation assay (Dox doxycycline, AUC area under curve). **D** Heatmap representation of proliferation assay data after shRNA-mediated target knockdown (2× shRNAs/gene) showing mean percent of iRFP670-positive cells over 21 days, with positive controls shown at the top and the negative control shown at the bottom (n = 2, biological replicates). **E** Scatter plot of SLAM-seq and H3K27ac CUT&Tag data showing log2FC values upon dTAG13 (35 nM) treatment compared to DMSO and essential direct target genes are

highlighted in color. **F** Hockey stick plot of H3K27ac CUT&Tag data from primary material of a NUP98::KDM5A PDX model (left). Genes are ranked according to normalized read signal and essential direct target genes are highlighted in color including a quantitative box plot representation (right, essential direct targets n = 12, all genes n = 26,917, two-sided Wilcoxon rank sum test with continuity correction, p-value = 0.0001632). Data are presented as boxplots where the center line represents the median, the bounds of the box indicate the first (25th percentile) and third (75th percentile) quartiles and the whiskers extend to 1.5 × inter-quartile range from the hinges. Data points beyond this range are shown as individual dots and represent outliers. Parts of the figure were created in BioRender. Grebien, F. (2025) https://BioRender.com/n44f632. Source data are provided as a Source Data file.

The fast degradation kinetics of the dTAG-NUP98::KDM5A model enabled us to determine immediate transcriptional changes upon loss of the fusion protein. Intersection of this dataset with results from a genome-wide CRISPR/Cas9 screen identified 12 direct target genes of the NUP98::KDM5A fusion oncoprotein that are essential for AML cell growth. This set of genes is enriched for known master regulators of NUP98::KDM5A-driven AML but also features novel candidates that have not been implicated in AML pathogenesis before. It is well known that NUP98 fusion oncoproteins cause overexpression of *HOXA* cluster genes, which are important regulators of development and self-renewal[2,62]. Furthermore, deregulated expression of *NKX2-3* has been described in a variety of other hematologic malignancies[31,63,64]. Interestingly, *NKX2-3* plays a role in megakaryoblastic development and is important in AMKL, which is the AML subtype with a particular high prevalence of the *NUP98::KDM5A* fusion[31]. The SIX1 homeobox transcription factor has also been linked to the maintenance of leukemia stem cells in AML and is important for oncogenic transformation in KMT2A fusion-driven leukemia[65–67]. The BCL11A transcription factor represses the expression of fetal hemoglobin and represents an attractive target in sickle cell anemia and β-thalassemia[68]. In AML, BCL11A was shown to repress PU.1 target genes, which is in line with our observation that PU.1 motifs are less accessible in *NUP98::KDM5A* AML cells, and high expression of *BCL11A* is linked to worse prognosis in AML patients[69,70]. The oncogenic properties of the RNA-binding protein IGF2BP3 have been studied across many different cancer types, including AML[71,72]. IGF2BP3 targets mRNAs encoding important factors in leukemia, like *MYC*, *CDK6*, *HOXA* genes, and *EPOR*, causing their stabilization and overexpression in leukemia cells[32,33,45]. The direct regulation of *Igf2bp3* by NUP98 fusion oncoproteins could therefore represent a mechanism that contributes to the over-expression of *Cdk6* that we have previously reported[10]. The deubiquitinating enzyme VCPIP1, the exocyst complex member EXOC8 and the Sarcolemma associated protein SLMAP were not known to play roles in malignant hematopoiesis and will have to be investigated in future studies.

The direct NUP98::KDM5A target CDK12 is an important regulator of transcriptional elongation via phosphorylating Serine 2 at the C-terminal domain of RNA polymerase II[46]. Additionally, CDK12 is involved in transcription termination, pre-mRNA splicing, RNA turnover and suppression of intronic polyadenylation[47,48]. CDK12 is important for the transcription of DNA damage repair genes, in particular those related to homologous recombination (HR)-mediated repair, as these transcripts are long and harbor multiple intronic polyadenylation sites[47]. In solid cancers, CDK12 has been associated with both tumor suppressive and oncogenic functions[73]. Our data show that CDK12 is crucial for the growth of *NUP98::KDM5A*-driven leukemia in vitro and in vivo and that small-molecule-mediated perturbation of CDK12 leads to a profound downregulation of genes involved in DNA repair, as reported in previous studies[47,48,53,74]. In line with *CDK12* being a direct target of NUP98::KDM5A, downregulation of the fusion oncoprotein itself phenocopied the effects of CDK12 loss.

We propose that expression of NUP98 fusion proteins and other AML oncogenes cause oncogene-induced replication stress, which has been described as a source of genomic instability in cancer[75,76]. In line with this, CDK12 has been shown to protect genome integrity by preventing transcription-replication conflicts[77,78]. However, its role in preventing excessive accumulation of DNA damage via the efficient induction of repair mechanisms likely represents one aspect amongst the broader role of CDK12 in mediating leukemia cell survival (Supplementary Fig. 8B).

Preclinical studies have demonstrated that CDK12 is amenable to pharmacological targeting[73]. Our data as well as a recent study using several human cell lines show high sensitivity of AML cells towards dual CDK12/13 inhibition, suggesting that CDK12 might be a more broadly relevant target for hematologic malignancies[56]. While THZ531 has not been optimized for in vivo use, animal studies using the precursor compound THZ1 and related agents showed effective cancer killing without reports of hematotoxicity[38,49,50]. Further preclinical studies will be required to investigate the potential of CDK12-targeting compounds in *NUP98*-oncofusion-driven pediatric AML and related diseases.

## Methods

All research complies with relevant ethical regulations of the University of Veterinary Medicine Vienna. Fresh-frozen samples of primary bone-marrow mononuclear cells (MNCs) were obtained from the biobank of the St. Anna Children's Hospital Vienna, Austria. Collection of samples and their use in research was performed with clearance of the appropriate Ethics committee (ethics vote No.1500/2014 of the Ethics Committee of the Medical University Vienna, Vienna, Austria). Samples were collected at diagnosis from patients enrolled in the AML-BFM studies. All patients or their respective legal guardians gave written informed consent prior to the study. The use of human material was approved by the institutional review boards in accordance with the Declaration of Helsinki.

### Expression vectors and viral vectors

The dTAG-NUP98::KDM5A construct was generated from a plasmid for constitutive expression of NUP98::KDM5A as previously described[10]. The FKBP12[F36V] degradation tag (dTAG) and a triple-V5 affinity tag were fused to the NUP98 N-terminus of the of NUP98::KDM5A fusion protein, as we have previously shown that the addition of exogenous tag sequences does not affect the leukemogenic potential of NUP98::KDM5A[10]. The FKBP12[F36V]-tag[24] and 3× V5-tag were ordered as one double-stranded DNA fragment and introduced in frame with the N-terminus of the NUP98 sequence via Gibson assembly. The sequence of the 3× V5-tag was kindly provided by the J. Zuber lab. The dTAG-eGFP-NUP98::KDM5A construct was generated from the dTAG-NUP98::KDM5A construct by restriction enzyme-guided removal of the eGFP-IRES-FKBP12[F36V]-3 × V5 sequence and introduction of an in frame FKBP12[F36V]-3 × V5-eGFP double-stranded DNA fragment via Gibson assembly. MSCV-rtTA3-IRES-Nras[G12D]-EF1as-Luc2 has been

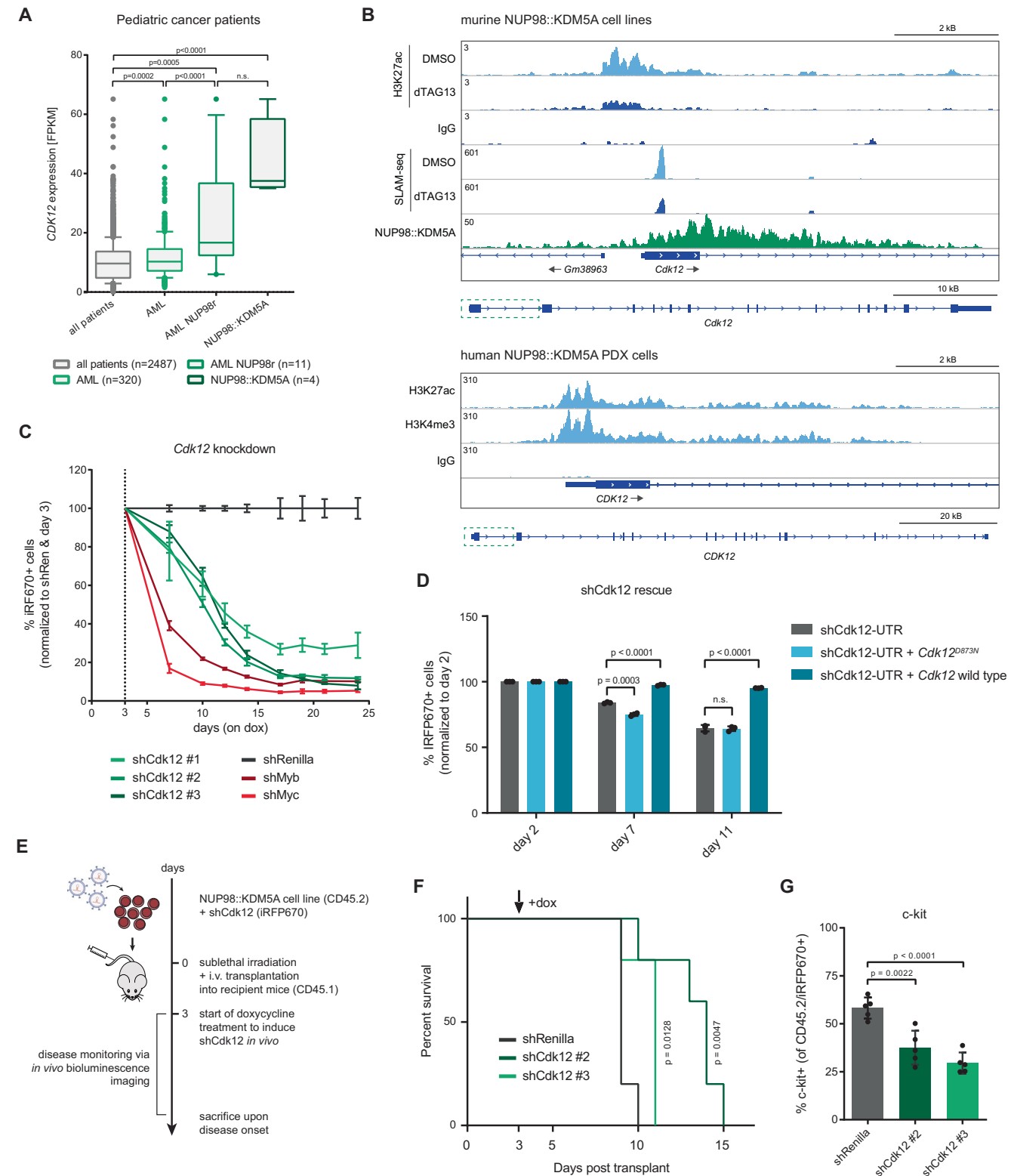

described previously[41]. For the generation of the CDK12 wild type overexpression construct pCR8/GW_TOPO_Cdk12_Nterm_FlagHa (Addgene plasmid #127177) expressing the mouse *Cdk12* transgene (NM_001109626.1 isoform) was cloned into pLX303-Gw (Addgene plasmid #25897) using Gateway LR Clonase II Enzyme Mix (Invitrogen, USA) according the manufacturer's protocol. The CDK12 kinase-dead variant CDK12^D873N was designed as the murine equivalent of the previously published human CDK12^D877N kinase-dead mutant[57]. The

CDK12^D873N overexpression construct was generated by inserting the corresponding point mutation into the CDK12 wild type expression construct using the Q5 Site-Directed Mutagenesis Kit (NEB, USA) and the NEBase Changer tool (primers in Supplementary Data 7) according to the manufacturer's protocol. Plasmid lentiCas9-Blast was a gift from Feng Zhang (Addgene plasmid #52962)[79]. Lentiviral plasmid psPAX2 (Addgene plasmid #12260) and pMD2.G (Addgene plasmid #12259) were gifts from Didier Trono. Retroviral plasmid pCMV-gag/pol was

**Fig. 6 | CDK12 is a direct essential target gene of NUP98::KDM5A. A** Gene expression data for *CDK12* of 2487 pediatric cancer patients, AML patients ($n = 320$), AML with *NUP98* rearrangements ($n = 11$) and AML with *NUP98::KDM5A* ($n = 4$) from the St. Jude Cloud data repository (two-sided, unpaired *t*-test). Data are presented as boxplots, where the center line represents the median, and the bounds of the box indicate the first (25th percentile) and third (75th percentile) quartiles. The whiskers extend down to the 10th percentile and up to the 90th percentile. Data points beyond this range are shown as individual dots and represent outliers. **B** Representative IGV tracks of H3K27ac CUT&Tag data and SLAM-seq data from dTAG-NUP98::KDM5A cells, and HA-NUP98::KDM5A ChIP-seq data showing the murine *Cdk12* locus (top). H3K27ac and H3K4me3 CUT&Tag data in NUP98::KDM5A PDX cells showing the human *CDK12* locus (bottom) (PDX patient-derived xenograft). **C** Competition-based proliferation assay upon doxycycline-inducible shRNA-mediated knockdown of *Cdk12* in a murine NUP98::KDM5A (rtTA3) AML cell line, showing percentage of iRFP670-positive cells over 24 days

($n = 3$ biological replicates). The upper and lower ends of the bars represent the minimum and maximum percent of iRFP670+ cells (normalized to shRen and day 3) for the respective point. **D** Competition-based proliferation assay upon doxycycline-inducible shRNA-mediated knockdown of *Cdk12* in dTAG-NUP98::KDM5A cells stably expressing exogenous CDK12 wild type and CDK12$^{D873N}$, showing percentage of iRFP670-positive cells on days 2, 7, and 11 on doxycycline ($n = 3$ biological replicates, mean ± SD, two-sided, unpaired *t*-test). **E** Schematic representation of the experimental setup for in vivo shRNA-induced knockdown of *Cdk12*. **F** Kaplan–Meier curve of sublethally irradiated recipient mice transplanted with doxycycline-inducible shCdk12-expressing NUP98::KDM5A (rtTA3) AML cells ($n = 5$ biological replicates, Log-rank (Mantel-Cox) test). **G** Flow cytometric analysis of CD45.2/iRFP670+ AML blasts from bone marrow of moribund mice showing the percentage of c-Kit surface marker expression ($n = 5$, mean ± SD, two-sided, unpaired *t*-test). Source data are provided as a Source Data file.

acquired from Cell Biolabs, USA. The mouse genome-wide Vienna sgRNA library was used as previously described[80]. shRNA sequences were cloned into pRRL-TRE3G-iRFP670-miR-PGK-NeoR (Supplementary Data 7)[81].

## Cell culture

Murine fetal liver cells were cultivated in DMEM/IMDM (50:50% vol/vol, Gibco, USA), supplemented with 10% heat-inactivated fetal bovine serum (FBS) (Merck, Germany), 100 U/ml penicillin, 100 µg/ml streptomycin, 4 mM L-Glutamine and 50 µM 2-Mercaptoethanol (all Gibco, USA) in the presence of 100 ng/ml mouse stem cell factor (mSCF), 10 ng/ml mIL-3 nd 10 ng/ml mIL-6 (all from PeproTech, Vienna, Austria). Ex vivo-isolated leukemia cells were cultivated in RPMI 1640 (Gibco, USA), supplemented with 10% FBS, 100 U/ml penicillin, 100 µg/ml streptomycin, 4 mM L-Glutamine, 100 ng/ml mSCF, and 10 ng/ml mIL-3 (all from PeproTech, Austria). After one week, media of the ex vivo-isolated cells was switched to RPMI 1640 supplemented with 10% FBS, 100 U/ml penicillin and 100 µg/ml streptomycin, 4 mM L-Glutamine, 1 mM Sodium pyruvate (Merck, Germany), 50 µM 2-Mercaptoethanol (Gibco, USA) and 20 mM 4-(2-Hydroxyethyl)−1-piperazineethanesulfonic acid (HEPES) (Merck, Germany). Stable cell lines were established by continuous culture for over 4 weeks and all murine NUP98::KDM5A cell lines (Table 1) were maintained as described above. Murine AML cell lines driven by *NUP98::NSD1*[10], *KMT2A::MLLT3* (RN2)[54], *RUNX1::MECOM*[41], N-terminal *CEBPA* mutant (*Cebpa*$^{p30/p30}$)[55], and N/C-terminal *CEBPA* mutants (CNC)[82] were maintained as previously described. All human cell lines, Nomo-1 (ACC 542), HL-60 (ACC 3), MOLM-13 (ACC 554), Kasumi-1 (ACC 220), OCI-AML3 (ACC 582) and K562 (ACC 10) were maintained using RPMI 1640 supplemented with 10% FBS, 100 U/ml penicillin and 100 µg/ml streptomycin and 4 mM L-Glutamine and were obtained from DSMZ. Platinum-E (Cell Biolabs, USA) and Lenti-X 293 T cells (Takara, France) were cultivated in DMEM (Gibco, USA) supplemented with 10% FBS, 100 U/ml penicillin, 100 µg/ml streptomycin and 2 or 4 mM L-Glutamine, respectively. Primary patient-derived leukemia cells, human BM CD34+ Progenitor Cells (#2M-101C) and Human BM MNC (#2M-125C) (Lonza, Switzerland) and PDX cells were cultured in IMDM (Gibco, USA) containing 15% BIT 9500 Serum Substitute (STEMCELL Technologies, Canada), 100 ng/ml SCF, 50 ng/ml Flt3-Ligand, 20 ng/ml IL-3, 20 ng/ml G-CSF (all from PeproTech, Austria), 100 µM 2-Mercaptoethanol, 50 µg/ml gentamicin (Gibco, USA), and 10 µg/ml ciprofloxacin (Merck, Germany) plus 500 nM SR1 (APExBIO, USA) and 1 µm UM729 (APExBIO, USA) as previously described[83]. All cells were cultured at 37 °C, 5% $CO_2$, and 95% humidity. Human cell lines were validated using STR profiling. Primary cell lines were not authenticated, since testing is not applicable for primary samples and primary cell lines. All cell lines were routinely tested for mycoplasma contamination and confirmed negative.

## Virus production and infections

For retrovirus production, Platinum-E cells were co-transfected with 20 µg transfer vector and 5 µg pCMV-gag/pol using Polyethyleneimine (branched, MW 25.000, Sigma-Aldrich, Austria). Virus supernatant was harvested 48 h and 56 h post transfection, filtered (0.45 µm) and supplemented with recombinant mIL-3 (10 ng/ml), mIL-6 (10 ng/ml) (both from PeproTech, Austria) and mSCF (100 ng/ml). Murine progenitor cells were spinoculated with viral supernatants (1:2 diluted) for 45 min at 37 °C at $500 \times g$ in the presence of polybrene (4 µg/ml) (Merck, Germany). For lentivirus production Lenti-X 293 T cells were transfected with 4 µg transfer vector, 2 µg psPAX2 and 1 µg pMD2.G using PEI. Lentivirus was harvested 48 h and 72 h post transfection and filtered (0.45 µm). Target cells were spinoculated after addition of virus supernatant (1:5 diluted) followed by centrifugation for 90 min at 37 °C at $1000 \times g$ in the presence of polybrene (5 µg/ml).

## Transplantation-based models and cell line generation

NUP98::KDM5A (control) and dTAG-NUP98::KDM5A cell lines were generated by retroviral co-transduction of either MSCV-eGFP-IRES-NUP98::KDM5A or MSCV-eGFP-IRES-dTAG-3 × V5-NUP98::KDM5A together with MSCV-rtTA3-IRES-Nras$^{G12D}$-EF1a-Luc2 of murine fetal liver cells (C57BL/6, Ly5.2), followed by continuous culture for 12 weeks. $4 \times 10^6$ cells (96% GFP+) cells were transplanted into sublethally irradiated (4,5 Gy) recipient mice (C57BL/6, Ly5.1) via tail vain injection. Disease progression was monitored by whole-body luminescence imaging as previously described[84]. Mice were euthanized upon disease onset and BM and spleen cells were harvested. Stable cell lines were established by continuous culture of BM cells for 4 weeks without supplemented cytokines. For the secondary transplant of the dTAG-NUP98::KDM5A model, $2 \times 10^6$ GFP+ spleen cells (93% GFP+) of primary transplant mice were transplanted into recipient mice (C57BL/6, Ly5.1) via tail vain injection. The dTAG-GFP-NUP98::KDM5A cell line was generated by retroviral co-transduction of MSCV-dTAG-3 × V5-eGFP-NUP98::KDM5A and MSCV-rtTA3-IRES-Nras$^{G12D}$-EF1a-Luc2 of murine fetal liver cells (C57BL/6, Ly5.2). $4 \times 10^6$ cells (6% GFP+) were transplanted into sublethally irradiated (4.5 Gy) recipient mice (C57BL/6, Ly5.1) via tail vain injection. Stable cell lines were established by continuous culture of harvested BM cells of leukemia-bearing mice for 4 weeks without supplemented cytokines. For in vivo shRNA-mediated knock down of *Cdk12*, the NUP98::KDM5A (rtTA3, Ly5.2) cell line was lentivirally transduced with indicated doxycycline-inducible shRNA constructs and selected with G418 (1 mg/ml, InvivoGen, France). $1 \times 10^6$ cells were transplanted into sublethally irradiated (4.5 Gy) recipient mice (C57BL/6, Ly5.1) via tail vain injection. Disease progression was monitored by whole-body luminescence imaging. Upon detectable engraftment at day 3 doxycycline hyclate (4 mg/ml) (Merck, Germany) and Sucrose (20 mg/ml) (Merck, Germany) was dissolved in water and administration was started and exchanged every 2 and 3 days. Mice were euthanized upon disease onset and BM cells were harvested and

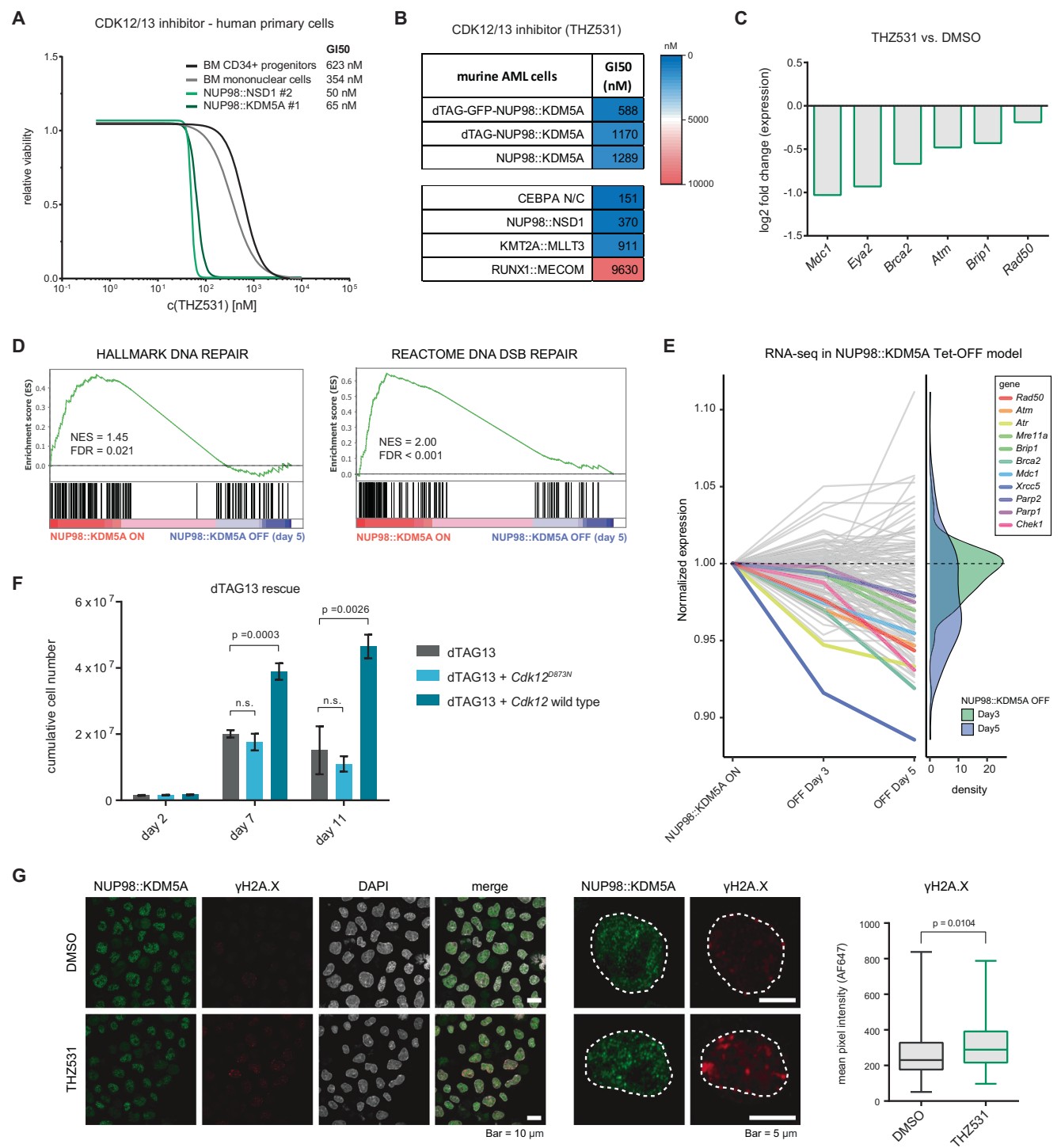

**Fig. 7 | CDK12-controlled DNA repair is a vulnerability in NUP98::KDM5A-driven leukemia. A** Cell viability assay of primary human *NUP98*-rearranged AML cells, healthy donor BM MNC and CD34+ progenitors treated with indicated concentrations of THZ531 for 3 days (*n* = 3 technical replicates). **B** GI50 values from cell viability assays of murine AML cells treated with THZ531 for 3 days. **C** RT-qPCR analysis of dTAG-NUP98::KDM5A cells treated with THZ531 (2 μM, 24 h) showing log2FC values (*n* = 3 technical replicates). **D** Gene set enrichment analysis of RNA-seq data from a doxycycline-controlled NUP98::KDM5A cell line (*Tet-Off*) after 5 days of doxycycline-induced NUP98::KDM5A downregulation compared to DMSO. **E** Time series plot of *Tet-Off* RNA-seq data showing gene expression levels of DNA double strand break repair genes at day 3 and day 5 after NUP98::KDM5A downregulation normalized to NUP98::KDM5A-expressing cells, highlighting selected genes (left), with a density plot of normalized expression of all genes

(right). **F** Proliferation assay of dTAG-NUP98::KDM5A cells stably expressing exogenous CDK12 wild type and CDK12$^{D873N}$, showing the cumulative cell number on days 2, 7, and 11 of dTAG13 treatment (initial dTAG13 concentration, 35 nM, maintained at 3,5 nM after day 2) (*n* = 3 biological replicates, mean ± SD, two-sided, unpaired *t*-test). **G** Representative images of γH2A.X immunofluorescence staining of dTAG-GFP-NUP98::KDM5A cells treated with THZ531 (2 μM, 24 h) (left) and quantification of γH2A.x signal (right). Midline represents the median, bounds of the box represent the inter-quartile range (Q1 to Q3), and whiskers represent minimum and maximum values (DMSO: *n* = 125 nuclei (technical replicates), THZ531: *n* = 131 nuclei (technical replicates) two-sided, unpaired *t*-test, right). Representative images of *n* = 8 samples. Source data are provided as a Source Data file.

analyzed. In all cases leukemia development was regularly monitored and animals were euthanized before maximal disease burden was reached. All animal studies were approved by the Ethics and Animal Welfare Committee of the University of Veterinary Medicine, Vienna in accordance with the University's guidelines for Good Scientific Practice and authorized by the Austrian Federal Ministry of Education, Science and Research (ref BMBWF 68.205/0199-V/3b/2018, 2022-0.874.042) in accordance with current legislation. Animals were maintained on a 12-h light-dark cycle with a room temperature of $21 \pm 1 \,°C$ and a relative humidity of 40–55%. Animals were housed in specific pathogen-free conditions, fed a standard chow (V1534; Ssniff, Germany), and had access to acidified drinking water *ad libitum*. Qualified personnel executed all animal experiments and monitored animal welfare daily. Upon the first signs of illness (loss of escape reflex, white paws), the mice were humanely and painlessly euthanized by cervical dislocation. The PDX model of a *NUP98::KDM5A* AML sample was established as previously described under approval of the legal authorities (131/19)[85]. Sex was not considered in the study design because no sex bias for *NUP98::KDM5A*-driven AML has been reported.

### Intracellular flow cytometry

NUP98::KDM5A (control) and dTAG-NUP98::KDM5A cells were treated with indicated concentrations of dTAG13 (Merck, Germany) or DMSO for indicated durations. $1 \times 10^6$ cells were harvested, washed with PBS and stained with Zombie Aqua Fixable Viability Dye (1:1000, BioLegend, USA) for 10 minutes. After PBS wash, cells were fixed with 2% phosphate-buffered formalin solution (Roti-Histofix 4,5%, Carl Roth, Germany) for 15 min. After washing with PBS, cells were permeabilized with 0.2% TritonX-100 (PanReac AppliChem, Germany) in PBS supplemented with 10% FBS for 15 min, followed by incubation in 0.1% TritonX-100 in PBS supplemented with 10% FBS for 30 min. Next, cells were incubated with anti-mouse CD16/CD32 antibody (1:200, Mouse BD Fc Block, clone 2.4G2, BD Biosiences, Germany) for 10 min, followed by the direct addition of the 2× primary antibody staining solution (1:400 final dilution) in 0.1% TritonX-100 buffer (V5-Tag, clone D3H8Q, Cell signaling, USA) and incubation for 45 min. After a wash with 0.1% TritonX-100 buffer, cells were incubated in secondary antibody staining solution (1:200, Anti-rabbit IgG AF647, #A-21246, Thermo Fisher Scientific, USA) for 45 min. All incubation steps were performed at room temperature with light protection. Stained cells were washed twice with 0.1% TritonX-100 buffer and samples were analyzed on a BD FACSCanto II (BD Biosiences, Germany). Data analysis was done using the FlowJo software package (FlowJo LLC, Ashland, USA).

### Flow cytometry

For characterization of ex-vivo-isolated leukemia blasts, cells from BM and spleen were washed with PBS and resuspended in PBS with 0.5% FCS, followed by staining for 30 min with dilutions (1:200) of the following antibodies (all from Biolegend, USA): anti-mouse CD45.2 PE-Cy7 (clone 104), anti-mouse CD11b/Mac-1 PerCP-Cy5.5 (clone M1/70), anti-mouse Gr-1/Ly-6C BV421 (clone RB6-8C5), anti-mouse CD117/c-Kit APC (clone 2B8) and anti-mouse CD117/c-Kit biotin (clone 2B8) together with anti-streptavidin BV421. Engraftment of murine leukemia was assessed by measurement of GFP fluorescence marker expression or staining with anti-CD45.2 antibody (1:200). Analysis of surface marker expression of leukemia cells was performed after gating on GFP-positive or CD45.2/iRFP670-positive cells. Cellular effects after NUP98::KDM5A degradation were assessed after treatment with 35 μM dTAG13 for indicated durations. Myeloid differentiation was monitored by measurement of surface marker expression as described above using the following antibodies (1:200) (all from Biolegend, USA): anti-mouse CD11b/Mac-1 PerCP-Cy5.5 (clone M1/70), anti-mouse Gr-1/Ly-6C PE-Cy7 (clone RB6-8C5) and anti-mouse CD117/c-Kit APC (clone 2B8). Apoptosis analysis was performed using the Annexin V AF647 Conjugate (#A23204, Invitrogen, USA) according to the manufacturer's

instructions. For cell cycle analysis, cells were fixed in 70% Ethanol for 30 min before staining with propidium iodide solution for 30 min. Samples were analyzed on a BD FACSCanto II or on an Intellicyt IQue Screener (Sartorius, Germany). Data analysis was done using the FlowJo or ForeCyte (Essen Bioscience, USA) software packages.

### Cytospins

Cells were cytocentrifuged onto glass slides and stained with Rapid-Chrome Kwik-Diff Staining Kit (Thermo Fisher Scientific, USA) followed by microscopic analysis. Photographs were taken on a Zeiss Imager Z.1 microscope and images were processed using Zeiss ZEN software (Zeiss, Germany).

### Competitive cell proliferation assay with shRNA knockdown and shCdk12 rescue experiment

To investigate the effect of the gene knockdown, competition-based proliferation assays were performed using rtTA3-expressing AML cells driven by *NUP98::KDM5A* (with or without stable expression of exogenous CDK12 wild type or CDK12^D873N), *KMT2A::MLLT3* and N-terminal *CEBPA* mutations *(Cebpa^p30/p30)* that were transfected with doxycycline-inducible shRNAs at an infection rate of approximately 50%[10]. $5 \times 10^5$ cells were seeded in 24-well plates in triplicates. Doxycycline (Merck, Germany) (1 μg/ml) was added to the media to induce knockdown and fluorescence expression of iRFP670 (coupled to shRNAs) was measured every 2 and 3 days using the IntelliCyt iQue Screener Plus (Sartorius, Germany). To compare the effects of candidate gene knockdown on cell growth we calculated the mean area under the curve (AUC) of the iRFP670-positive population, which accounts for the dynamics of the shRNA-induced growth inhibitory effect over time. Data analysis was done using the ForeCyte (Essen Bioscience, USA) software packages and values were normalized to day 3 or 2 after doxycycline induction and the shRenilla negative control.

### Proliferation assay for dTAG13 rescue experiment

dTAG-NUP98::KDM5A cells (with or without stable expression of exogenous CDK12 wild type or CDK12^D873N) were seeded in triplicates and treated with 35 nM dTAG13 and maintained on 3,5 nM dTAG13 after day 2. Cells were split and counted at regular intervals. Growth rates and cumulative cell numbers were calculated using Microsoft Excel. Growth rate was determined according to Eq. (1).

$$\frac{cell\ number(t1)}{cell\ number(t0)} * splitting\ factor \tag{1}$$

### Cell viability assay

Cells were seeded in white 96-well plates in triplicates and treated with BSJ-4-116 or THZ531 (both MedChemExpress, USA) in a dilution series at indicated concentrations, followed by incubation for 3 days. Cell viability was determined using the CellTiter-Glo® Luminescent Cell Viability Assay (Promega, USA), on a Spark multimode microplate reader (TECAN, Switzerland). Dose–response curves were calculated using the Prism software v6.0.1 (GraphPad, USA).

### Immunofluorescence staining and microscopy

dTAG-GFP-NUP98::KDM5A cells were treated with dTAG13, THZ531, BSJ-4-116 or DMSO with indicated concentrations samples were taken after 4 h and 72 h. Cells were cytocentrifuged onto glass slides and spots were then fixed with 4% formaldehyde (Histofix, P0087.6, Carl Roth, Germany) for 10 min and permeabilized with 0.2% Triton X-100 in PBS for 15 min at room temperature. For detection of DNA damage, samples were incubated overnight with a polyclonal anti-gamma H2A.X antibody (ab2893, Abcam, UK) (1:1500) followed by 1 h of incubation with a secondary Alexa Fluor-647-conjugated antibody (#A-21246, Invitrogen, USA) (1:333). Slides were counterstained with DAPI and

mounted using a water-based fluorescence mounting medium (Agilent, S302380-2). Images were acquired using an inverted confocal microscope with Airyscan super-resolution capability Zeiss LSM 880 Airyscan (Zeiss, Germany) and a 63× oil objective (Zeiss 63×/1.40 Plan-Apochr., Oil, DIC III) in Airyscan mode and Zen Black software. Laser intensities and microscope detectors were carefully set to avoid pixel saturation. The same settings were used to acquire all the samples. Raw.czi files were imported into Arivis Vision 4D (Arivis AG, Germany) and an automated segmentation pipeline was designed manually. The segmented objects were manually proofread and adjusted and the mean pixel intensity was calculated for every individual nucleus.

## Western blotting
Western blotting was performed according to standard protocols. Briefly, dTAG-NUP98::KDM5A cells were treated with 35 nM dTAG13 or DMSO for indicated durations, harvested by centrifugation, washed with PBS, snap frozen in liquid nitrogen. Whole-cell lysates were prepared by incubating samples at 95 °C for 10 min in Laemmli buffer, followed by 30 min of sonication. Proteins were separated on a 15% sodium dodecyl sulfate polyacrylamide gel and transferred to nitrocellulose membranes. The following primary antibodies were used for immunoblotting: Histone marks: anti-H3 (ab1791) (1:1000), anti-H3K4me3 (ab8580) (1:1000) and anti-H3K27ac (ab4729) (Abcam, UK) (1:1000), anti-HSC70 (sc-7298, Santa Cruz, USA) (1:10000) and anti-NRas (sc-31, Santa Cruz, USA) (1:1000). The membrane was stripped for 20 min in between antibody incubations.

## Real-time quantitative PCR
dTAG-NUP98::KDM5A cells were treated with THZ531 or DMSO for 24 h, harvested by centrifugation, washed with PBS and snap frozen in liquid N2. Total RNA was extracted using the Monarch Total RNA Miniprep Kit (NEB, USA). Reverse transcription was done using RevertAid RT Reverse Transcription Kit (Thermo Fisher Scientific, USA) and quantitative PCR was performed using the SsoAdvanced Universal SYBR Green Supermix (Bio-Rad, USA) on a Bio-Rad CFX-Connect Real-Time PCR Detection System. Results were normalized to *Gapdh* and analyzed using the $2^{-ddC(t)}$ method. Primers are listed in Supplementary Data 7.

## Chromatin immunoprecipitation (ChIP)-qPCR
ChIP was performed as previously described[86]. In brief, dTAG-NUP98::KDM5A cells were treated with 35 nM dTAG13 (Sigma-Aldrich, USA) or DMSO for indicated durations. Cells were harvested and washed with PBS, followed by crosslinking with 11% formaldehyde (Thermo Fisher Scientific, USA). The reaction was quenched with 2.5 M glycine (Sigma-Aldrich, USA), and cells were lysed in 1% SDS-buffer. After sonication, cells were incubated overnight with 1 µg/µl anti-H3K4me3 (ab8580, Abcam, UK) or anti-H3K27ac (ab4729, Abcam, UK) antibodies. Antibody-bound DNA was isolated using G-protein coupled magnetic beads (Dynabeads Protein G, Invitrogen, USA) and decrosslinked. Enrichment of selected genomic regions was tested by qRT-PCR and compared to a negative region control. SsoAdvanced Universal SYBR Green Supermix (Bio-Rad, USA) was used for qRT-PCR, which was performed on a Bio-Rad CFX96 TouchTM Real-Time PCR Detection System (Bio-Rad, USA). Data of technical triplicates were analyzed as percent of input. qRT-PCR primer sequences can be found in Supplementary Data 7.

## Cleavage under targets and tagmentation (CUT&Tag)
For measurement of H3K27ac and H3K4me histone mark distribution the iDeal CUT&Tag kit for histones compatible with histones and some non-histone proteins (Cat# C01070020, Diagenode, Belgium) was used according to the manufacturer's protocol. dTAG-*NUP98::KDM5A* cells were treated with 35 nM dTAG13 (Merck, Germany) or DMSO for 8 h and harvested for processing with the iDeal CUT&Tag kit. Three individual cryopreserved BM samples from a *NUP98::KDM5A* PDX model were thawed and stained for human CD45 (all approx. 97%

hCD45+) and immediately processed with the iDeal CUT&Tag kit. After wash with complete CT Wash Buffer, ConA beads were added and samples were incubated on a rotator. Cells were permeabilized with Complete CT Antibody Buffer and samples were incubated with 1 µg of either anti-H3K4me3 (ab8580), anti-H3K27ac (ab4729) (Abcam, UK) or rabbit IgG (Antibody Package for CUT&Tag anti-rabbit, Diagenode, Belgium) antibodies overnight at 4 °C, while rotating. After removing supernatant, anti-rabbit secondary antibody (1:100) (Antibody Package for CUT&Tag anti-rabbit, Diagenode, Belgium) was added and samples were incubated for 45 min at RT, followed by washing with complete CT Wash Buffer 2. pA-Tn5 transposase diluted in complete CT Wash Buffer 3 (1:250) was added and samples were incubated for 1 h at RT. After washing with complete CT Wash Buffer 3, CT Tagmentation Buffer was added for activation of tagmentation and samples were incubated at 37 °C for 1 h at 800 rpm. To stop tagmentation, CT Buffer E, CT Buffer S, and proteinase K were added and samples were incubated at 55 °C for 1 h at 800 rpm. Column-based DNA purification was performed by the addition of ChIP DNA Binding Buffer. For library amplification, purified DNA was mixed with primer index pairs (24 UDI for Tagmented libraries Set I & II, Diagenode, Belgium) and 2× High Fidelity Mastermix using the manufacturer's PCR protocol. Libraries were purified using AMPure XP beads (Beckman Coulter, Germany) and analyzed with an Agilent TapeStation instrument (Agilent Technologies, USA) to assess the library quality, size, and concentration. Purified libraries were sequenced on an Illumina NextSeq2000 instrument (Illumina, USA) using paired-end 50 bp chemistry.

## RNA sequencing
For RNA-seq analysis, dTAG-NUP98::KDM5A cells were treated with 35 nM dTAG13 (Merck, Germany) or 100 nM BSJ-4-116 (MedChemExpress, USA) for indicated durations. RNA was isolated according to the standard protocol using the NEB Monarch Total RNA Miniprep Kit (NEB, USA). Sequencing libraries were generated using the QuantSeq 3' mRNA-Seq Library Prep Kit (FWD) for Illumina (Lexogen, Austria) according to manufacturer's protocol. Purified libraries were sequenced on an Illumina NextSeq2000 instrument (Illumina, USA) using single-read 100 bp chemistry. SLAM-seq was performed as previously described[28]. dTAG-NUP98::KDM5A cells were treated with 35 nM of dTAG13 (Merck, Germany) for 1 h, followed by the addition of 100 µM 4-thiouridine (4sU, Merck, Germany) for 1 h. RNA was isolated according to the standard protocol using the RNeasy Plus Mini Kit (Qiagen, Germany). Total RNA was alkylated using 10 mM iodoacetamide (Merck, Germany) and RNA was purified by ethanol precipitation. Sequencing libraries were generated using the QuantSeq 3' mRNA-Seq Library Prep Kit (FWD) for Illumina (Lexogen, Austria) according to manufacturer's protocol. Purified libraries were sequenced on an Illumina NovaSeq instrument (Illumina, USA) using single-read 100 bp chemistry.

## ATAC-seq
Fresh-frozen samples of primary bone-marrow mononuclear cells were rapidly thawed, resuspended in Iscove's Modified Dulbecco's Medium (IMDM, Gibco, USA) and washed with PBS (Gibco, USA). Blast populations were identified with a routine diagnostics panel of monoclonal antibodies and sorted via flow cytometry (FACSCalibur, BD Biosciences, USA). dTAG-NUP98::KDM5A cells were harvested by centrifugation and washed with PBS. Subsequent DNA-isolation and library preparation for ATAC-seq was performed on 50,000 sorted blasts using Tagment DNA TDE1 Enzyme and Buffer Kits (Illumina, USA) as previously described[87].

## Genome-wide CRISPR/Cas9 screen
The design and construction of the murine genome-wide Vienna sgRNA library, as well as the experimental workflow for loss-of-function screening have been previously described[80,88]. Briefly, a single-cell-derived Cas9-expressing clone of a *NUP98::KDM5A*-driven

AML cell line was infected with a lentivirally-packaged sgRNA library at a low multiplicity of infection (MOI) (≤10%). Infected cells were selected using G418 (1 mg/ml, InvivoGen, France), which was kept in the medium until the end of the screen. Screen endpoint samples were harvested after 14 cell population doublings, and cell pellets corresponding to at least 500-fold library representation were harvested. Next generation sequencing libraries of screen endpoint samples and sgRNA plasmid pools were prepared and sequenced on a HiSeq 2500 instrument (Illumina, USA) using single-read 50 bp chemistry.

### Bioinformatics analysis

**Genome-wide CRISPR screen.** Raw.bam files underwent processing using the crispr-process-nf Nextflow[89] pipeline, available at https://github.com/ZuberLab/crispr-process-nf, which included barcode trimming, filtering, and alignment using Bowtie2[90]. The crispr-mageck-nf Nextflow pipeline, available at https://github.com/ZuberLab/crispr-mageck-nf incorporating MAGeCK[91], was employed for comprehensive analysis of the CRISPR screen data.

**RNA-seq (QUANT-seq).** Raw files underwent preprocessing using prinseq-lite[92] (version 0.20.4). Subsequently, alignment against the mouse reference genome (mm10) was performed using STAR[93] (version 2.7.9a). Post-processing and sorting of aligned reads was conducted using samtools[94] (version 1.4). Counts per gene were obtained utilizing featureCounts[95] (version 2.0.3) from the subread package. Normalized expression levels and identification of differentially expressed genes were calculated using the DESeq2[96] R package. Heatmaps depicting expression patterns were created using the heatmap.2 function from the gplots[97] R package. Time-series plots illustrating dynamic expression changes over time were generated using the maSigPro[98] R library. Gene set enrichment analysis was performed with GSEA software[99] (version 4.3.2). GO-enrichment was performed with EnrichR[100] and g:Profiler[101] (g:GOst) and plots were generated with SR-plot[102].

**Nascent RNA-seq (SLAM-seq).** Raw.bam files underwent processing using the slamseq Nextflow pipeline, accessible at https://github.com/nf-core/slamseq. This pipeline integrates multiple tools including fastqc, trim_galore[103] and slamdunk[104] available at: https://github.com/t-neumann/slamdunk, for quality assessment, adapter trimming, and read mapping. DESeq2 was employed for comparative analysis, facilitating the identification of newly transcribed genes. Heatmaps depicting expression patterns were created using the heatmap.2 function from the gplots R package.

**CUT&Tag.** Raw files underwent preprocessing using prinseq-lite for quality control and filtering. Alignment against either the mouse (mm10) or human (hg38) reference genomes was conducted using BWA[105] (version 0.7.17-r1188). Post-processing and sorting of aligned reads were performed using samtools (version 1.13). The function bamCoverage from Deeptools[106] (version 3.5.1) was utilized to generate BigWig files with a binSize of 10, employing Counts Per Million for normalization. Peaks were called using macs2[107] (version 2.1.0) with the –broad option. Differential CUT&Tag regions were identified using the R package DiffBind[108], applying DESeq2. Tornado and profile plots were created using the generateEnrichedHeatmap from the profileplyr[109] R package or Deeptools. Read counts per genes were obtained using featureCounts from the subread package. All hockey-stick and scatter plots were generated using ggplot2[110] in R. Coverage plots were generated with IGV[111].

**ATAC-seq.** Publicly available AML patient ATAC-seq datasets were obtained from GEO accessions listed in Supplementary Data 2. Pediatric AML patient data were obtained from the St. Anna Children's Cancer Research Institute (CCRI) and Yokohama City University Hospital[19].

A modified version of the ATAC-seq Data Processing Pipeline[112,113] was applied to the raw Bam files, accessible at: https://github.com/epigen/atacseq_pipeline. The pipeline utilized fastp[114] for adapter removal and Bowtie2 for read alignment to the GRCh38 (hg38) human reference genome. Duplicate marking was performed with samblaster[115] and aligned BAM files were sorted, indexed, and filtered for ENCODE blacklisted regions using samtools. Peaks were called with macs2 (version 2.1.0) applying the –broad option. Differential ATAC-seq regions were identified with DiffBind and region-to-gene annotation was conducted with ChIPseeker[116]. Counts over exons were obtained using featureCounts and normalization was performed with DESeq2. Principal Component Analysis (PCA) plots were generated using the ggplot2[110] package in R. For the minimum spanning tree visualization batch correction was done with ComBat[117] and we utilized vegdist from the VEGAN[118,119] package and ggraph[120]. Tornado and profile plots were created with Deeptools and the differential ATAC-seq regions showcased via heatmaps were generated with DiffBind.

**Motif enrichment analysis.** Motif enrichment analysis was conducted using the findMotifsGenome.pl function from HOMER[121] (version 4.1). This analysis was performed on 1921 regions with increased accessibility and 6054 regions with decreased accessibility in the hg38 genome, to identify enriched motifs.

**Analysis of AML patient gene expression data.** Gene expression data from AML patients were retrieved using the St. Jude Cloud PeCan tool (https://pecan.stjude.cloud/), which provides access to the St. Jude Cloud data repository[23]. Raw count data were normalized using DESeq2, followed by further data processing in R. Visualizations were generated using the ggplot2[110] package.

### Statistical analysis

For Cleavage Under Targets and Tagmentation (CUT&Tag), RNA sequencing (RNA-seq), genome-wide CRISPR screen, Gene Ontology (GO) term analyses, assay for transposase-accessible chromatin using sequencing (ATAC-seq), chromatin immunoprecipitation DNA sequencing (ChIP-seq) and gene set enrichment analyses published statistical packages were used as referenced in the respective methods sections. Otherwise, the Prism software (version 6.0.1, GraphPad, USA) was used for statistical analyses, and data are shown as mean ± SD. The unpaired Student's $t$ test or the Mann–Whitney test were used for $P$ value determination. Results were considered significant when $P < 0.05$. $P$ values are indicated in each figure. The individual sample size is reported in figure legends.

Schematic figures were generated with Adobe Illustrator CS6 (Adobe, USA) or BioRender.com.

### Reporting summary

Further information on research design is available in the Nature Portfolio Reporting Summary linked to this article.

## Data availability

The publicly available *Tet-Off* RNA-seq data used in this study are available in the Gene Expression Omnibus (GEO) database under accession code GSE134784[10]. The raw ATAC-seq data from primary pediatric AML samples obtained from the Yokohama City University (YCU)[19] are not available due to data privacy laws. The processed ATAC-seq count data are available on Zenodo under https://doi.org/10.5281/zenodo.14943880. The publicly available ATAC-seq data from 15 primary pediatric AML samples obtained from the St. Anna Children's Cancer Research Institute (CCRI) used in this study are available in the GEO database under accession code GSE282258. The mapping of sample names to GEO accessions is provided in Supplementary Data 1. The genome-wide CRISPR-screen, RNA (QUANT)-seq, nascent (SLAM)-seq, Cut&Tag and ATAC-seq cell line data generated in this study have been

deposited in the GEO database under accession code GSE255808. The remaining data are available within the Article, Supplementary Information or Source Data file. Source data are provided with this paper.

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

## Acknowledgements

All members of the Grebien group took part in scientific discussions, Georg Winter advised on the establishment of the dTAG model, Michaela Fellner provided experimental support with the genome-wide CRISPR screen, Raúl Jiménez Heredia and Anna Hurt provided experimental support with patient material and ATAC-seq library preparations and Gabriele Stefanzl gave scientific advice on healthy donor primary cells. Elizabeth Heyes assisted in the finalization of the manuscript. The team of the Vienna BioCenter Core Facilities (https://www.vbcf.ac.at) performed next generation sequencing services. Norio Shiba (Yokohama City University; Japan) provided ATAC-seq raw data and additional information on pediatric patient samples[19]. This work was supported by a starting grant from the European Research Council to F.G. (636855) (F.G.), the Austrian Science Fund (Grants TAI490, P35298 (F.G.) and KLI 1056-B (K.B.)) and a Heidi Ras Grant of the FZK University Children's Hospital Zurich (N.S.). S.T., J.S., and M.A. are recipients of a Doctoral Fellowship (DOC) of the Austrian Academy of Sciences at the Institute for Medical Biochemistry at the University of Veterinary Medicine Vienna. J.S. is supported by the Fellinger Cancer Research Fund.

## Author contributions

F.G. and S.T. conceived the work and designed the study; T.E. performed the bioinformatics analyses; S.T., N.W., M.P., P.F.P., J.S., B.H., G.M., M.A., and M.M.G. performed experiments; B.H., K.N., M.N.D., and K.B. provided ATAC-seq data and scientific input on the patient data; N.S. and B.B. established the PDX model; R.M. provided patient material; P.V. provided healthy donor primary material; S.T., F.G., and T.E. wrote the manuscript; J.Z., K.B., and G.S.F. were involved in experimental design and scientific discussions; F.G. supervised the study and all authors revised and approved the manuscript.

## Competing interests

J.Z. is a founder, shareholder, and scientific advisor of Quantro Therapeutics GmbH. J.Z. and the Zuber laboratory receive research support and funding from Boehringer Ingelheim. P.V. received consultancy honoraria from Pfizer, AOP Orphan, Delbert, Novartis, and Blueprint. The other authors declare no conflict of interest.
