## [Transparent Peer Review file · Nature Communications]

Transcriptional and epigenetic rewiring by the NUP98::KDM5A fusion oncoprotein directly activates CDK12

Corresponding Author: Professor Florian Grebien

Version 0:

Reviewer comments:

Reviewer #1

(Remarks to the Author)

Transcriptional and epigenetic rewiring by the NUP98::KDM5A fusion oncoprotein directly activates CDK12-controlled DNA repair

The study by Troester et al. interrogated epigenetic profiles from pediatric AML patients and healthy controls to identify the molecular consequences of NUP98::KDM5A-mediated transformation. Thereby, the authors derived a GMP-like state of NUP98::KDM5A patient samples with additional stem cell like features. For functional interrogation, the authors developed a fetal cell-derived NUP98::KDM5A/NRas-G12D model with the ability to rapidly deplete NUP98::KDM5A with the dTag13 system. Depletion of NUP98::KDM5A in this model caused rapid differentiation of leukemic cells followed by cell death. Epigenetic profiling upon NUP98::KDM5A depletion highlighted loss of H3K27ac and H3K4me3 at known (Hox cluster) and unknown potential targets of the fusion oncogene. To narrow down direct targets of NUP98::KDM5A more precisely, the authors performed SLAM-Seq highlighting 45 putative direct targets. To derive targetable vulnerabilities, the authors next performed a genome-wide CRISPR-Cas9 screening with 12 genes of the 45 direct targets (SLAM-Seq) being essential for leukemic cells, including known NUP98-rearranged leukemia drivers like the HoxA cluster genes. Individual validation of the CRISPR screening highlighted CDK12 as major susceptibility of the NUP98::KDM5A leukemia cells. Following up on CDK12, the authors analyzed a public patient dataset and found significant overexpression of CDK12 in NUP98-rearranged and NUP98::KDM5A patients. Utilizing shRNA and a CDK12 degrader, sensitivity of NUP98::KDM5A cells (murine and PDX) is highlighted and mechanistically explained by the DNA-repair-associated function of CDK12. The study thereby highlights how NUP98::KDM5A controls epigenetic regulation of gene expression to enable sufficient DNA repair essential for its survival, and how this dependency may be exploited therapeutically to treat a severe form of pediatric AML. Given the authors can provide additional pre-clinical data on CDK12-inhibition, the study will add significantly to the understanding and the prospective treatment of NUP98::KDM5A-rearranged pediatric AML.

Major points

1. Relating to Figure 1: Could the authors please comment on the populations used for ATAC-seq. In the Material and Methods section "Blast populations" are mentioned without further specification. Could the authors please also elaborate on the "mature" cells used for comparison? Taking into account that NUP98::KDM5A is strongly enriched in AMKL, the finding of close proximity to GMPs instead of MEPs is interesting but may need further analysis.
2. ATAC-seq analysis of NUP98::KDM5A samples (vs. rest) deviates in one patient. Can the authors derive any information explaining this deviating chromatin accessibility (e.g. blast population immunophenotype)?
3. Did the authors perform ATAC-seq of their murine model and compare it to the patient samples?
4. Did regions of reduced H3K27ac and H3K4me3 upon degradation of NUP98::KDM5A (murine model) overlap with regions of open chromatin in patients, or with H3K27Ac and H3K4me3 in the analyzed PDX model?
5. The authors noted loss of H3K27ac and loss of CDK12 expression at 8h post NUP98::KDM5A degradation. What is the ATAC-seq and histone status of CDK12 in patient samples?
6. SLAM-seq resulted in detection of 45 deregulated genes 1h-2h post NUP98::KDM5A degradation. It would be helpful to highlight CDK12 (as main gene of interest for follow-up studies) in Figure 4B.
7. On page 9, lines 250-251, the authors note that other genes, such as CDK12, are unexplored or implicated in other cancers. The authors may include citation 68 and the work by Hoshii and colleagues (DOI 10.1016/j.cell.2018.01.032).

8. The authors performed a genome-wide CRISPR screen and highlighted the overlap with their SLAM-seq. Could the authors please comment on the 87 genes found deregulated 8h post NUP98::KDM5A degradation?
9. Also relating to this, did the authors see changes of DNA repair genes in their gene expression profiling upon NUP98::KDM5A (at later time points)? This refers particularly to the genes highlighted in Figure 6F. The authors only refer to a previously published dataset.
10. The authors highlight the sensitivity of NUP98::KDM5A cells to CDK12 KD or degradation. Can the authors provide any other NUP98-rearranged and non NUP98-rearranged data to further underline the high sensitivity of NUP98::KDM5A leukemia?
11. Can the authors (at least partially) rescue the NUP98::KDM5A degradation via CDK12 overexpression?
12. Can the authors provide any in vivo data for efficacy of CDK12 inhibition of NUP98::KDM5A cells (e.g. their murine model; if more suitable for in vivo use, a CDK12/13-inhibitor will still underscore the results).
13. Fan and colleagues (citation 19) noted functional redundancy of CDK12 and CDK13. Interestingly, the authors note a selective dependency on CDK12 in NUP98::KDM5A leukemia. This should be pointed out and CDK13 expression and depletion in the CRISPR-screening may need additional attention.

Minor points

1. The authors note more subtle depletion of HoxA genes with shRNA knockdown compared to CDK12. Was this also true for the CRISPR-based screening? The authors hypothesize on redundant function, which is reasonable and should be true for the CRISPR-based approach, too. Otherwise less reduction via shRNA may cause the observed effect.
2. The authors may tame down the statement in lines 457-459. Indeed, functional and mechanistic studies on CDK12 in myeloid malignancies have been sparse and this work adds significantly to the knowledge in the field.

Reviewer #2

(Remarks to the Author)

In this interesting, well-conducted and clearly written study, Troester et al use a combination of epigenomic, transcriptomic, and functional assays to identify CDK12 as a dependency in NUP98Ar leukemias. The authors first identify differentially-accessible peaks in NUP98Ar AMLs compared to other AMLs and healthy hematopoietic cells using ATAC-seq. As NUP98 fusions are difficult to study and relevant models are scarce, they develop a murine HSPC model with a dTAG fusion that allows for investigation of the impact of acute fusion loss on the transcriptome and epigenome of NUP98::KDM5A fusion-driven cells. They then use this model to perform a CRISPR dependency screen, and by integrating with SLAM-seq and CUT&Tag data identify CDK12 as a dependency in NUP98Ar AML. They further pin the dependency on the role of CDK12, a kinase that phosphorylates RNA Pol II, on its regulation of DNA damage response genes. Targeting CDK12 with small molecule degraders leads to cell death in multiple NUP98Ar. It would be nice to see a survival advantage in a NUP98Ar PDX treated with CDK12 inhibitor, but given difficulty with modeling this disease the current results sufficiently show a role and potential for this approach in NUP98Ar patients. NUP98Ar AML is difficult to study and mechanistic insights like those reported here represent an important advance with relevance for pediatric leukemia researchers.

Major Points

It seems like the authors performed ATAC-seq on patient samples and report generally descriptive but not particularly informative results, and then independently developed their model to understand how fusion binding drives a novel dependency. As reported, these approaches seem distinct and they should try better integrate the ATAC data with the rest of the manuscript, or even consider leaving it out as it distracts from the rest of the paper (e.g. they do not thoroughly explore CDK12 or other direct targets in this dataset, and even turn to previously-published RNA-seq for important comparisons). Many of the comments below would be addressed satisfactorily by removing the ATAC dataset.

1. Figure 1A: A major focus of the manuscript is on their ATAC-seq dataset. The authors perform bulk ATAC-seq on NUP98::NSD1 and NUP98::KDM5A positive AMLs and integrate with publicly available AML and HSPC datasets. While the AMLs generally cluster together and are mostly distinct from normal cells, the distributions of the NUP98 fusions within all AMLs do not show any specific pattern. Previous studies showed that NUP98::KDM5A fusions are found across all FAB types with enrichment in M7. Can the authors re-annotate 1A using FAB classification and/or fusion type (e.g. KMT2Ar, CBF::GLIS2) to determine whether this explains the variance? An alternative is that data quality explains this variance, I did not see mention or report of QC metrics such as FrIP scores, can they be added to Supp Table 1?
2. Fig 1C-D show 1921 regions that are more accessible in NUP98r versus GMP cells. Are there any regions that are less accessible in NUP98r than in GMP? Since epigenetic silencing is a major driver of cellular differentiation, analyzing both more and less accessible regions could show cell of origin more convincingly. In addition, motif analysis of the 1921 peaks should be included to identify TFs that may mediate differential accessibility.
3. Fig 1D-E: The data are presented in a manner suggesting that the NUP98 fusion is directly responsible for the majority of these changes. The authors do not report ChIP-seq or CUT&RUN results for the NUP98 fusion itself. Please integrate NUP98 ChIP-seq from ref 9 onto the differentially-accessible peaks, and in particular the 1921 enriched compared to GMPs, to assess which are direct fusion targets.
4. Fig 1 D-E: ATAC-seq is an important but indirect surrogate for gene expression. The authors should integrate their differential ATAC peaks with RNA-seq from NUP98 patients that the authors used in Figure 6A (or PMID: 36815378) to assess which regions are associated with differential gene expression.

5. Lines 144-146 are not fully substantiated as the authors report examining histone modifications associated with gene activation only. To further corroborate this statement, the authors should at least report how many peaks were present in GMP that were not identified in NUP98r samples. Few inaccessible peaks in GMPs would suggest a that NUP98 fusions drive a more accessible landscape compared to healthy cells.

6. Regarding the degnon system, the authors cite reports that LLPS is important for NUP98 fusion activity. However, the quick and substantial loss of the fusion after dTAG13 molecule treatment is inconsistent with profound stability often afforded by LLPS. The authors should perform IF for the fusion in their dTag system to determine whether puncta consistent with LLPS formation is observed. If such puncta are not seen, it would suggest that LLPS-independent fusion activity is important for transformation, and if present, enhances the reliability of their model compared to prevailing models for NUP98r fusion activity.

7. Figure 3A, C: Given F1's dependence on ATAC-seq and that more ATAC peaks are present than K27ac or K4me3 C&T peaks, the authors should perform ATAC in their degnon system and directly compare to differentially-enriched regions reported in F1.

8. Figure 3E: The authors should plot RNA-seq signal (153 downregulated and 96 upregulated) from 8h and 24h timepoints from the degnon experiment onto the differentially-enriched regions from F1 to assess the relevance of these findings to patient samples.

9. Figure 3F/G: There is a missed opportunity to integrate the degnon and patient ATAC-seq datasets here as well. Please plot ATAC signal from all samples in 1E onto the 38 NUP98::KDM5A "target genes" to assess relevance and specificity to NUP98Ar patient sample epigenomes.

10. Figure 6D: Please treat other AML cell lines with predicted relative insensitivity or dependency on CDK12 with BSH-4-116. The authors should consider adding KASUMI-1, K562 (both predicted to be insensitive based on DepMap) and AML193 and OCI-AML3 (predicted to be sensitive). If KASUMI-1 or K562 show relative insensitivities (>100nM), the authors should perform RNA-seq from those cells treated with BSJ and compare to their NUP98Ar results to assess whether their reported DDR defect sensitivity is specific to NUP98Ar leukemias, as they suggest.

Minor Points:

1. Figure 2: The authors should report Nras levels (Western blotting is sufficient) in the degnon system to assess whether they change in the context of NUP98 fusion loss. If not impacted, this would further demonstrate the differentiation phenotype is solely dependent on the fusion. This is important because while NRAS mutations can co-occur in NUP98r, some reports suggest they are not present in the majority of cases

2. Figure 5A: Were any of the 33 "direct" NUP98::KDM5A target genes found in the 645 up list? Presumably no but please state in the text. Would be interesting if the fusion drives expression of a potential tumor suppressor that is mitigated by expression of other targets.

3. Line 628 typo "IRFP670-postive"

Reviewer #3

(Remarks to the Author)

Chromosomal translocations involving the NUP98 locus are among the most prevalent rearrangements observed in pediatric acute myeloid leukemia (AML). AML expressing NUP98 fusion proteins are characterized by a distinct transcriptional signature, marked by high expression of HOXA and MEIS1 genes, and are associated with unfavorable clinical outcomes. While NUP98 fusion proteins are established drivers of AML development, the precise molecular mechanisms underpinning NUP98 fusion protein-dependent leukemogenesis remain elusive, thereby impeding the development of targeted therapies aimed at improving the clinical outcomes for patients with NUP98 fusion-positive AML.

This study therefore investigated the epigenetic and transcriptional signatures controlled by NUP98 fusion proteins in AML, with a primary focus on the specific NUP98-KDM5A fusion oncoprotein, which is observed in ~2% of all pediatric AML patients. Using a series of genome-wide approaches, Troester et al. identified a unique epigenetic signature in NUP98 fusion-driven AML samples, marked by overlapping activating histone marks (e.g., H3K27ac, H3K4me3). To further delineate the molecular impact of NUP98-KDM5A fusion in AML, the authors developed an elegant murine AML model where a degradation tag (dTAG) system was fused to NUP98-KDM5A and expressed in NrasG12D murine fetal liver-derived hematopoietic stem cell progenitors, before being transplanted in irradiated recipient mice. Degradation of the dTAG-NUP98-KDM5A fusion oncoprotein led to cell cycle arrest, differentiation and apoptosis of leukemic cells, highlighting the dependency on NUP98-KDM5A in this murine model. Importantly, NUP98-KDM5A degradation correlated with a significant and localized reduction in H3K27ac and, to a lesser extent, H3K4me3 levels. Differential expression analysis by RNA-seq identified 153 downregulated genes upon NUP98-KDM5A degradation, while SLAM-seq analysis revealed 45 target genes that exhibited significant loss in H3K27ac mark following NUP98-KDM5A degradation and high binding occupancy of NUP98-KDM5A in the promoter of these genes. To gain further insight into the genetic dependencies of murine NUP98-KDM5A AML cells, the authors performed a genome-wide CRISPR-based loss-of-function screen and identified 4105 genes that caused a fitness defect in NUP98-KDM5A driven AML cells. Interestingly, 12 of the 45 target genes previously identified were essential for NUP98-KDM5A driven AML cells and validated in a shRNA-based competition growth assay in vitro. The authors focused on the cyclin-dependent kinase CDK12 and its role in NUP98-KDM5A driven AML cells. Depletion of Cdk12 by shRNA impaired the proliferation of murine NUP98-KDM5A AML cells in

vitro, while pharmacological inhibition of CDK12 induced cell death in both murine and human NUP98-KDM5A AML models. Depletion of NUP98-KDM5A or pharmacological inhibition of Cdk12 resulted in a significant downregulation of genes linked to DNA repair and induced the formation of g-H2AX foci in AML cells, suggesting that CDK12-dependent control of DNA repair gene expression may promote the survival of NUP98-KDM5A AML cells in vitro.

Overall, the manuscript is extremely well written, and the authors employed elegant and cutting-edge approaches to map the molecular mechanisms controlled by NUP98-KDM5A fusion oncoprotein in AML cells. For the most parts, the conclusions drawn by the authors align with the data provided in this manuscript. At this stage, the study provides limited validation(s) in human NUP98-KDM5A AML cells and no functional confirmation in vivo. Furthermore, some key controls are lacking, which should be provided before the manuscript can be considered for publication:

1. Epigenomic characterization of pediatric NUP98-driven AML samples: PCA analysis revealed that pediatric NUP98-driven AML samples are broadly distributed compared normal healthy blood cells (Fig.1A), while inter sample distance visualization by minimum spanning tree placed NUP98-driven AML samples between granulocyte-monocyte progenitor cells and mature cells (Fig.1B). Can the authors elaborate on this “discrepancy”? Where would adult vs. pediatric AML samples localize in this trajectory analysis? More generally, I failed to understand why the authors would expect distinct ATAC-seq clustering of pediatric vs. adult AML samples based on differences in somatic mutations (line 118-120). One pediatric sample greatly differs from the 4 NUP98-KDM5A-driven AML sample analyzed (Fig.1E). Could the authors provide more details about the specificity of this patient (e.g., characteristics of the NUP98-KDM5A fusion, mutational landscape...)?
2. Characterization of the dTAG-NUP98-KDM5A murine AML model: The authors developed a unique murine AML model to define the molecular mechanisms underpinning NUP98-KDM5A fusion protein-dependent leukemogenesis. Still, limited comparison with NUP98-KDM5A-driven pediatric AML samples has been completed to ensure that their model is recapitulating the molecular and clinical features of NUP98-KDM5A-driven pediatric AML. For example, what is the overlap in term of differential gene expression between NUP98-KDM5A-driven pediatric AML samples and murine cells?
3. Identification of NUP98-KDM5A target genes: RNA-seq-based analysis identified a subset of 38 target genes that were significantly downregulated and displayed loss in H3K27ac mark upon fusion protein degradation, while exhibiting high binding occupancy of NUP98-KDM5A in the promoter of these genes (Fig.3). On the other hand, SLAM-seq analysis identified 45 target genes whose expression was significantly downregulated upon fusion protein degradation (Fig.4). What is the overlap between the RNA-seq-based 38 target genes and the SLAM-seq-based 45 target genes? More generally, some key controls are lacking in this set of experiments, including the re-expression of the fusion protein to ensure that these changes are specific to NUP98-KDM5A and a dose-response analysis of dTAG13 on the de novo synthesis of mRNA to distinguish direct vs. indirect control of NUP98-KDM5A. Furthermore, no differential expression analysis of these target genes in human AML samples is provided at this stage, which limits the interest/impact of these findings.
4. Identification of essential genes in NUP98-KDM5A AML cells: Fitness genes can be subdivided between core and context-dependent essential genes. To what extent the 4105 genes that caused a fitness defect in NUP98-KDM5A driven AML cells are core essential genes in any cell type or AML specific fitness genes? To what extent are these genes dysregulated in NUP98-KDM5A-driven AML vs. AML vs. HSC cells?
5. Characterization of CDK12 as an essential gene in NUP98-KDM5A AML cells: Rescue and separation-of-function(kinase-dead mutant) experiments should be considered in Cdk12-depleted murine AML cells and human AML cells to confirm the essential role of CDK12 in NUP98-KDM5A driven AML. RNA-seq analysis suggests that pharmacological inhibition of Cdk12 disrupts HR-mediated DNA repair in AML cells, in line with prior study (Johnson et al. Cell Reports 2016). Does CDK12 inhibition impairs the recruitment of RAD51 to g-H2AX foci in AML cells? Validation of these findings in vivo should be considered to strengthen these findings.

Reviewer #4

(Remarks to the Author)

This manuscript, by Troester and colleagues, investigates the role and mechanisms through which the NUP98::KDM5A fusion oncoprotein mediates its oncogenic functions in AML. Comparison of NUP98::KDM5A AML with GMPs identified promotor sites with increased accessibility and an enrichment in activation marks, particularly H3K27ac. This enrichment of activation marks and chromatin accessibility was decreased upon ligand induced protein degradation of NUP98::KDM5A which led the cells towards myeloid differentiation. 12 genes out of 45 direct NUP98::KDM5A target genes were shown to be essential for NUP98::KDM5A driven AML growth using an invitro model. Targeting of CDK12, one of the direct gene targets, reduced the survival advantage of NUP98::KDM5A AML cells, emphasizing CDK12 inhibition as a potential therapeutic option against NUP98::KDM5A fusion gene driven AML.

The manuscript is well-written and the experiments support the conclusions, for the most part. The major criticism of the manuscript are 1) many of the experiments seem to depend on results with a single NUP98::KDM5A murine cell line, and 2) there are no control experiments addressing the effect of the CDK12 inhibitor on WT hematopoietic cells, or on other tissues in vivo. Thus we don't know if targeting CDK12 is effective in vivo, or whether toxicity of CDK12 inhibition is excessive.

Specific comments:

1) HOXA genes, especially HOXA7-11, are well known target genes for NUP98 fusion proteins, and the authors see increased HOXA mRNA expression as one of their major findings. Same question for NKX2-3. Why do the authors think that their ATAC assay didn't identify HOXA genes in Supp table 2? Also, can the authors compare their data with published ATAC and/or NUP98 fusion CHIP target datasets?

- 2) Fig 1E, it would be useful to see WT GMP comparison here.
- 3) S1E – Can the authors show some representative FACS results ?
- 4) Fig 2 E, F indicate that the cells are arrested at G1-cell cycle phase at day 3 and are followed by apoptosis at day 7 and 10. In Fig 2G – they are showing the cells are differentiating. What time point do cells start to differentiate ? Also, why do the DMSO controls show less apoptosis at d. 10? Are the authors splitting or “feeding” the cells fresh media?
- 5) The authors focus on a single NUP98::KDM5A/Nras cell line, that seems to have evolved in vivo. Did the cell line have insertional activation of important cancer genes? Or spontaneous mutations of cancer genes? These could strongly effect the phenotype, including downstream gene activation. The overall manuscript would be improved if the authors could show key datasets were reproducible with 3 or more independent NUP98::KDM5A/Nras cell lines.
- 6) Can the authors show an intersection of target genes from their RNA-Seq, Chip Seq, and ATAC Seq datasets?
- 7) Fig S2 seems to be based on a previously published dataset. Can the authors explain this dataset? Is it mouse or human, etc? What are the “other genes” in figure S2I? How were “target genes” and “other genes” defined?
- 8) Line 222 – 228 – The authors have stated that NUP98::KDM5A actively maintains H3K27ac marks. How does the fusion protein maintain the histone mark, does it protect them from deacetylases ? maybe they can show if there is an effect on deacetylases ?
- 9) Fig 6H – The images are not very clear. Also, maybe they can add immunofluorescence images for CDK12 in the same panel.
- 10) The supplemental tables should be supplied as sortable Excel files; the pdfs are quite difficult to read.

Reviewer #5

(Remarks to the Author)

Reviewer #6

(Remarks to the Author)

In this paper, the authors wanted to better understand NUP98::KDM5A fusion protein driven AML. They started out by comparing ATAC-seq across multiple normal cells and patient samples. NUP98::KDM5A are closest to GMP's but also contain unique open chromatin regions. They performed Cut&Tag in a single PDX model and found these regions are generally enriched for H3K27ac and H3K4me3. A dTAG-NUP98::KDM5A construct was used to transform mouse fetal liver HSCs. By intracellular flow, dTAG treatment induced a complete loss of NUP98::KDM5A and led to cell cycle arrest, differentiation and apoptosis of the AML cells. By ChIP Q-PCR, there was a loss of H3K27ac levels at Meis1 and Hoxa9 promoters by 3 hours of dTAG treatment. H3K27ac Cut&Tag and RNAseq at 8 hours identified several regions of reduced H3K27ac that were also associated with reduced gene expression, although Meis1 did not display reduced expression until the 24 hours timepoint. Overall, the authors identified 38 genes that appear to be sensitive to NUP98::KDM5A regulation. They then performed SLAM-seq after 1 hour dTAG treatment to identify genes that displayed rapid loss of transcription upon NUP98::KDM5A degradation. Comparing this with NUP98::KDM5A ChIP-seq, they identified 45 gene targets that are direct targets of NUP98::KDM5A regulation. They performed a CRISPR screen in the NUP98::KDM5A cell line and found that 12 of the 45 direct targets are essential for cell survival. They further validated some of these targets and then focussed in CDK12. NUP98 fusion patient samples display higher expression levels of CDK12 than other AMLs. NUP98::KDM5A cells are sensitive to CDK12 KD and inhibitor treatment, and loss of CDK12 activity increased DNA damage.

Overall, the experiments are well performed and technically sound. I have two general concerns and several more specific concerns.

General concerns

1. The mechanistic data is somewhat descriptive, and it is not entirely clear what NUP98::KDM5A binding does to gene regulation. For instance, the authors show a strong correlation between H3K27ac levels (and H3K4me3) and changes in NUP98::KDM5A binding. From this, they propose that NUP98::KDM5A directly controls the epigenetic landscape at target genes. However, it is also possible that NUP98::KDM5A impacts transcription in some other way (e.g., it could demethylate a non-histone target), and changes in H3K27ac are indirect effects. In support of this, the changes in H3K27ac levels don't always correlate with changes in gene expression upon dTAG treatment. For example, according to Figure S2H, 67 genes show reduced expression with no impact on H3K27ac levels. Also, why are some genes sensitive to loss of NUP98::KDM5A binding while others are relatively non-responsive? Some additional work on the mechanism would be helpful.
2. The authors propose that CDK12 could be a new therapeutic target in NUP98::KDM5A AML. This is an interesting possibility but there are several issues with this idea. First of all, inhibiting kinases can have harsh toxic side effects. It would be useful to know how well normal cells respond to CDK12 KD or inhibition. Also, it would be useful to know if CDK12 dependency is specific to NUP98::KDM5A, or is a more general AML target. In addition, without in vivo work it is difficult to assess how good of a target this really is (see points 7-10 below).

Some specific concerns are outlined below:

1. A dTAG western blot would be useful, to validate the intracellular flow which can sometimes produce artifacts.
2. Why was the 8 hour timepoint chosen for the Cut&Tag and RNA-seq experiments when it looks like there is a significant H3K27ac reduction by 3 hours?
3. Why do some genes show changes in H3K27ac without displaying changes in transcription? Conversely, why do some genes show changes in transcription without showing changes in H3K27ac (or H3K4me3)? Does this data suggest that NUP98::KDM5A controls gene activation through mechanisms other than altering the epigenome?
4. Is it possible that NUP98::KDM5A is having global effects on either H3K4me3 or H3K27ac? Global effects can sometimes cause artifacts when normalizing genomic data for analysis. This can be easily checked using western blots for these and other marks in dTAG vs DMSO treated samples.
5. Where is Meis1 on Figure 3G?
6. Considering the SLAM-seq experiments, does loss of H3K27ac precede reduced transcription or occur afterwards? In other words, are the high H3K27ac levels at NUP98::KDM5A target genes a cause or consequence of transcription?
7. Based on previous NUP98::KDM5A ChIP-seq data, how many genes are bound by NUP98::KDM5A that don't change after degran treatment? In other words, how many genes bound by NUP98::KDM5A are insensitive to its regulation?
8. How specific is CDK12 dependency to NUP98 leukemias? Are other AMLs also sensitive to KD's?
9. How sensitive are normal hematopoietic cells to CDK12 KD or BSJ-4-116 treatment?
10. To claim that CDK12 is a possible therapeutic target, there needs to be some in vivo evidence for this, preferably using a PDX model if possible rather than the dTAG-NUP98::KDM5A model.

Version 1:

Reviewer comments:

Reviewer #2

(Remarks to the Author)

I appreciate the authors' significant and earnest efforts to address reviewer comments, particularly with regard to the ATAC-seq dataset. While many of the additional analyses showed modest, subtle or no detectable effect of the fusion on the transcriptome (e.g. S1D), this is likely due to difficulties integrating different data types rather than major issues with interpretations or biological systems. I do think the authors' initial presentation of the CDK12 dependency suggested specificity for NUP98 AMLs, and appreciate that this is now recognized as a general dependency across most AMLs. The NUP98 fusion may provide one route to a dependency on this kinase. The manuscript has ATAC-seq and CUT&Tag datasets that are likely to be useful for the field, and the murine HSPC degree model is also a useful resource. The bulk of the manuscript, the model, and the datasets as presented raise no concerns that they are likely to be invalidated by additional analyses or NUP98 models. I have no further recommendations and believe the manuscript is suitable for publication in Nature Communications.

Reviewer #3

(Remarks to the Author)

In this study, Troester et al. investigated the epigenetic and transcriptional signatures controlled by NUP98 fusion proteins in AML, with a primary focus on the specific NUP98-KDM5A fusion oncoprotein, which is observed in ~2% of all pediatric AML patients. Using a series of orthogonal approaches, the authors identified 12 direct NUP98-KDM5A target genes that are essential for AML cell growth, including the cyclin-dependent kinase 12 (CDK12). Targeting Cdk12 resulted in a significant down-regulation of genes linked to DNA repair and induced the formation of γ -H2AX foci in AML cells, suggesting that CDK12-dependent control of DNA repair gene expression may promote the survival of NUP98-KDM5A AML cells in vitro.

Overall, the authors have made a significant and careful effort to address the elements raised in my initial review, and I want to applaud them for that.

However, one remaining and central issue is the focus on DNA repair at end of the manuscript. While the presence of γ -H2AX foci suggests some involvement of CDK12 in DNA repair, the data presented are largely correlative. Without direct evidence of a DNA repair deficiency upon targeting CDK12, the strong focus on the role of CDK12 in DNA repair—as reflected in the title—may be somewhat misleading at this stage. I would recommend tuning down the emphasis on DNA repair and ensuring that the conclusions more accurately reflect the current data, which suggest a broader role for CDK12 beyond just DNA repair.

Reviewer #4

(Remarks to the Author)

The authors have added a large amount of new data, and addressed most of the questions that I raised.

Unfortunately, I do not think that the authors have adequately addressed the two major criticisms that were raised by this and other reviewers.

1) Many experiments were based on a single murine NUP98::KDM5A/Nras cell line, that selectively grew out after serial passage. What enabled this specific clone to outcompete and grow in vitro as a cell line? Did this cell line acquire mutations in genes that conferred a fitness advantage (in vivo or in vitro)? Or was there an integration site effect that conferred a fitness advantage? To confirm that downstream effects on gene expression, etc are indeed due to the NUP98::KDM5A fusion, I think it is important to show that these effects (for instance, global gene expression profiles, chromatin accessibility) can be seen in at least one other NUP98::KDM5A cell line.

2) The critical question regarding efficacy and toxicity in vivo is not adequately addressed. The experiment shown in Fig 6 seems to show a subtle survival advantage (11 or 14 d vs 9 d), that would be clinically insignificant. However, there is no mock transplant group, so it isn't clear that the mice died of leukemia vs radiation toxicity. But the most significant criticism is that this experiment does not address the toxicity of CDK12 inhibition on WT hematopoiesis, or other organ function. Despite the in vitro experiments shown in Fig 7, it is not clear that there is a therapeutic window in vivo.

Reviewer #5

(Remarks to the Author)

Reviewer #6

(Remarks to the Author)

The authors have done a very nice job with additional experiments showing that CDK12 may be a general AML vulnerability and with the inhibitor at least, there may be a therapeutic window for impacting AML cells without impacting normal cells. However, in the discussion the authors state "these data suggest that the NUP98::KDM5A-mediated active maintenance of H3K27ac and H3K4me3 marks at their target genes plays a pivotal role in the regulation of their expression." Since some genes show changes in H3K27ac without displaying changes in transcription and some genes show changes in transcription without showing changes in H3K27ac (or H3K4me3), I don't think the data supports this statement. In the discussion, the authors should contextualize their ideas about the mechanism a bit better, and emphasize some alternative models for how the fusion could be functioning, with less of an emphasis on the histone marks which don't seem to correlate all that well with the function of the fusion protein.

Otherwise, the authors have answered all my questions and this is a very nicely done study.

Version 2:

Reviewer comments:

Reviewer #4

(Remarks to the Author)

The authors have presented additional data and arguments to address the remaining questions.

1) The authors didn't present additional data (eg, whole exome sequencing, insertion site sequencing, or analysis of additional NUP98::KDM5a/Nras cell lines) to demonstrate that the phenotype of the NUP98::KDM5a/Nras cell line was not, at least in part, driven by additional acquired mutations. They do present a reasonable argument that additional NUP98::KDM5A (no Kras) cell lines behave similar to the NUP98::KDM5A/Nras cell line in various in vitro experiments.

2) The authors consider an increase from 9 days survival to 11 days survival in a single experiment to be a substantial response. This reviewer would be more cautious in interpretation of that data. Especially in light of the fact that there was no difference in percent malignant cells in the bone marrow at time of death (Fig S6D). The argument that the model is too aggressive to detect clear differences is somewhat weak; I would guess that the model could be made less aggressive by simply injecting fewer cells. It also would have been more convincing if the authors had shown clear evidence of AML (CBC abnormalities, invasion of nonhematopoietic tissue) as recommended by the MMHC consortium (PMID: 12070033), and had a radiation alone control, as sublethal radiation alone is not benign, and will lead to pancytopenia and weight loss at 7-14 days post radiation. The revised discussion is improved.

Reviewer #5

(Remarks to the Author)

Response to the reviewers' comments to manuscript NCOMMS-24-13705 entitled "*Transcriptional and epigenetic rewiring by the NUP98::KDM5A fusion oncoprotein directly activates CDK12-controlled DNA repair*" by Troester et al.

Reviewer #1, expertise in functional genomics (Remarks to the Author):

The study by Troester et al. interrogated epigenetic profiles from pediatric AML patients and healthy controls to identify the molecular consequences of NUP98::KDM5A-mediated transformation. Thereby, the authors derived a GMP-like state of NUP98::KDM5A patient samples with additional stem cell like features. For functional interrogation, the authors developed a fetal cell-derived NUP98:KDM5A/NRas-G12D model with the ability to rapidly deplete NUP98::KDM5A with the dTag13 system. Depletion of NUP98::KDM5A in this model caused rapid differentiation of leukemic cells followed by cell death. Epigenetic profiling upon NUP98::KDM5A depletion highlighted loss of H3K27ac and H3K4me3 at known (Hox cluster) and unknown potential targets of the fusion oncogene. To narrow down direct targets of NUP98::KDM5A more precisely, the authors performed SLAM-Seq highlighting 45 putative direct targets. To derive targetable vulnerabilities, the authors next performed a genome-wide CRISPR-Cas9 screening with 12 genes of the 45 direct targets (SLAM-Seq) being essential for leukemic cells, including known NUP98-rearranged leukemia drivers like the HoxA cluster genes. Individual validation of the CRISPR screening highlighted CDK12 as major susceptibility of the NUP98::KDM5A leukemia cells. Following up on CDK12, the authors analyzed a public patient dataset and found significant overexpression of CDK12 in NUP98-rearranged and NUP98::KDM5A patients. Utilizing shRNA and a CDK12 degrader, sensitivity of NUP98::KDM5A cells (murine and PDX) is highlighted and mechanistically explained by the DNA-repair-associated function of CDK12. The study thereby highlights how NUP98::KDM5A controls epigenetic regulation of gene expression to enable sufficient DNA repair essential for its survival, and how this dependency may be exploited therapeutically to treat a severe form of pediatric AML. Given the authors can provide additional pre-clinical data on CDK12-inhibition, the study will add significantly to the understanding and the prospective treatment of NUP98::KDM5A-rearranged pediatric AML.

We thank this reviewer for the positive evaluation. As we outline in our response below, we invested significant efforts into further characterizing the role of CDK12 in NUP98 fusion-driven AML, and we are convinced that our new results add more insights into this aspect of the work. We also provide additional results and analyses in response to the other points raised by this reviewer.

Major points

1. Relating to Figure 1: Could the authors please comment on the populations used for ATAC-seq. In the Material and Methods section ???Blast populations??? are mentioned without further specification. Could the authors please also elaborate on the ???mature??? cells used for comparison? Taking into account that NUP98::KDM5A is strongly enriched in AMKL, the finding of close proximity to GMPs instead of MEPs is interesting but may need further analysis.

As most of the data used in Figure 1 were extracted from other publications, we were unable to obtain more details on the cell populations from these reports. But we added a description of how blast populations were identified for the samples obtained by our local clinical partners in the Methods section of the revised manuscript. To elaborate on the status of "mature cells", we changed the text in the legend of Figure 1 to "mature myeloid cells". More information about these cell types can be found in Supplemental Table 1. Finally, we repeated the minimum spanning tree analysis as shown in Figure 1B upon including expression profiles of erythrocytes. As shown below (**Reviewer Figure 1**), this analysis placed several NUP98::KDM5A AML samples in close vicinity to mature erythrocytes, indicating a high degree of erythroid-specific gene expression in these cases. Yet, all NUP98::KDM5A samples still exhibited a close relationship to GMPs and

mature myeloid cells but not to MPPs, MEPs and stem cells. While the close relatedness of individual NUP98::KDM5A AML samples to mature erythrocytes could be investigated further in the future, this analysis did not change our original conclusion about the overall relationship of NUP98::KDM5A AML to distinct normal progenitor subtypes.

Reviewer Figure 1: Minimum spanning tree analysis of NUP98::KDM5A AML cells and the indicated normal cell types of the hematopoietic system.

2. ATAC-seq analysis of NUP98:KDM5A samples (vs. rest) deviates in one patient. Can the authors derive any information explaining this deviating chromatin accessibility (e.g. blast population immunophenotype)?

We thank the reviewer for this comment. The patient sample NUP98:KDM5A #1 which deviates from the others in terms of chromatin accessibility is classified as AML M5, which might explain this difference. More information about the classification and characteristics of NUP98 fusion-expressing AML samples can be found in Supplementary Table 1.

3. Did the authors perform ATAC-seq of their murine model and compare it to the patient samples?

To address this point, we performed ATAC-seq in our dTAG-NUP98::KDM5A cell line and compared the results to human patient data. We found a strong correlation of accessible regions, further substantiating the clinical relevance of our mouse model for the analysis of the human disease. These new results are shown in Supplementary Figure S2G in the revised manuscript.

4. Did regions of reduced H3K27ac and H3K4me3 upon degradation of NUP98:KDM5A (murine model) overlap with regions of open chromatin in patients, or with H3K27Ac and H3K4me3 in the analyzed PDX model?

To address this question, we restricted our analysis on genes that are proximal to the regions showing differential H3K27AC or H3K4me3 levels upon dTAG13 treatment and compared their accessibility in ATAC-seq data in the mouse model vs. patient cells. We found that the accessibility of these genes was highly conserved between mouse and human cells, further indicating that the NUP98:KDM5A fusion oncoprotein controls similar regulatory circuits in mouse and human AML. These new results are shown in Supplementary Figure S3F in the revised manuscript.

5. The authors noted loss of H3K27ac and loss of CDK12 expression at 8h post NUP98::KDM5A degradation. What is the ATAC-seq and histone status of CDK12 in patient samples?

We found that the *CDK12* locus is accessible in NUP98::KDM5A patient samples. These new results are shown in Supplementary Figure S6A in the revised manuscript. H3K27ac CUT&Tag data of the NUP98::KDM5A PDX samples for the *CDK12* locus are shown in Figure 6B.

6. SLAM-seq resulted in detection of 45 deregulated genes 1h-2h post NUP98::KDM5A degradation. It would be helpful to highlight CDK12 (as main gene of interest for follow-up studies) in Figure 4B.

We highlighted *Cdk12* in the volcano plot shown in Figure 4B in the revised version of the manuscript.

7. On page 9, lines 250-251, the authors note that other genes, such as CDK12, are unexplored or implicated in other cancers. The authors may include citation 68 and the work by Hoshii and colleagues (DOI 10.1016/j.cell.2018.01.032).

Thank you for pointing this out. We have added these citations in the revised version of the manuscript.

8. The authors performed a genome-wide CRISPR screen and highlighted the overlap with their SLAM-seq. Could the authors please comment on the 87 genes found deregulated 8h post NUP98::KDM5A degradation?

We found that 30% (26 genes) of the 87 genes whose expression was downregulated early upon NUP98::KDM5A degradation are essential in the CRISPR/Cas9 screen. This selection contains most of the 12 essential direct NUP98::KDM5A target genes identified in Figure 5B.

9. Also relating to this, did the authors see changes of DNA repair genes in their gene expression profiling upon NUP98::KDM5A (at later time points)? This refers particularly to the genes highlighted in Figure 6F. The authors only refer to a previously published dataset.

We thank the reviewer for this question. Upon inspection of our RNA-seq datasets, we found a trend towards downregulated expression of DNA repair genes at both 8h and 24h time points after NUP98::KDM5A degradation. While the observed changes were not statistically significant at these time points, we find a highly significant downregulation of DNA repair genes after transcriptional downregulation of NUP98::KDM5A, as published earlier¹ (Figure 7D, E in the revised version of the manuscript). This likely indicates that expression changes in these genes involved in DNA repair need days not hours to manifest in NUP98::KDM5A AML cells.

10. The authors highlight the sensitivity of NUP98::KDM5A cells to CDK12 KD or degradation. Can the authors provide any other NUP98-rearranged and non NUP98-rearranged data to further underline the high sensitivity of NUP98::KDM5A leukemia?

We performed several experiments to address this point. First, we knocked down *Cdk12* via shRNA in murine AML cell lines driven by the *KMT2A::MLL3* fusion and N-terminal CEBPA mutations, respectively. *Cdk12* knockdown caused significant antiproliferative effects in these cell models. Furthermore, we tested the effect of the CDK12 inhibitor THZ531 and the CDK12 degrader BSJ-4-116 in additional murine and human AML models expressing different driver mutations. These results clearly indicate that both inhibition of CDK12 kinase activity as well as CDK12 degradation was incompatible with AML cell proliferation, in line with data reported by Savoy et al². These new results are shown in Supplementary Figure S6B, C (*Cdk12* shRNA), Figure 7 (CDK12 inhibition by THZ531) and Supplementary Figure S7B (CDK12 degradation by BSJ-4-116) in the revised manuscript.

11. Can the authors (at least partially) rescue the NUP98::KDM5A degradation via CDK12 overexpression?

To test this, we exogenously expressed CDK12 wild type and a kinase-dead mutant of CDK12 (CDK12^{D873N}) in dTAG-NUP98::KDM5A cells. Cells were treated with dTAG-13 and proliferation was measured over 11 days. While mock-transduced cells and cells expressing the CDK12^{D873N} kinase-dead mutant showed some initial expansion but ceased to proliferate between day 7 and day 11, cells expressing the CDK12 wild type construct continued to proliferate at a significantly higher rate until day 11. This indicates that sustained expression of kinase-active CDK12 can partially overcome the proliferation arrest that is associated with dTAG-13-mediated NUP98::KDM5A degradation. These results are shown in Figure 7F in the revised version of the manuscript.

12. Can the authors provide any in vivo data for efficacy of CDK12 inhibition of NUP98:KDM5A cells (e.g. their murine model; if more suitable for in vivo use, a CDK12/13-Inhibitor will still underscore the results).

As application reliable protocols for the in vivo use of the CDK12 inhibitor THZ531 and the CDK12 degrader BSJ-4-116 are missing, it is unclear if these compounds are suitable for in vivo use. We therefore decided to validate pre-clinical relevance of our findings using an in vivo RNAi experiment. We introduced Doxycycline-inducible variants of *Cdk12*-targeting shRNAs into NUP98::KDM5A AML cells (CD45.2) stably expressing the reverse Tet-transactivator (rtTA3). In the system used by us, shRNA expression is coupled to an IRFP670 marker, which enables the identification of shRNA-expressing cells by flow cytometry. Cells were transplanted into recipient CD45.1 animals and expression of the shRNAs was induced by addition of Doxycycline to the drinking water of mice three days after transplantation. Disease progression was monitored by in vivo bioluminescence imaging, as the cells also express a luciferase gene¹. Knockdown of *Cdk12* significantly extended the survival of recipient animals, showing that CDK12 is also required for the growth of NUP98::KDM5A AML cells in vivo. Immunophenotyping of leukemic blasts in moribund animals showed that *Cdk12*-deficient, IRFP670/CD45.2-positive AML cells displayed reduced levels of the progenitor marker c-Kit, which is in line with their reduced leukemogenic potential. Altogether, these data strengthen the conclusion that CDK12 represents a faithful target in NUP98::KDM5A-driven AML. These results are shown in Figure 6E-G and in Supplementary Figure 6D in the revised version of the manuscript, and the comments to these points are related to our response to point #5 of Reviewer 3 and point #10 of Reviewer 6.

13. Fan and colleagues (citation 19) noted functional redundancy of CDK12 and CDK13. Interestingly, the authors note a selective dependency on CDK12 in NUP98::KDM5A leukemia. This should be pointed out and CDK13 expression and depletion in the CRISPR-screening may need additional attention.

In our CRISPR/Cas9 screen in NUP98::KDM5A cells, knockout of CDK13 caused a slight proliferative disadvantage, but this was below the threshold we used to identify essential genes. Furthermore, dTAG13 treatment for 8h and 24 h caused a slight reduction in CDK13 expression, which was also below the threshold we used to call differentially expressed genes. Therefore, we conclude that while functional redundancy has been reported for CDK12 and CDK13, these genes have distinct functions in the context of NUP98 fusion-driven AML.

Minor points

1. The authors note more subtle depletion of HoxA genes with shRNA knockdown compared to CDK12. Was this also true for the CRISPR-based screening? The authors hypothesize on

redundant function, which is reasonable and should be true for the CRISPR-based approach, too. Otherwise less reduction via shRNA may cause the observed effect.

To answer this question, we compared the effect of CRISPR/Cas9-mediated knockout of *HoxA* genes in the genome wide screen with the effects of *Cdk12* knockout. We found that knockout of several *HoxA* genes, including *HoxA10*, *HoxA9*, *HoxA3*, *HoxA6*, caused stronger effects than knockout of *Cdk12* in the CRISPR screen. However, the mutational inactivation of *HoxA7*, *HoxA1*, *HoxA5*, *HoxA2*, *HoxA11*, *HoxA4* and *HoxA13* had less severe effects. Therefore, the differential effects observed in the shRNA experiments are likely explained by less efficient reduction of CDK12 levels in this scenario.

2. The authors may tame down the statement in lines 457-459. Indeed, functional and mechanistic studies on CDK12 in myeloid malignancies have been sparse and this work adds significantly to the knowledge in the field.

The corresponding statement has been adjusted.

Reviewer #2, expertise in epigenetics, AML and pediatric cancers (Remarks to the Author):

In this interesting, well-conducted and clearly written study, Troester et al use a combination of epigenomic, transcriptomic, and functional assays to identify CDK12 as a dependency in NUP98Ar leukemias. The authors first identify differentially-accessible peaks in NUP98Ar AMLs compared to other AMLs and healthy hematopoietic cells using ATAC-seq. As NUP98 fusions are difficult to study and relevant models are scarce, they develop a murine HSPC model with a dTAG fusion that allows for investigation of the impact of acute fusion loss on the transcriptome and epigenome of NUP98::KDM5A fusion-driven cells. They then use this model to perform a CRISPR dependency screen, and by integrating with SLAM-seq and CUT&Tag data identify CDK12 as a dependency in NUP98Ar AML. They further pin the dependency on the role of CDK12, a kinase that phosphorylates RNA Pol II, on its regulation of DNA damage response genes. Targeting CDK12 with small molecule degraders leads to cell death in multiple NUP98Ar. It would be nice to see a survival advantage in a NUP98Ar PDX treated with CDK12 inhibitor, but given difficulty with modeling this disease the current results sufficiently show a role and potential for this approach in NUP98Ar patients. NUP98Ar AML is difficult to study and mechanistic insights like those reported here represent an important advance with relevance for pediatric leukemia researchers.

We thank this reviewer for their insightful comments and are glad to see that they endorse our work. We performed several experiments and analyses to address the points raised by this reviewer, as outlined in detail below.

Major Points

It seems like the authors performed ATAC-seq on patient samples and report generally descriptive but not particularly informative results, and then independently developed their model to understand how fusion binding drives a novel dependency. As reported, these approaches seem distinct and they should try better integrate the ATAC data with the rest of the manuscript, or even consider leaving it out as it distracts from the rest of the paper (e.g. they do not thoroughly explore CDK12 or other direct targets in this dataset, and even turn to previously-published RNA-seq for important comparisons). Many of the comments below would be addressed satisfactorily by removing the ATAC dataset.

1. Figure 1A: A major focus of the manuscript is on their ATAC-seq dataset. The authors perform bulk ATAC-seq on NUP98::NSD1 and NUP98::KDM5A positive AMLs and integrate with publicly available AML and HSPC datasets. While the AMLs generally cluster together and are mostly distinct from normal cells, the distributions of the NUP98 fusions within all AMLs do not show any specific pattern. Previous studies showed that NUP98::KDM5A fusions are found across all FAB types with enrichment in M7. Can the authors re-annotate 1A using FAB classification and/or fusion type (e.g. KMT2Ar, CBF::GLIS2) to determine whether this explains the variance? An alternative is that data quality explains this variance, I did not see mention or report of QC metrics such as FrIP scores, can they be added to Supp Table 1?

To address this question, we re-annotated the pediatric AML samples in the PCA analysis of the ATAC-seq data for FAB type and fusion type. Although this information was not available for all samples in the dataset, both analyses show a clear sub-clustering of pediatric AML samples of the M5 type which appears to partially overlap with the presence of the NUP98::KDM5A fusion. These results are shown in Supplementary Figure 1A, B in the revised version of the manuscript. FrIP scores of the ATAC-seq datasets have been added to Supplementary Table 1.

2. Fig 1C-D show 1921 regions that are more accessible in NUP98r versus GMP cells. Are there any regions that are less accessible in NUP98r than in GMP? Since epigenetic silencing is a major driver of cellular differentiation, analyzing both more and less accessible regions could

show cell of origin more convincingly. In addition, motif analysis of the 1921 peaks should be included to identify TFs that may mediate differential accessibility.

We thank the reviewer for this comment. We found 6054 genomic regions that were less accessible in NUP98::KDM5A patient samples compared to GMP. These regions are also highly accessible in HSCs and associated with reduced accessibility in monocytes and neutrophils, which is in line with the comment of the reviewer that epigenetic silencing is a major driver of cellular differentiation. These results are shown in Supplementary Figure 1C in the revised version of the manuscript.

We also performed TF motif analysis on the 1921 regions that were more accessible in NUP98::KDM5A AML vs. GMP and also in the 6054 less accessible regions. While consensus motifs of GATA-, and AP1-families were enriched in genomic regions with high accessibility in NUP98::KDM5A AML, we found an enrichment of binding sites for ETS and IRF transcription factors in regions that were less accessible in NUP98::KDM5A-expressing cells. These data represent important additional insights into our epigenomic analysis of NUP98::KDM5A-driven AML and are shown in Supplementary Figure 1F in the revised version of the manuscript.

3. Fig 1D-E: The data are presented in a manner suggesting that the NUP98 fusion is directly responsible for the majority of these changes. The authors do not report ChIP-seq or CUT&RUN results for the NUP98 fusion itself. Please integrate NUP98 ChIP-seq from ref 9 onto the differentially-accessible peaks, and in particular the 1921 enriched compared to GMPs, to assess which are direct fusion targets.

To address this point we analyzed the signal intensities of NUP98::KDM5A chromatin binding from dataset previously published by us¹ for the 1477 murine genes that correspond to the 1921 regions that were more accessible in NUP98::KDM5A AML cells vs. GMPs. Indeed, the signal intensities were higher for this subset of genes when compared to all other genes, suggesting that chromatin binding of the fusion oncoprotein is involved in the epigenomic changes at these loci. These results are shown in Supplementary Figure 1E in the revised manuscript.

4. Fig 1 D-E: ATAC-seq is an important but indirect surrogate for gene expression. The authors should integrate their differential ATAC peaks with RNA-seq from NUP98 patients that the authors used in Figure 6A (or PMID: 36815378) to assess which regions are associated with differential gene expression.

Thank you for this interesting question. We analyzed the expression levels of genes associated with the 1921 more accessible regions in pediatric AML samples with different mutational backgrounds. In line with a direct and specific role of NUP98::KDM5A in the regulation of these genes, we found that within this cohort, their expression was highest in AML patients with the NUP98::KDM5A fusion. These results are shown in Supplementary Figure 1D in the revised manuscript.

5. Lines 144-146 are not fully substantiated as the authors report examining histone modifications associated with gene activation only. To further corroborate this statement, the authors should at least report how many peaks were present in GMP that were not identified in NUP98r samples. Few inaccessible peaks in GMPs would suggest that NUP98 fusions drive a more accessible landscape compared to healthy cells.

Our analysis of less accessible peaks in NUP98::KDM5A AML vs. GMP (see also our response to comment #2 and Supplementary Figure S1C) suggests that chromatin is globally less accessible in NUP98::KDM5A samples. This is in line with the statement of this reviewer in comment #2, noting that global epigenetic silencing drives differentiation. It is also in agreement with the observation that global chromatin accessibility patterns of AML patient samples cluster between healthy hematopoietic progenitors and mature myeloid cells (Fig. 1A, B).

6. Regarding the degranulation system, the authors cite reports that LLPS is important for NUP98 fusion

activity. However, the quick and substantial loss of the fusion after dTAG13 molecule treatment is inconsistent with profound stability often afforded by LLPS. The authors should perform IF for the fusion in their dTag system to determine whether puncta consistent with LLPS formation is observed. If such puncta are not seen, it would suggest that LLPS-independent fusion activity is important for transformation, and if present, enhances the reliability of their model compared to prevailing models for NUP98r fusion activity.

We added confocal images of dTAG-GFP-NUP98::KDM5A cells at a higher magnification to Fig. 7G and Supplementary Figure S7F, G in the revised manuscript. These images clearly show that the NUP98::KDM5A fusion oncoprotein localizes to characteristic nuclear punctae that have been reported by us and others³⁻⁵.

7. Figure 3A, C: Given F1's dependence on ATAC-seq and that more ATAC peaks are present than K27ac or K4me3 C&T peaks, the authors should perform ATAC in their degron system and directly compare to differentially-enriched regions reported in F1.

Thanks for this suggestion. We treated dTAG-NUP98::KDM5A cells with dTAG13 (35nM) or DMSO for 8 hours and subsequently performed ATAC-seq. We chose this time point to be able to directly compare the results to our CUT&Tag data on H3K27ac and H3K4me3 marks. Our analysis revealed a global trend towards more inaccessible regions, only very few genomic regions showed differential accessibility regions in the dTAG13-treated cells (7 regions, FDR < 0.05, all down in dTAG13 treated samples, **Reviewer Figure 2**). Therefore, it is likely that the histone marks investigated by us show a more dynamic distribution upon NUP98::KDM5A degradation than changes in chromatin accessibility. Early changes in gene expression after NUP98::KDM5A degradation are therefore likely not mediated by changes in chromatin accessibility.

Reviewer Figure 2: Volcano Plot showing differentially accessible regions in dTAG-NUP98::KDM5A AML cells treated with dTAG-13 or DMSO for 8 h as measured by ATAC-seq.

8. Figure 3E: The authors should plot RNA-seq signal (153 downregulated and 96 upregulated) from 8h and 24h timepoints from the degron experiment onto the differentially-enriched regions from F1 to assess the relevance of these findings to patient samples.

To address this point, we plotted the ATAC-seq signal of the 153 downregulated and 96 upregulated genes (from the RNA-seq data of samples treated with dTAG13 for 24 h) from NUP98::KDM5A patient samples and the dTAG-NUP98::KDM5A model. This analysis revealed that chromatin accessibility was significantly higher at genes that were downregulated upon NUP98::KDM5A degradation. These new data are shown in Supplementary Figure 4B in the revised version of the manuscript.

9. Figure 3F/G: There is a missed opportunity to integrate the degron and patient ATAC-seq datasets here as well. Please plot ATAC signal from all samples in 1E onto the 38 NUP98::KDM5A "target genes" to assess relevance and specificity to NUP98Ar patient sample epigenomes.

To address this suggestion, we plotted the ATAC-seq signal for the regions associated with the 38 NUP98::KDM5A target genes for all pediatric AML patient samples from Fig. 1E. Chromatin accessibility at these genes was highest in the NUP98::KDM5A patient samples, substantiating a direct role of the NUP98::KDM5A fusion oncoprotein in inducing and maintaining high expression of these genes. These new data are shown in Supplementary Figure 4D in the revised version of the manuscript.

10. Figure 6D: Please treat other AML cell lines with predicted relative insensitivity or dependency on CDK12 with BSH-4-116. The authors should consider adding KASUMI-1, K562 (both predicted to be insensitive based on DepMap) and AML193 and OCI-AML3 (predicted to be sensitive). If KASUMI-1 or K562 show relative insensitivities (>100nM), the authors should perform RNA-seq from those cells treated with BSJ and compare to their NUP98Ar results to assess whether their reported DDR defect sensitivity is specific to NUP98Ar leukemias, as they suggest.

This comment is in line with comment #10 of Reviewer 1. We performed several experiments to address this point. First, we knocked down *Cdk12* via shRNA in murine AML cell lines driven by the *KMT2A::MLL3* fusion and N-terminal CEBPA mutations, respectively. *Cdk12* knockdown caused significant antiproliferative effects in these cell models. Furthermore, we tested the effect of the CDK12 inhibitor THZ531 and the CDK12 degrader BSJ-4-116 in additional murine and human AML models expressing different driver mutations. These results clearly indicate that both inhibition of CDK12 kinase activity as well as CDK12 degradation was incompatible with AML cell proliferation, in line with data reported by Savoy et al. These new results are shown in Supplementary Figure S6B, C (*Cdk12* shRNA), Figure 7 (CDK12 inhibition by THZ531) and Supplementary Figure S7B (CDK12 degradation by BSJ-4-116) in the revised manuscript. Although we observed differences in the sensitivity of AML cells to THZ531 or BSJ-4-116 we did not find a resistant cell line in our selection of cellular models and therefore did not perform RNA-seq analysis as suggested.

Minor Points:

1. Figure 2: The authors should report Nras levels (Western blotting is sufficient) in the degron system to assess whether they change in the context of NUP98 fusion loss. If not impacted, this would further demonstrate the differentiation phenotype is solely dependent on the fusion. This is important because while NRAS mutations can co-occur in NUP98r, some reports suggest they are not present in the majority of cases.

Western Blot analysis showed that NRAS levels did not change substantially upon degradation of NUP98::KDM5A. These new results are shown in Supplementary Figure 2H in the revised version of the manuscript.

2. Figure 5A: Were any of the 33 "direct" NUP98::KDM5A target genes found in the 645 up list? Presumably no but please state in the text. Would be interesting if the fusion drives expression of a potential tumor suppressor that is mitigated by expression of other targets.

We found that two of the direct NUP98::KDM5A target genes lead to a proliferative advantage upon knock out in the CRISPR screen. These are *Mxi1* (which was previously described as a tumor suppressor) and *Tnrc18*. This information was included in the results section in the revised version of the manuscript.

3. Line 628 typo ???iRFP670-postive???

Thanks for spotting this mistake. It has been corrected in the revised version of the manuscript.

Reviewer #3, expertise in DNA repair in AML (Remarks to the Author):

Chromosomal translocations involving the NUP98 locus are among the most prevalent rearrangements observed in pediatric acute myeloid leukemia (AML). AML expressing NUP98 fusion proteins are characterized by a distinct transcriptional signature, marked by high expression of HOXA and MEIS1 genes, and are associated with unfavorable clinical outcomes. While NUP98 fusion proteins are established drivers of AML development, the precise molecular mechanisms underpinning NUP98 fusion protein-dependent leukemogenesis remain elusive, thereby impeding the development of targeted therapies aimed at improving the clinical outcomes for patients with NUP98 fusion-positive AML.

This study therefore investigated the epigenetic and transcriptional signatures controlled by NUP98 fusion proteins in AML, with a primary focus on the specific NUP98-KDM5A fusion oncoprotein, which is observed in ~2% of all pediatric AML patients. Using a series of genome-wide approaches, Troester et al. identified a unique epigenetic signature in NUP98 fusion-driven AML samples, marked by overlapping activating histone marks (e.g., H3K27ac, H3K4me3). To further delineate the molecular impact of NUP98-KDM5A fusion in AML, the authors developed an elegant murine AML model where a degradation tag (dTAG) system was fused to NUP98-KDM5A and expressed in NrasG12D murine fetal liver-derived hematopoietic stem cell progenitors, before being transplanted in irradiated recipient mice. Degradation of the dTAG-NUP98-KDM5A fusion oncoprotein led to cell cycle arrest, differentiation and apoptosis of leukemic cells, highlighting the dependency on NUP98-KDM5A in this murine model. Importantly, NUP98-KDM5A degradation correlated with a significant and localized reduction in H3K27ac and, to a lesser extent, H3K4me3 levels. Differential expression analysis by RNA-seq identified 153 downregulated genes upon NUP98-KDM5A degradation, while SLAM-seq analysis revealed 45 target genes that exhibited significant loss in H3K27ac mark following NUP98-KDM5A degradation and high binding occupancy of NUP98-KDM5A in the promoter of these genes. To gain further insight into the genetic dependencies of murine NUP98-KDM5A AML cells, the authors performed a genome-wide CRISPR-based loss-of-function screen and identified 4105 genes that caused a fitness defect in NUP98-KDM5A driven AML cells. Interestingly, 12 of the 45 target genes previously identified were essential for NUP98-KDM5A driven AML cells and validated in a shRNA-based competition growth assay in vitro. The authors focused on the cyclin-dependent kinase CDK12 and its role in NUP98-KDM5A driven AML cells. Depletion of Cdk12 by shRNA impaired the proliferation of murine NUP98-KDM5A AML cells in vitro, while pharmacological inhibition of CDK12 induced cell death in both murine and human NUP98-KDM5A AML models. Depletion of NUP98-KDM5A or pharmacological inhibition of Cdk12 resulted in a significant downregulation of genes linked to DNA repair and induced the formation of γ -H2AX foci in AML cells, suggesting that CDK12-dependent control of DNA repair gene expression may promote the survival of NUP98-KDM5A AML cells in vitro.

Overall, the manuscript is extremely well written, and the authors employed elegant and cutting-edge approaches to map the molecular mechanisms controlled by NUP98-KDM5A fusion oncoprotein in AML cells. For the most parts, the conclusions drawn by the authors align with the data provided in this manuscript. At this stage, the study provides limited validation(s) in human NUP98-KDM5A AML cells and no functional confirmation in vivo. Furthermore, some key controls are lacking, which should be provided before the manuscript can be considered for publication:

We thank this reviewer for their very positive evaluation. We performed a series of additional experiments and analysis to address this reviewer's comments. A detailed response to all points is provided below.

1. Epigenomic characterization of pediatric NUP98-driven AML samples: PCA analysis revealed that pediatric NUP98-driven AML samples are broadly distributed compared normal healthy blood cells (Fig.1A), while inter sample distance visualization by minimum spanning tree placed NUP98-driven AML samples between granulocyte-monocyte progenitor cells and mature cells (Fig.1B).

Can the authors elaborate on this ???discrepancy???? Where would adult vs. pediatric AML samples localize in this trajectory analysis? More generally, I failed to understand why the authors would expect distinct ATAC-seq clustering of pediatric vs. adult AML samples based on differences in somatic mutations (line 118-120). One pediatric sample greatly differs from the 4 NUP98-KDM5A-driven AML sample analyzed (Fig.1E). Could the authors provide more details about the specificity of this patient (e.g., characteristics of the NUP98-KDM5A fusion, mutational landscape????)?

Thank you for these insightful comments. The discrepancy noted by the reviewer likely arises from the fact that PCA represents both similarity and dissimilarity between samples in two dimensions, while the minimum spanning tree analysis only connects each sample to its closest neighbor by the shortest path, but not all relationships between points are considered. However, we would like to stress that overall, both analyses place the AML samples between healthy hematopoietic progenitors and mature myeloid cells. We added the adult AML samples to the minimum spanning tree analysis to confirm this observation (**Reviewer Figure 3**).

Reviewer Figure 3: Minimum spanning tree analysis of adult and pediatric AML samples and indicated populations of normal hematopoietic progenitors as well as mature myeloid cells.

We agree with the reviewer's comment that ATAC-seq clustering of pediatric vs. adult AML samples based on differences in somatic mutations cannot be expected. We therefore deleted the sentence in lines 118-120.

Finally, the patient sample NUP98::KDM5A #1, which deviates from the others in terms of chromatin accessibility is classified as AML M5, which might explain this difference. More information about the classification and characteristics of NUP98-fusion expressing AML samples can be found in Supplementary Table 1. This comment is in line with comment #2 of Reviewer 1.

2. Characterization of the dTAG-NUP98-KDM5A murine AML model: The authors developed a unique murine AML model to define the molecular mechanisms underpinning NUP98-KDM5A fusion protein-dependent leukemogenesis. Still, limited comparison with NUP98-KDM5A-driven pediatric AML samples has been completed to ensure that their model is recapitulating the molecular and clinical features of NUP98-KDM5A-driven pediatric AML. For example, what is the

overlap in term of differential gene expression between NUP98-KDM5A-driven pediatric AML samples and murine cells?

The mouse model used in this work was generated as described in detail in earlier work of our group¹. In this publication, we provide an extensive characterization of the similarities of our models to human NUP98 fusion-driven AML. In addition, we compared our dTAG-NUP98::KDM5A model to human NUP98::KDM5A patient samples with respect to chromatin accessibility and gene expression. We found a strong and significant correlation in both analyses. These new results are shown in Supplementary Figure 2G in the revised version of the manuscript. Moreover, we show that the 153 genes that were downregulated upon NUP98::KDM5A degradation (i.e. whose expression is maintained by the fusion oncoprotein) show high chromatin accessibility in human NUP98::KDM5A-expressing AML. Finally, the 38 direct target genes identified in our murine model showed highest chromatin accessibility in NUP98::KDM5A patient samples when compared to other pediatric AML samples. Altogether, we believe that these datasets underscore the clinical relevance of our murine model for studies of human AML driven by the NUP98::KDM5A fusion. These new results are shown in Supplementary Figure 4B, D in the revised version of the manuscript, and the comments are related to our response to points #8 and #9 of Reviewer 2.

3. Identification of NUP98-KDM5A target genes: RNA-seq-based analysis identified a subset of 38 target genes that were significantly downregulated and displayed loss in H3K27ac mark upon fusion protein degradation, while exhibiting high binding occupancy of NUP98-KDM5A in the promoter of these genes (Fig.3). On the other hand, SLAM-seq analysis identified 45 target genes whose expression was significantly downregulated upon fusion protein degradation (Fig.4). What is the overlap between the RNA-seq-based 38 target genes and the SLAM-seq-based 45 target genes? More generally, some key controls are lacking in this set of experiments, including the re-expression of the fusion protein to ensure that these changes are specific to NUP98-KDM5A and a dose-response analysis of dTAG13 on the de novo synthesis of mRNA to distinguish direct vs. indirect control of NUP98-KDM5A. Furthermore, no differential expression analysis of these target genes in human AML samples is provided at this stage, which limits the interest/impact of these findings.

Thank you for these comments. The overlap of the 38 RNA-seq based target genes and the 45 direct target genes identified by SLAM-seq contains 24 genes that are identified by both approaches. This intersection contains 10 of the 12 direct essential targets of NUP98::KDM5A, including *Cdk12*.

The requested experiments including re-expression of the NUP98::KDM5A fusion oncoprotein upon dTAG13 wash-out and a more detailed dose-dependent analysis of gene expression changes upon dTAG13 treatment are part of a follow-up research project that is currently ongoing in the laboratory. The data is not available yet.

Finally, we performed differential expression analysis of the 45 direct NUP98::KDM5A target genes identified in our SLAM-seq approach using datasets from pediatric AML patients with different genetic aberrations. The expression levels of this gene set was highest in human AML samples with NUP98::KDM5A, which further highlights the relevance of our findings for an improved mechanistic understanding of human AML with NUP98 fusions. These new results are shown in Figure 4I in the revised version of the manuscript.

4. Identification of essential genes in NUP98-KDM5A AML cells: Fitness genes can be subdivided between core and context-dependent essential genes. To what extent the 4105 genes that caused a fitness defect in NUP98-KDM5A driven AML cells are core essential genes in any cell type or AML specific fitness genes? To what extent are these genes dysregulated in NUP98-KDM5A-driven AML vs. AML vs. HSC cells?

We annotated core essential genes in the genome wide CRISPR/Cas9 screen based on the *Mus musculus* essentialome as published earlier⁶. This publication reports a list of 870 core essential genes, of which 852 are among the 4105 genes we identified as essential for the proliferation of

NUP98::KDM5A AML cells in our screen. This high degree of recovery of previously annotated essential genes underscores the high quality of the screening data. We then analyzed the expression of the remaining 3253 context-dependent essential genes in publicly available datasets of pediatric AML with different genetic abnormalities. Consistent with a role for NUP98::KDM5A in driving the expression of genes that mediate context-specific genetic dependencies. AML samples with the NUP98::KDM5A fusion showed highest expression levels of this gene set (**Reviewer Figure 4**).

Reviewer Figure 4: Gene expression analysis of 3253 genes identified as NUP98::KDM5A-specific dependencies in the genome wide CRISPR/Cas9 screen.

5. Characterization of CDK12 as an essential gene in NUP98-KDM5A AML cells: Rescue and separation-of-function (kinase-dead mutant) experiments should be considered in Cdk12-depleted murine AML cells and human AML cells to confirm the essential role of CDK12 in NUP98-KDM5A driven AML. RNA-seq analysis suggests that pharmacological inhibition of Cdk12 disrupts HR-mediated DNA repair in AML cells, in line with prior study (Johnson et al. Cell Reports 2016). Does CDK12 inhibition impairs the recruitment of RAD51 to γ-H2AX foci in AML cells? Validation of these findings in vivo should be considered to strengthen these findings.

We performed several experiments to address these points. First, we exogenously expressed CDK12 wild type and the murine equivalent of a reported kinase-dead mutant of CDK12 (CDK12^{D873N})⁷ in NUP98::KDM5A cells that stably expressed Doxycycline-inducible shRNA constructs to knockdown the endogenous *Cdk12* gene by targeting its 3'UTR. In the system used by us, shRNA expression is coupled to an IRFP670 fluorescent reporter, which allows tracking of shRNA-expressing cells over time. Cells were treated with Dox and growth kinetics of shRNA-expressing/IRFP670-positive cells vs. shRNA(IRFP670-negative cells) was followed over 11 days by flow cytometry. While mock-transduced cells and cells expressing the CDK12^{D873N} kinase dead mutant were depleted over time, cells expressing the CDK12 wild type construct continued to proliferate at a significantly higher rate until day 11. This indicates that the kinase activity of CDK12 is required for the proliferation of NUP98::KDM5A AML cells. These results are shown in Figure 6D in the revised version of the manuscript.

To test whether CDK12 inhibition impairs the recruitment of RAD51 to γ-H2AX foci we treated dTAG-GFP-NUP98::KDM5A cells with either dTAG13 (35 nM, 72 h), The CDK12 degrader BSJ-

4-116 (100 nM, 4 h) or with DMSO (72 h). Cells were stained with antibodies against RAD51 and g-H2AX and analyzed by confocal imaging (**Reviewer Figure 5**). Degradation of both NUP98::KDM5A and CDK12 induced DNA damage, as visualized by an increase in g-H2AX signal. We also found a clear upregulation of RAD51 under these conditions. The RAD51 signal was found to co-localize with the g-H2AX signal, indicating recruitment of RAD51 to DNA damage foci that are marked by g-H2AX. However, as CDK12 degradation still led to efficient RAD51/g-H2AX colocalization, we conclude that CDK12 does not play a role in RAD51 recruitment to g-H2AX foci.

Reviewer Figure 5: Representative images (A) and quantification (B) of confocal imaging analysis of g-H2AX and RAD51 staining in dTAG-GFP-NUP98::KDM5A cells treated with dTAG13 or BSJ-4-116 as indicated in the text.

Finally, to validate *Cdk12* as an essential direct target gene for NUP98::KDM5A AML cells, we performed an in vivo RNAi experiment. We introduced Doxycycline-inducible variants of *Cdk12*-targeting shRNAs into NUP98::KDM5A AML cells (CD45.2) stably expressing the reverse Tet-transactivator (rtTA3). In the system used by us, shRNA expression is coupled to an IRFP670 marker, which enables the identification of shRNA-expressing cells by flow cytometry. Cells were transplanted into recipient CD45.1 animals and expression of the shRNAs was induced by addition of Doxycycline to the drinking water of mice three days after transplantation. Disease progression was monitored by in vivo bioluminescence imaging, as the cells also express a luciferase gene¹. Knockdown of *Cdk12* significantly extended the survival of recipient animals, showing that CDK12 is also required for the growth of NUP98::KDM5A AML cells in vivo. Immunophenotyping of leukemic blasts in moribund animals showed that *Cdk12*-deficient, IRFP670/CD45.2-positive AML cells displayed reduced levels of the progenitor marker c-Kit, which is in line with their reduced leukemogenic potential. Altogether, these data strengthen the conclusion that CDK12 represents a faithful target in NUP98::KDM5A-driven AML. These results are shown in Figure 6E-G and in Supplementary Figure 6D in the revised version of the manuscript, and the comments to these points are related to our response to point #12 of Reviewer 1 and point #10 of Reviewer 6.

Reviewer #4, expertise in NUP98 fusion oncoproteins, AML and epigenetics (Remarks to the Author):

This manuscript, by Troester and colleagues, investigates the role and mechanisms through which the NUP98::KDM5A fusion oncoprotein mediates its oncogenic functions in AML. Comparison of NUP98::KDM5A AML with GMPs identified promoter sites with increased accessibility and an enrichment in activation marks, particularly H3K27ac. This enrichment of activation marks and chromatin accessibility was decreased upon ligand induced protein degradation of NUP98::KDM5A which led the cells towards myeloid differentiation. 12 genes out of 45 direct NUP98::KDM5A target genes were shown to be essential for NUP98::KDM5A driven AML growth using an invitro model. Targeting of CDK12, one of the direct gene targets, reduced the survival advantage of NUP98::KDM5A AML cells, emphasizing CDK12 inhibition as a potential therapeutic option against NUP98::KDM5A fusion gene driven AML.

The manuscript is well-written and the experiments support the conclusions, for the most part.

The major criticism of the manuscript are 1) many of the experiments seem to depend on results with a single NUP98::KDM5A murine cell line, and 2) there are no control experiments addressing the effect of the CDK12 inhibitor on WT hematopoietic cells, or on other tissues in vivo. Thus we don't know if targeting CDK12 is effective in vivo, or whether toxicity of CDK12 inhibition is excessive.

We thank this reviewer for their insightful comments. We have done our best to respond to the criticism by performing additional experiments and analyses, which we describe in detail in our response below.

Specific comments:

1) HOXA genes, especially HOXA7-11, are well known target genes for NUP98 fusion proteins, and the authors see increased HOXA mRNA expression as one of their major findings. Same question for NKX2-3. Why do the authors think that their ATAC assay didn't identify HOXA genes in Supp table 2? Also, can the authors compare their data with published ATAC and/or NUP98 fusion CHIP target datasets?

While we found that chromatin around most *HoxA* genes and the *Nkx2-3* gene was slightly more accessible in NUP98-rearranged AML cells than in GMP samples, the differences were below the threshold we used to annotate differentially accessible regions ($\log_2FC < -1$ in GMP vs. NUP98-rearranged AML).

While we agree with the reviewer that it would be interesting to compare our data to other ATAC-seq or ChIP-seq datasets of AML samples with NUP98 fusions, we are not aware of any other datasets in addition to the ones that we use in this work. However, the revised version of the manuscript contains a series of novel analyses of these datasets that should help to increase the mechanistic understanding of our results.

2) Fig 1E, it would be useful to see WT GMP comparison here.

We have added the data of 10 ATAC-seq samples from normal GMPs in the heatmap shown in Figure 1E (**Reviewer Figure 6**). As we feel that the inclusion of these samples does not provide any novel information to these results, we refrain from including this version in the revised version of the manuscript.

Reviewer Figure 6: Heatmap representation of ATAC-seq data of indicated samples for 893 regions that show differential accessibility in NUP98-rearranged AML vs. NUP98 wild type AML samples.

3) S1E ??? Can the authors show some representative FACS results?

Representative FACS plots have been added to Supplementary Figure 2E, F in the revised version of the manuscript.

4) Fig 2 E, F indicate that the cells are arrested at G1-cell cycle phase at day 3 and are followed by apoptosis at day 7 and 10. In Fig 2G ??? they are showing the cells are differentiating. What time point do cells start to differentiate ? Also, why do the DMSO controls show less apoptosis at d. 10? Are the authors splitting or ???feeding??? the cells fresh media?

Thank you for these questions. In Figure 2G we show that the morphology of AML cells is changing from immature, blast-like towards cells with the characteristic phenotype of differentiated myeloid cells (i.e. accumulation of cytoplasmic granules, condensation and lobulation of nuclei) on day 3, and these changes become more pronounced on day 5 of dTAG13 treatment. Terminal myeloid differentiation on day 7 is confirmed by a strong increase in the surface expression of differentiation markers Gr-1 and Mac-1 and near-complete loss of c-kit (Figure 2H, I, J). We have added time points to the x-axis of the schematic figure in Supplementary Figure 2I to clarify this point.

In our experimental setup, cells are subjected to splitting every 2-3 days in assays with prolonged dTAG13 treatment. This includes the addition of fresh media as well replenishment of any treatment/stimulant. Therefore, slight variations in proliferation, differentiation or apoptosis might be apparent depending on the timepoint of individual measurements.

5) The authors focus on a single NUP98::KDM5A/Nras cell line, that seems to have evolved in vivo. Did the cell line have insertional activation of important cancer genes? Or spontaneous mutations of cancer genes? These could strongly effect the phenotype, including downstream gene activation. The overall manuscript would be improved if the authors could show key datasets were reproducible with 3 or more independent NUP98::KDM5A/Nras cell lines.

The reviewer is right. Our approach of generating cell lines involves the introduction of transgenes into normal murine fetal-liver derived hematopoietic stem/progenitor cells, the serial passaging of these cells in recipient mice and the establishment of cell lines from leukemia-bearing animals. We have used similar approaches in previous studies^{1,3,6}. While we did not systematically test for the activation of oncogenes/inactivation of tumor suppressor genes, we are confident that most of these pathways are still active in our cell lines. Evidence for this is obtained from our genome-wide CRISPR/Cas9 screens, in which we find that the mutational inactivation of important tumor suppressor genes such as *Trp53* or *Kdm5c* cause a proliferative advantage of cells, while mutational disruption of oncogenes like *Myc* or *Brd4* induce decreased proliferation. To illustrate this, we highlighted these genes in the volcano plot in Supplementary Figure 5B in the revised version of the manuscript. Furthermore, we would like to emphasize that several NUP98::KDM5A-expressing cell lines have been used in this study, as is described in the Methods section “Cell culture” in Table 1 “Murine NUP98::KDM5A cell lines”.

6) Can the authors show an intersection of target genes from their RNA-Seq, Chip Seq, and ATAC Seq datasets?

To answer this question, we intersected the list of genes that are maintained by the NUP98::KDM5A fusion (87 genes in the early down cluster from the dTAG13 8h/24h RNA-seq experiment, Figure 3D) with the 832 genes bound by the NUP98::KDM5A fusion as measured by ChIP-seq as reported earlier by us¹ and 9839 genes linked to accessible regions as measured by ATAC-seq in dTAG-NUP98::KDM5A cells. The intersection of these datasets contained 41 genes, and this list contains several of the direct target genes identified by our SLAM-seq approach including *Cdk12*, but also other genes with roles in AML, such as *Hoxa7*, *Igf2bp3* or *Bcl11a*. The intersection of RNA-seq/ChIP-seq/ATAC-seq datasets is a common strategy to define bona fide direct target genes, and we have used a similar approach to identify target genes of three NUP98 fusion oncoproteins in the past¹. However, we would argue that the approach of using SLAM-seq after dTAG13-induced degradation of the NUP98::KDM5A fusion oncoproteins represents a more direct way of identifying target genes of this fusion oncoprotein.

7) Fig S2 seems to be based on a previously published dataset. Can the authors explain this dataset? Is it mouse or human, etc? What are the ???other genes??? in figure S2I? How were ???target genes??? and ???other genes??? defined?

We apologize for not being clear enough. The analysis referred to by the reviewer is based on a published ChIP-seq dataset for NUP98::KDM5A¹. We generated an AML cell line expressing HA-tagged NUP98::KDM5A as outlined above (our response to point #5 of this reviewer) and ChIP-seq was performed with an anti-HA antibody. These data are explained in detail in Figure 4 of the Schmoellerl et al 2020 work. In contrast to this, ChIP-qPCR and CUT&Tag for H3K27ac and H3K4me3 histone marks was performed in the dTAG-NUP98::KDM5A cell line model that was generated in this study.

The 38 target genes (as described in Supplementary Figure 4C of the revised manuscript) were identified by intersecting genes that exhibit differential H3K27ac marks upon NUP98::KDM5A degradation and genes whose expression was downregulated upon 8h of dTAG13 treatment. The

38 target genes highlighted in orange in Supplementary Figure 4B are listed in Suppl. Table 5. “Other genes” refers to all other genes except those 38 target genes.

8) Line 222 ??? 228 ??? The authors have stated that NUP98::KDM5A actively maintains H3K27ac marks. How does the fusion protein maintain the histone mark, does it protect them from deacetylases ? maybe they can show if there is an effect on deacetylases ?

We thank the reviewer for this question. The rapid loss of the H3K27ac mark upon NUP98::KDM5A degradation indeed suggests that the fusion oncoproteins actively protects these regions from histone deacetylases (HDACs). To investigate this, we analyzed if the expression of histone deacetylases is directly affected by the NUP98::KDM5A fusion oncoprotein. We did not find any evidence for a transcriptional deregulation of HDACs within 8 h or 24 h after dTAG13-mediated NUP98::KDM5A degradation. However, several studies have shown that NUP98::KDM5A acts in the context of chromatin-associated biomolecular condensates that may serve to concentrate the molecular machinery that is necessary for the transcriptional activation of target genes⁸. Thus, in an alternative scenario, NUP98::KDM5A-containing biomolecular condensates could induce the physical exclusion of HDACs at transcriptional hubs and condensates to prevent the inactivation of gene transcription at target loci. We discuss this aspect in the Discussion section of the manuscript.

9) Fig 6H ??? The images are not very clear. Also, maybe they can add immunofluorescence images for CDK12 in the same panel.

We apologize for the suboptimal quality of the images. The size of immunofluorescence images has been increased in the revised version of the manuscript. High resolution images will be uploaded with the final version of the manuscript, provided our work will be accepted for publication. We tested several commercial antibodies for murine CDK12, but none of them provided data of acceptable quality. Unfortunately, all published antibodies only recognize human CDK12. Therefore, we cannot provide immunofluorescence images of CDK12 in our murine cell model.

10) The supplemental tables should be supplied as sortable Excel files; the pdfs are quite difficult to read.

The submission system of *Nature Communications* only allowed to upload Supplementary Files in .pdf format. Excel files will be provided in case this work is accepted for publication.

Reviewer #5 (Remarks to the Author):

Reviewer #6, expertise in AML, nascent transcription, CUT&Tag and ATAC-seq (Remarks to the Author):

In this paper, the authors wanted to better understand NUP98::KDM5A fusion protein driven AML. They started out by comparing ATAC-seq across multiple normal cells and patient samples. NUP98::KDM5A are closest to GMPs but also contain unique open chromatin regions. They performed Cut&Tag in a single PDX model and found these regions are generally enriched for H3K27ac and H3K4me3. A dTAG-NUP98::KDM5A construct was used to transform mouse fetal liver HSCs. By intracellular flow, dTAG treatment induced a complete loss of NUP98::KDM5A and led to cell cycle arrest, differentiation and apoptosis of the AML cells. By ChIP Q-PCR, there was a loss of H3K27ac levels at Meis1 and Hoxa9 promoters by 3 hours of dTAG treatment. H3K27ac Cut&Tag and RNAseq at 8 hours identified several regions of reduced H3K27ac that were also associated with reduced gene expression, although Meis1 did not display reduced expression until the 24 hours timepoint. Overall, the authors identified 38 genes that appear to be sensitive to NUP98::KDM5A regulation. They then performed SLAM-seq after 1 hour dTAG treatment to identify genes that displayed rapid loss of transcription upon NUP98::KDM5A degradation. Comparing this with NUP98::KDM5A ChIP-seq, they identified 45 gene targets that are direct targets of NUP98::KDM5A regulation. They performed a CRISPR screen in the NUP98::KDM5A cell line and found that 12 of the 45 direct targets are essential for cell survival. They further validated some of these targets and then focussed in CDK12. NUP98 fusion patient samples display higher expression levels of CDK12 than other AMLs. NUP98::KDM5A cells are sensitive to CDK12 KD and inhibitor treatment, and loss of CDK12 activity increased DNA damage.

Overall, the experiments are well performed and technically sound. I have two general concerns and several more specific concerns.

We thank this reviewer for their positive and supportive comments to our manuscript. We performed several experiments and analyses to address the points raised by this reviewer, and we are convinced that addressing these points has contributed to an improvement of the manuscript during the revision process. Our responses to the reviewer's comments are outlined in detail below.

General concerns

1. The mechanistic data is somewhat descriptive, and it is not entirely clear what NUP98::KDM5A binding does to gene regulation. For instance, the authors show a strong correlation between H3K27ac levels (and H3K4me3) and changes in NUP98::KDM5A binding. From this, they propose that NUP98::KDM5A directly controls the epigenetic landscape at target genes. However, it is also possible that NUP98::KDM5A impacts transcription in some other way (e.g., it could demethylate a non-histone target), and changes in H3K27ac are indirect effects. In support of this, the changes in H3K27ac levels don't always correlate with changes in gene expression upon dTAG treatment. For example, according to Figure S2H, 67 genes show reduced expression with no impact on H3K27ac levels. Also, why are some genes sensitive to loss of NUP98::KDM5A binding while others are relatively non-responsive? Some additional work on the mechanism would be helpful.

We thank the reviewer for these questions. While we believe that our data add significant insights to the mechanisms of target gene activation by NUP98 fusion oncoproteins, we agree that despite our efforts, some aspects of the mechanism could not be fully elucidated. However, the NUP98::KDM5A fusion oncoprotein itself cannot demethylate any targets because the enzymatic JmjC domain of KDM5A is not retained in the coding region of the fusion oncoprotein. Alternatively, NUP98::KDM5A might influence the deposition of histone marks by other mechanisms. Several studies have shown that NUP98::KDM5A and other NUP98 fusion oncoproteins act in the context of chromatin-associated biomolecular condensates that may serve to concentrate the molecular machinery that is necessary for the transcriptional activation of

target genes⁸. Thus, in an alternative scenario, NUP98::KDM5A-containing biomolecular condensates could induce the recruitment of histone acetyltransferases or other epigenetic regulators to target loci. The same mechanism could account for the physical exclusion of antagonizing regulators, such as histone deacetylases at transcriptional hubs and condensates to prevent the inactivation of gene transcription at target loci. We discuss this aspect in the Discussion section of the manuscript.

Finally, we would like to emphasize that we only investigated the activating histone marks H3K27ac and H3K4me3 in the present study. Thus, other epigenetic modifications that were not investigated in this work might explain why the expression of individual genes does not change upon loss of H3K27ac/H3K4me3 and vice versa.

2. The authors propose that CDK12 could be a new therapeutic target in NUP98::KDM5A AML. This is an interesting possibility but there are several issues with this idea. First of all, inhibiting kinases can have harsh toxic side effects. It would be useful to know how well normal cells respond to CDK12 KD or inhibition. Also, it would be useful to know if CDK12 dependency is specific to NUP98::KDM5A, or is a more general AML target. In addition, without in vivo work it is difficult to assess how good of a target this really is (see points 7-10 below).

We thank the reviewer for these comments, which we address in our responses to the specific comments #8-10 below.

Some specific concerns are outlined below:

1. A dTAG western blot would be useful, to validate the intracellular flow which can sometimes produce artifacts.

Western Blot analysis of NUP98 fusion proteins has proven difficult. While validated antibodies recognizing proteins involved in the fusion exist, they also recognize the endogenous counterparts of the fusion oncoprotein. This can complicate the analysis because of overlapping molecular weights of endogenous proteins vs. the fusion oncoprotein. Thus, affinity tags have to be used to selectively detect the fusion oncoprotein. Furthermore, low expression levels, their low degree of stability and their relatively large size further complicate their detection via immunoblotting. Thus, despite intense efforts, we have not managed to reliably detect NUP98 fusion oncoproteins via western blotting in the murine AML cell lines generated by us.

2. Why was the 8 hour timepoint chosen for the Cut&Tag and RNA-seq experiments when it looks like there is a significant H3K27ac reduction by 3 hours?

The 8 hour time point was chosen because in our initial ChIP-qPCR analyses (Supplementary Figure 3A) the dTAG13-induced reduction of H3K4me3 levels on the promoters of important NUP98::KDM5A target genes reached stable levels after 6 hours. As we aimed to perform all analyses at the same time point, we chose the 8 hour time point after dTAG13 mediated NUP98::KDM5A degradation, as immediate and direct effects on both H3K27ac and H3K4me3 levels were presumed to prevail over any potential secondary effects.

3. Why do some genes show changes in H3K27ac without displaying changes in transcription? Conversely, why do some genes show changes in transcription without showing changes in H3K27ac (or H3K4me3)? Does this data suggest that NUP98::KDM5A controls gene activation through mechanisms other than altering the epigenome?

We only investigated the activating histone marks H3K27ac and H3K4me3 in the present study. Thus, other epigenetic modifications that were not investigated in this work might explain why the expression of individual genes does not change upon loss of H3K27ac/H3K4me3 and vice versa. Please also see our response to general concern #1 above.

4. Is it possible that NUP98::KDM5A is having global effects on either H3K4me3 or H3K27ac?

Global effects can sometimes cause artifacts when normalizing genomic data for analysis. This can be easily checked using western blots for these and other marks in dTAG vs DMSO treated samples.

We thank the reviewer for this question. We treated dTAG-NUP98::KDM5A cells with dTAG13 (35 nM) for 8 hrs. Lysates were subjected to Western Blot analysis for total H3, H3K27ac and H4K4me3. We did not detect any changes in the global levels of these histone marks or in total H3 levels. These new results are shown in Supplementary Figure 3B in the revised manuscript.

5. Where is *Meis1* on Figure 3G?

Meis1 is among the top 10% of genes with the highest H3K27ac signal in dTAG-NUP98::KDM5A cells (Reviewer Figure 7).

Reviewer Figure 7: Hockeystick plot of H3K27ac signal in dTAG-NUP98::KDM5A-driven AML cells. 38 NUP98::KDM5A target genes are represented in orange. *Meis1* is shown red.

6. Considering the SLAM-seq experiments, does loss of H3K27ac precede reduced transcription or occur afterwards? In other words, are the high H3K27ac levels at NUP98::KDM5A target genes a cause or consequence of transcription?

While our SLAM-seq experiment was performed after 1 h of NUP98::KDM5A degradation the earliest time point of H3K27ac CUT&Tag after NUP98::KDM5A degradation was 8 hrs. Our H3K27ac ChIP-qPCR on the promoters of *Meis1* and *HoxA9* show a significant decrease of the H3K27ac mark at 3 h after NUP98::KDM5A degradation. As we do not have H3K27ac data from earlier time points we can only speculate about the order of events, but favor a scenario in which NUP98::KDM5A nucleates the formation of chromatin-associated biomolecular condensates that recruit epigenetic regulators and transcription factors to target loci. Targeted degradation of the NUP98::KDM5A would cause the dissociation of these structures, leading to loss of H3K27ac and transcription at target gene loci. We are currently working on a follow-up experiment to determine the exact order of events in this regard, but the results are not available yet.

7. Based on previous NUP98::KDM5A ChIP-seq data, how many genes are bound by NUP98::KDM5A that don't change after degranulation treatment? In other words, how many genes bound by NUP98::KDM5A are insensitive to its regulation?

We find that only a relatively small fraction of genes (13%) that show differential expression upon NUP98::KDM5A degradation (after 24 h) are also bound by the NUP98::KDM5A fusion oncoprotein (from ChIP-seq data published earlier)¹. Conversely, 96% of genes that are bound by NUP98::KDM5A do not change their expression within 24 hrs of fusion oncoprotein degradation. Our dataset does not provide enough temporal resolution to determine how many of the NUP98::KDM5A-bound genes change their expression at later time points. In any case this indicates that the commonly used strategy to infer direct transcriptional targets by intersecting differentially expressed genes with genes whose promoter is bound by the protein of interest is associated with significant noise, as the consequences of promoter binding of a transcriptional regulator might be highly context-dependent. Furthermore, deregulated genes observed at later stages after NUP98::KDM5A degradation might be secondary effects and there might be other mechanisms than only NUP98::KDM5A chromatin binding that initiate their deregulation.

8. How specific is CDK12 dependency to NUP98 leukemias? Are other AMLs also sensitive to CDK12 KD?

This comment is in line with comment #10 of Reviewer 1 and comment #10 of Reviewer 2. We performed several experiments to address this point. First, we knocked down *Cdk12* via shRNA in murine AML cell lines driven by the *KMT2A::MLL3* fusion and N-terminal CEBPA mutations, respectively. *Cdk12* knockdown caused significant antiproliferative effects in these cell models. Furthermore, we tested the effect of the CDK12 inhibitor THZ531 and the CDK12 degrader BSJ-4-116 in additional murine and human AML models expressing different driver mutations. These results clearly indicate that both inhibition of CDK12 kinase activity as well as CDK12 degradation was incompatible with AML cell proliferation, in line with data reported by Savoy et al². Although we observed differences in the sensitivity of AML cells expressing different driver mutations to THZ531 or BSJ-4-116 we did not find a resistant cell line. Therefore, we conclude that CDK12 represents a vulnerability of most, if not all AML subsets. These new results are shown in Supplementary Figure S6B, C (*Cdk12* shRNA), Figure 7 (CDK12 inhibition by THZ531) and Supplementary Figure S7B (CDK12 degradation by BSJ-4-116) in the revised manuscript.

9. How sensitive are normal hematopoietic cells to CDK12 KD or BSJ-4-116 treatment?

Thank you for this important question. We performed drug sensitivity assays with bone marrow mononuclear cells (MNCs) and CD34+ progenitor cells from healthy donors and compared the sensitivity of THZ531 and BSJ-4-116 to primary human AML patient samples with NUP98 rearrangements. The sensitivity of healthy cells and AML cells to the CDK12 degrader BSJ-4-116 was comparably low (GI₅₀ values between 4 and 15 nM). This could indicate that complete degradation of CDK12 causes effects that are too strong to observe differential sensitivity between healthy and AML cells. Alternatively, BSJ-4-116 could have some unknown off-target effects that cause enhanced cytotoxicity. These results are shown in Supplementary Figure 7A in the revised version of the manuscript.

Importantly, however, we found that healthy MNCs as well as normal CD34+ progenitor cells were up to 10 times less sensitive than NUP98::KDM5A and NUP98::NSD1 AML cells to the CDK12 inhibitor THZ531, indicating a possible therapeutic window. These important new results are shown in Figure 7A of the revised version of the manuscript. These data are also in line with results from the rescue experiments shown in Figure 6D and Figure 7F of the revised manuscript, which clearly show that the kinase activity of CDK12 is required for the proliferation of NUP98::KDM5A cells.

10. To claim that CDK12 is a possible therapeutic target, there needs to be some in vivo evidence for this, preferably using a PDX model if possible rather than the dTAG-NUP98::KDM5A model.

To validate *Cdk12* as an essential direct target gene for NUP98::KDM5A AML cells, we performed an in vivo RNAi experiment. We introduced Doxycycline-inducible variants of *Cdk12*-targeting shRNAs into NUP98::KDM5A AML cells (CD45.2) stably expressing the reverse Tet-transactivator (rtTA3). In the system used by us, shRNA expression is coupled to an IRFP670 marker, which enables the identification of shRNA-expressing cells by flow cytometry. Cells were transplanted into recipient CD45.1 animals and expression of the shRNAs was induced by addition of Doxycycline to the drinking water of mice three days after transplantation. Disease progression was monitored by in vivo bioluminescence imaging, as the cells also express a luciferase gene¹. Knockdown of *Cdk12* significantly extended the survival of recipient animals, showing that CDK12 is also required for the growth of NUP98::KDM5A AML cells in vivo. Immunophenotyping of leukemic blasts in moribund animals showed that *Cdk12*-deficient, IRFP670/CD45.2-positive AML cells displayed reduced levels of the progenitor marker c-Kit, which is in line with their reduced leukemogenic potential. Altogether, these data strengthen the conclusion that CDK12 represents a faithful target in NUP98::KDM5A-driven AML. These results are shown in Figure 6E-G and in Supplementary Figure 6D in the revised version of the manuscript, and the comments to these points are related to our response to point #12 of Reviewer 1 and point #5 of Reviewer 3.

References

1. Schmoellerl, J. *et al.* CDK6 is an essential direct target of NUP98 fusion proteins in acute myeloid leukemia. *Blood* **136**, 387–400 (2020).
2. Savoy, L. *et al.* CDK12/13 dual inhibitors are potential therapeutics for acute myeloid leukemia. *Br. J. Haematol.* (2023). doi:10.1111/BJH.18843
3. Terlecki-Zaniewicz, S. *et al.* Biomolecular condensation of NUP98 fusion proteins drives leukemogenic gene expression. *Nat. Struct. Mol. Biol.* **28**, 190–201 (2021).
4. Ahn, J. H. *et al.* Phase separation drives aberrant chromatin looping and cancer development. *Nat.* 2021 5957868 **595**, 591–595 (2021).
5. Chandra, B. *et al.* Phase Separation Mediates NUP98 Fusion Oncoprotein Leukemic Transformation. *Cancer Discov.* **12**, 1152–1169 (2022).
6. Schmoellerl, J. *et al.* EVI1 drives leukemogenesis through aberrant ERG activation. *Blood* **141**, 453–466 (2023).
7. Böskén, C. A. *et al.* The structure and substrate specificity of human Cdk12/Cyclin K. *Nat. Commun.* **5**, (2014).
8. Jevtic, Z., Allram, M., Grebien, F. & Schwaller, J. Biomolecular Condensates in Myeloid Leukemia: What Do They Tell Us? *HemaSphere* **7**, e923 (2023).

Response to reviewers' comments to manuscript NCOMMS-24-13705A

Reviewer #2 (Remarks to the Author):

I appreciate the authors' significant and earnest efforts to address reviewer comments, particularly with regard to the ATAC-seq dataset. While many of the additional analyses showed modest, subtle or no detectable effect of the fusion on the transcriptome (e.g. S1D), this is likely due to difficulties integrating different data types rather than major issues with interpretations or biological systems. I do think the authors' initial presentation of the CDK12 dependency suggested specificity for NUP98 AMLs, and appreciate that this is now recognized as a general dependency across most AMLs. The NUP98 fusion may provide one route to a dependency on this kinase. The manuscript has ATAC-seq and CUT&Tag datasets that are likely to be useful for the field, and the murine HSPC degree model is also a useful resource. The bulk of the manuscript, the model, and the datasets as presented raise no concerns that they are likely to be invalidated by additional analyses or NUP98 models. I have no further recommendations and believe the manuscript is suitable for publication in Nature Communications.

We thank this reviewer for their comments.

Reviewer #3 (Remarks to the Author):

In this study, Troester et al. investigated the epigenetic and transcriptional signatures controlled by NUP98 fusion proteins in AML, with a primary focus on the specific NUP98-KDM5A fusion oncoprotein, which is observed in ~2% of all pediatric AML patients. Using a series of orthogonal approaches, the authors identified 12 direct NUP98-KDM5A target genes that are essential for AML cell growth, including the cyclin-dependent kinase 12 (CDK12). Targeting Cdk12 resulted in a significant down-regulation of genes linked to DNA repair and induced the formation of γ -H2AX foci in AML cells, suggesting that CDK12-dependent control of DNA repair gene expression may promote the survival of NUP98-KDM5A AML cells in vitro.

Overall, the authors have made a significant and careful effort to address the elements raised in my initial review, and I want to applaud them for that.

However, one remaining and central issue is the focus on DNA repair at end of the manuscript. While the presence of γ -H2AX foci suggests some involvement of CDK12 in DNA repair, the data presented are largely correlative. Without direct evidence of a DNA repair deficiency upon targeting CDK12, the strong focus on the role of CDK12 in DNA repair—as reflected in the title—may be somewhat misleading at this stage. I would recommend tuning down the emphasis on DNA repair and ensuring that the conclusions more accurately reflect the current data, which suggest a broader role for CDK12 beyond just DNA repair.

We thank this reviewer for their supporting comments. We introduced the following changes to the manuscript to tune down the emphasis on DNA repair processes in the context of CDK12:

- We changed the title to “Transcriptional and epigenetic rewiring by the NUP98::KDM5A fusion oncoprotein directly activates CDK12”.
- We adjusted the statements about the involvement of DNA repair mechanisms in the relevant sections of the results and discussion to clarify that the dependency of NUP98::KDM5A on CDK12 is likely due to the diverse roles of CKD12, and that our results show that efficient DNA repair gene expression might be one of them.

Reviewer #4 (Remarks to the Author):

The authors have added a large amount of new data, and addressed most of the questions that I raised.

Unfortunately, I do not think that the authors have adequately addressed the two major criticisms that were raised by this and other reviewers.

We were sorry to hear that our previous comments were not clear enough to substantiate the points that this reviewer raised. We provide a more detailed answer to the remaining open points below.

1) Many experiments were based on a single murine NUP98::KDM5A/Nras cell line, that selectively grew out after serial passage. What enabled this specific clone to outcompete and grow in vitro as a cell line? Did this cell line acquire mutations in genes that conferred a fitness advantage (in vivo or in vitro)? Or was there an integration site effect that conferred a fitness advantage? To confirm that downstream effects on gene expression, etc are indeed due to the NUP98::KDM5A fusion, I think it is important to show that these effects (for instance, global gene expression profiles, chromatin accessibility) can be seen in at least one other NUP98::KDM5A cell line.

Our approach to generate cellular models for NUP98 fusion oncoprotein-driven AML via transplantation of virally transduced murine HSPCs into irradiated recipients followed by the establishment of stably growing cell cultures is motivated by the fact that no human cell line models exist that can be used to study NUP98 oncofusions. The approach used by us is very robust, and the data in this manuscript as well as in a previous report from our laboratory¹ show that the cell cultured obtained through this procedure are faithful models of human AML with NUP98 fusion oncoproteins.

The robustness of our models is in part driven by the co-expression of NUP98::KDM5A and mutated *NRAS* (G12D), which represents a clinically relevant combination of driver mutations. As we did not systematically test if the cells acquire additional mutations during the establishment of stably growing cell cultures, we cannot rule out that this is the case. Yet, the data from our genome-wide CRISPR/Cas9 screens show that mutational inactivation of important tumor suppressors (*Tp53*, *Kdm5c*, and others) induce a fitness advantage. This indicates that these tumor suppressor genes are functional in our cell lines and have not been inactivated during their establishment. We also cannot rule out that integration site effects cause a proliferative advantage. However, our experimental strategy likely results in multiple viral integrations in the same cell, so we do not suspect that this contributes to increased fitness of our cells.

While it is true that many experiments shown in this manuscript were performed in the same dTAG-NUP98::KDM5A cell line, we would like to stress that this manuscript features six different cellular models of NUP98::KDM5A-driven leukemia. All these models have been independently established, extensively characterized, and several of them have been published by our laboratory before.¹

A comprehensive list of the cellular models used in this work is shown in the following:

- The **dTAG-NUP98::KDM5A** model was used for initial global transcriptional and epigenetic analysis of immediate fusion oncoprotein loss (**Figs. 3 & 4**).

- A cell line that constitutively expresses **NUP98::KDM5A and Cas9** was used for the identification of genetic dependencies via genome-wide CRISPR screening (**Fig. 5A, B**). The results from this cell line also validate the essentiality of DNA repair-associated genes, as indicated from data in the dTAG-NUP98::KDM5A model and the Tet-Off model (**Fig. S7E**).
- A cell line model that constitutively expresses **rtTA3 and NUP98::KDM5A** was used for the validation of the 12 essential direct target genes in vitro (**Fig 5C, D, 6C**) and in vivo (**Fig. 6E, F, G**) via doxycycline (dox)-inducible RNAi. Results from this cell line model therefore validate targets that have been identified with the dTAG-NUP98::KDM5A model and the NUP98::KDM5A Cas9 cell line.
- A cell line expressing **Strep-HA-tagged NUP98::KDM5A** was used for ChIP-seq.¹ Results from these cells validate binding of the NUP98::KDM5A fusion protein to promoters of transcriptionally and epigenetically regulated genes (38 target genes, **Fig. S4E**) and 45 direct target genes (**Fig. 4D**), as identified in the dTAG-NUP98::KDM5A model.
- RNA-seq data from a **Tet-Off NUP98::KDM5A model**¹ confirm upregulated expression of 45 direct target genes compared to controls (**Fig. 4H**) and downregulation of DNA repair-associated genes upon dox-induced repression of NUP98::KDM5A expression (**Fig. 7D, E**), which was also reflected in increased DNA damage upon NUP98::KDM5A degradation in the dTAG-GFP-NUP98::KDM5A model (**Fig. S7G**).
- Results from a **dTAG-GFP-NUP98::KDM5A** model confirmed the increase of DNA damage upon pharmacological CDK12 perturbation (**Fig. 7G, S7F**), strengthening the role of CDK12-controlled DNA repair gene expression as hypothesized from results obtained in the dTAG-NUP98::KDM5A model (**Fig. 7C, S7C, D**).
- The dose-dependent sensitivity to pharmacological CDK12 perturbation is shown in three different NUP98::KDM5A cell lines (dTAG, dTAG-GFP and rtTA3 NUP98::KDM5A models).
- Our direct comparison of ATAC-seq and RNA-seq data of the dTAG-NUP98::KDM5A model with data from NUP98::KDM5A AML patients (**Fig. S2G, S3F**) revealed a strong correlation. Furthermore, we confirm findings from our mouse models in data from human AML patients (**Fig. 4I, 6A, S4B, S4D, S6A**) and in an independent PDX model (**Fig. 5F, 6B**).

To clarify this aspect further, we have added references to Table 1 (listing all the mouse models that were used in this work) throughout the manuscript text.

2) The critical question regarding efficacy and toxicity in vivo is not adequately addressed. The experiment shown in Fig 6 seems to show a subtle survival advantage (11 or 14 d vs 9 d), that would be clinically insignificant. However, there is no mock transplant group, so it isn't clear that the mice died of leukemia vs radiation toxicity. But the most significant criticism is that this experiment does not address the toxicity of CDK12 inhibition on WT hematopoiesis, or other organ function. Despite the in vitro experiments shown in Fig 7, it is not clear that there is a therapeutic window in vivo.

While this comment addresses an important point, we are not sure if we communicated all aspects related to this point before. We attempt to provide further clarifications below:

- The survival advantage upon *in vivo* shRNA-mediated knockdown of *Cdk12* (**Fig. 6E-F**) might seem subtle but is highly significant in the light of the model and the experimental strategy. First, the model is very aggressive, as it causes a lethal disease with a latency of 9 days. Second, our *in vivo* shRNA experiment mimics a

setting of therapeutic intervention, as we only start adding doxycycline (to induce expression of the shRNA) to the drinking water of recipient animals after leukemia cells can be identified via *in vivo* bioluminescence imaging, which is usually 2-3 days after transplantation. Thus, given the aggressivity of the model and the experimental settings, we argue that the survival benefit of 22% and 55%, of two different *Cdk12*-targeting shRNAs compared to the control cohort is substantial.

- We can exclude radiation toxicity as the cause of death of recipients, because our transplantation settings only use sublethal irradiation (4.5 Gy). We have added this information to the revised version of the manuscript. Furthermore, analysis of the bone marrow of moribund mice shows >90% CD45.2+ transplant-derived leukemia blasts in the bone marrow, indicating development of a highly aggressive, lethal disease in these recipients (**Fig. S6D**).
- As the CDK12 inhibitor THZ531 is not optimized for *in vivo* use we can only show a potential therapeutic window in cultured cells. Others have performed animal studies using the precursor compound THZ1² and the CDK12/13 inhibitor BSJ-01-175³. While this work was in revision, the CDK12/13 degrader YJ1206 has been presented.⁴ While all these compounds showed effective cancer cell killing *in vivo*, hematotoxic effects of these treatments have not been reported. Given the strong interest of the cancer research field in modalities to target CDK12, it is likely that advanced chemical compounds will become available in the future that will allow to address the points raised by this reviewer.

Finally, we would like to stress that we do not suggest that CDK12-targeting compounds should be considered for clinical testing right away. However, our data clearly validates CDK12 as an essential direct target gene of the NUP98::KDM5A fusion oncoprotein that is directly tractable. Our work strongly encourages further investigations in a preclinical setting to demonstrate a therapeutic window of CDK12 inhibition *in vivo*. We have clarified the respective statements in the relevant sections of the introduction, results and discussion of the revised manuscript.

Reviewer #5 (Remarks to the Author):

Reviewer #6 (Remarks to the Author):

The authors have done a very nice job with additional experiments showing that CDK12 may be a general AML vulnerability and with the inhibitor at least, there may be a therapeutic window for impacting AML cells without impacting normal cells. However, in the discussion the authors state "these data suggest that the NUP98::KDM5A-mediated active maintenance of H3K27ac and H3K4me3 marks at their target genes plays a pivotal role in the regulation of their expression." Since some genes show changes in H3K27ac without displaying changes in transcription and some genes show changes in transcription without showing changes in H3K27ac (or H3K4me3), I don't think the data supports this statement. In the discussion, the authors should contextualize their ideas about the mechanism a bit better, and emphasize some alternative models for how the fusion could be functioning, with less of an emphasis on the histone marks which don't seem to correlate all that well with the function of the fusion protein.

Otherwise, the authors have answered all my questions and this is a very nicely done study.

We thank this reviewer for their positive evaluation. We have introduced the following changes to address their comments:

- The quoted statement in the discussion was adjusted to achieve a more balanced representation of our findings.
- The discussion section features potential models of how NUP98 fusions could regulate epigenetic and transcriptional changes. We discuss how NUP98-fusion oncoprotein-containing biomolecular condensates regulate the recruitment of epigenetic modifier complexes (CBP/p300, MLL1/MENIN) and/or the active exclusion of histone deacetylases and histone demethylases from target loci. In addition, we have added a statement about the likely, but unknown effect of other histone marks (besides H3K27ac and H3K4me3) that might be orchestrated by NUP98 fusion proteins.

References:

1. Schmoellerl, J. *et al.* CDK6 is an essential direct target of NUP98 fusion proteins in acute myeloid leukemia. *Blood* **136**, 387–400 (2020).
2. Amanda Balboni Iniguez, A. *et al.* EWS/FLI Confers Tumor Cell Synthetic Lethality to CDK12 Inhibition in Ewing Sarcoma . *Cancer Cell* **33**, 202-216.e6 (2018).
3. Jiang, B. *et al.* Structure-activity relationship study of THZ531 derivatives enables the discovery of BSJ-01-175 as a dual CDK12/13 covalent inhibitor with efficacy in Ewing sarcoma. *Eur. J. Med. Chem.* **221**, 113481 (2021).
4. Chang, Y. *et al.* Development of an orally bioavailable CDK12/13 degrader and induction of synthetic lethality with AKT pathway inhibition. *Cell Reports Med.* **5**, 101752 (2024).

Response to reviewers' comments to manuscript NCOMMS-24-13705B

Reviewer #4 (Remarks to the Author):

The authors have presented additional data and arguments to address the remaining questions.

1) The authors didn't present additional data (eg, whole exome sequencing, insertion site sequencing, or analysis of additional NUP98::KDM5a/Nras cell lines) to demonstrate that the phenotype of the NUP98::KDM5a/Nras cell line was not, at least in part, driven by additional acquired mutations. They do present a reasonable argument that additional NUP98::KDM5A (no Kras) cell lines behave similar to the NUP98::KDM5A/Nras cell line in various in vitro experiments.

2) The authors consider an increase from 9 days survival to 11 days survival in a single experiment to be a substantial response. This reviewer would be more cautious in interpretation of that data. Especially in light of the fact that there was no difference in percent malignant cells in the bone marrow at time of death (Fig S6D). The argument that the model is too aggressive to detect clear differences is somewhat weak; I would guess that the model could be made less aggressive by simply injecting fewer cells. It also would have been more convincing if the authors had shown clear evidence of AML (CBC abnormalities, invasion of nonhematopoietic tissue) as recommended by the MMHC consortium (PMID: 12070033), and had a radiation alone control, as sublethal radiation alone is not benign, and will lead to pancytopenia and weight loss at 7-14 days post radiation. The revised discussion is improved.

We thank these reviewers for their comments. We agree that more work needs to be performed to address the in vivo relevance of *CDK12* inactivation in models of AML with NUP98 fusions and beyond.

Reviewer #5 (Remarks to the Author):
